# Towards Universal Offline Black-Box Optimization via Learning Language Model Embeddings

**Rong-Xi Tan** [* 1 2]  **Ming Chen** [* 1 2]  **Ke Xue** [† 1 2]  **Yao Wang** [3]  **Yaoyuan Wang** [3]  **Sheng Fu** [3]  **Chao Qian** [† 1 2]

## Abstract

The pursuit of universal black-box optimization (BBO) algorithms is a longstanding goal. However, unlike domains such as language or vision, where scaling structured data has driven generalization, progress in offline BBO remains hindered by the lack of unified representations for heterogeneous numerical spaces. Thus, existing offline BBO approaches are constrained to single-task and fixed-dimensional settings, failing to achieve cross-domain universal optimization. Recent advances in language models (LMs) offer a promising path forward: their embeddings capture latent relationships in a unifying way, enabling universal optimization across different data types possible. In this paper, we discuss multiple potential approaches, including an end-to-end learning framework in the form of next-token prediction, as well as prioritizing the learning of latent spaces with strong representational capabilities. To validate the effectiveness of these methods, we collect offline BBO tasks and data from open-source academic works for training. Experiments demonstrate the universality and effectiveness of our proposed methods. Our findings suggest that unifying language model priors and learning string embedding space can overcome traditional barriers in universal BBO, paving the way for general-purpose BBO algorithms. The code is provided at https://github.com/lamda-bbo/universal-offline-bbo.

## 1. Introduction

The pursuit of universal black-box optimization (BBO) algorithms, i.e., capable of adapting to diverse problems, has been a long-standing challenge (Wolpert & Macready, 1997; Chen et al., 2022b; Lehre & Lin, 2024). Traditional BBO methods, including Bayesian optimization (Garnett, 2023) and evolutionary algorithms (Bäck, 1996; Zhou et al., 2019), excel in many tasks (Turner et al., 2021). However, these traditional BBO methods are not only confined to single-type and fixed-dimensional settings (Chen et al., 2022b; Song et al., 2025) but also struggle to leverage large-scale offline data (Song et al., 2024c; Wei et al., 2024), heavily relying on online evaluations, which is a severe limitation that becomes particularly challenging in many real-world expensive scenarios.

Offline black-box optimization (Trabucco et al., 2022; Kim et al., 2025) aims to identify optimal designs for an unknown objective function using only a fixed and pre-collected dataset, which attempts to mitigate the issue mentioned above. Existing offline BBO approaches have shown impressive performance, such as in real-world engineering design (Tanabe & Ishibuchi, 2020; Kumar et al., 2022), protein design (Khan et al., 2023; Chen et al., 2023b; Kim et al., 2023), and molecule design (Gaulton et al., 2012; Stanton et al., 2022; Dara et al., 2022; Xue et al., 2024). Their success relies on two assumptions: (1) the training and test tasks share identical input dimensions and variable types, and (2) sufficient historical data exists for each task.

While real-world optimization problems often exhibit inherent correlations across domains (Bai et al., 2023; Wang et al., 2024a), existing offline BBO methods face two challenges: (1) insufficient training data for model development (Nguyen et al., 2023), and (2) fundamental inability to exploit cross-task relationships. The critical barrier lies in the heterogeneity of search spaces (Fan et al., 2024). This limitation forces practitioners to collect large datasets for every new problem, due to their inability to unify parametric representations across domains, which is unsustainable in practical scenarios characterized by sparse data availability and diverse task requirements. Thus, there is an urgent need for universal offline BBO.

---

[*]Equal contribution [1]National Key Laboratory for Novel Software Technology, Nanjing University, China [2]School of Artificial Intelligence, Nanjing University, China [3]Advanced Computing and Storage Lab, Huawei Technologies Co., Ltd., China. Correspondence to: Ke Xue <xuek@lamda.nju.edu.cn>, Chao Qian <qianc@nju.edu.cn>.

*Proceedings of the 42nd International Conference on Machine Learning*, Vancouver, Canada. PMLR 267, 2025. Copyright 2025 by the author(s).

Fortunately, recent advances have demonstrated the feasibility of universal optimization in string-based search spaces using language models (LMs) (Song et al., 2024a;b; Nguyen et al., 2024; Tang et al., 2025). By tokenizing numerical parameters into string sequences, LM-based optimizers outperform traditional regression models in cross-task generalization, particularly when trained on multi-task datasets spanning diverse domains. However, there are still several critical gaps in universal offline BBO. First, prior work does not discuss distinct paradigms in detail, failing to clarify their relative strengths or compatibility. Second, these works neglect the unique requirements of offline BBO, where algorithms must avoid overfitting to limited historical data (Trabucco et al., 2021; Fu & Levine, 2021; Yu et al., 2021; Chen et al., 2022a; Qi et al., 2022; Dao et al., 2024b; Hoang et al., 2024). Besides, the geometric properties of learned latent spaces remain under-explored, despite their direct impact on optimization stability and sample efficiency.

In this paper, we propose a universal string-based offline BBO framework, UniSO. We first introduce several components for UniSO, including string-based data representation, and metadata formulation. Then, we use two model architectures for universal offline BBO, including token-targeted regressor (Song et al., 2024a) and numeric-targeted regressor (Nguyen et al., 2024), formulating two Vanilla UniSO variants, i.e., UniSO-T and UniSO-N, respectively. However, several issues exist when directly applying these Vanilla UniSO variants to offline BBO tasks, as shown in Fig. 3. To address these issues, we propose two improvements, i.e., embedding distribution alignment via metadata guidance and local embedding smoothness enhancement. Experiments demonstrate the universality and effectiveness of our proposed methods, particularly showing that UniSO-T achieves superior cross-task generalization through multi-task training on heterogeneous search spaces.

Our findings reveal three key insights: (1) a unified representation through string-based space enables universal offline BBO possible, (2) geometric regularization of embedding spaces significantly enhances optimization stability, and (3) metadata-guided learning effectively bridges the domain gap between training and unseen test tasks. These advancements collectively overcome traditional barriers in universal offline BBO, paving the way for general-purpose optimization across various types and dimensions.

## 2. Preliminaries

### 2.1. Offline BBO

Given the design space $\mathcal{X}$, where $\mathcal{X}$ could be be CONTINUOUS, INTEGER, CATEGORICAL, or PERMUTATION, and a fixed offline dataset $\mathcal{D}$, offline BBO (Trabucco et al., 2022; Kim et al., 2025) aims to seek an optimal design $\mathbf{x}^*$ that maximizes a black-box objective

function $f : \mathcal{X} \to \mathbb{R}$, i.e., $\mathbf{x}^* = \arg\max_{\mathbf{x} \in \mathcal{X}} f(\mathbf{x})$. This optimization process relies solely on the available dataset, with no online evaluations permitted during optimization. Specifically, an algorithm operates exclusively on a fixed dataset $\mathcal{D} = \{(\mathbf{x}_i, y_i)\}_{i=1}^N$, where each instance $\mathbf{x}_i$ represents a design (e.g., the composition of a DNA sequence), and its corresponding scalar value $y_i = f(\mathbf{x}_i)$ denotes the objective score (e.g., a specific property score of the designed DNA sequence). Besides, for an offline BBO task, there usually exists task-level metadata $m$, which can distinguish it from other tasks and potentially hint the information of the unknown objective function $f$. Thus, similar to the multi-task regression case (Song et al., 2024a), an offline BBO task can be formulated as $\mathcal{T} = (\mathcal{X}, f, \mathcal{D}, m)$.

A prevalent approach for offline BBO is the *forward* approach, which first trains a surrogate model, typically a deep neural network $\hat{f}_{\boldsymbol{\theta}} : \mathcal{X} \to \mathbb{R}$ parameterized by $\boldsymbol{\theta}$, to learn a scoring function for the designs where the output corresponds to the predicting score; then searches for the final design by maximizing the model's output. The scoring function can be learned by regression (Trabucco et al., 2021; Chen et al., 2023a) or ranking (Tan et al., 2025). We provide a comprehensive related work in Appendix A, including the detailed *forward* and *backward* approaches of offline BBO, and LLM for BBO.

### 2.2. Universal Offline BBO

Recent advances in foundational models for black-box optimization research (Song et al., 2024a;b; Nguyen et al., 2024; Tang et al., 2025; Song & Bahri, 2025) have shown the potential of universal optimization in string-based space. Thus, we first extend conventional offline BBO to the universal setting. The goal of universal offline BBO is to **simultaneously** address multiple tasks through a universal foundational model. Consider $n_{\mathcal{T}}$ optimization tasks $\{\mathcal{T}_i\}_{i=1}^{n_{\mathcal{T}}}$ with three key characteristics: (1) **Heterogeneous design spaces**: Mixed-type parameters and varying dimensionality across tasks; (2) **Divergent objectives**: Unique optimization functions for each task; and (3) **Task-specific metadata**: Distinct auxiliary information accompanying each task.

While more challenging than traditional single-task BBO, the universal approach offers two key advantages. First, it enables knowledge transfer between related tasks (e.g., morphology optimization in Ant (Brockman et al., 2016) and D'Kitty (Ahn et al., 2020)), overcoming the isolation assumption of conventional methods that process tasks independently. Second, it addresses the data scarcity challenge common in real-world applications (Nguyen et al., 2023). Traditional methods fail with limited historical data, while the universal offline BBO model can leverage cross-task patterns to guide optimization.

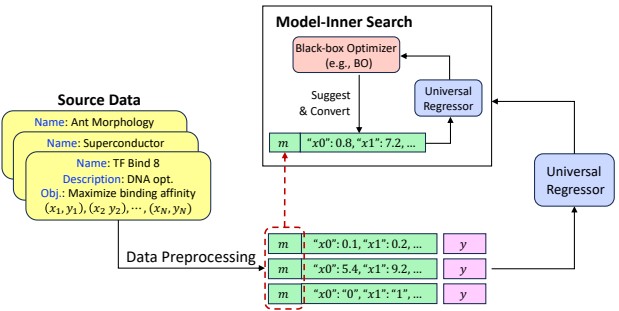

*Figure 1.* Framework of universal string-based offline BBO.

# 3. Method

In this section, we introduce potential methods to solve universal offline BBO. We begin by outlining the general framework of universal string-based offline BBO in Section 3.1, where we will present two variants based on modeling. Then, we discuss the issues of these variants in Section 3.2, which motivate us to improve the framework via *metadata guidance* and *smoothness enhancement* in Section 3.3 and Section 3.4, respectively. In Section 3.5, we discuss how to balance multiple losses and apply the improvement to the variants.

## 3.1. UniSO: Universal String-based Offline BBO

In this subsection, we present the general framework for universal offline BBO, which is illustrated in Fig. 1. The framework comprises four main components. Firstly, we convert the design-score offline data into *string representations* due to the heterogeneity of different search spaces from different tasks. Next, we discuss the importance and formulation for metadata, which facilitates the subsequent universal multi-task regressor training intended for downstream optimization. We propose two modeling variants for universal regressor instantiation based on different ways of handling objective scores, which are derived from recent advancements of string-based LLM for BBO (Song et al., 2024a; Nguyen et al., 2024; Tang et al., 2025). Finally, we elaborate on our string-based search strategy within the model. This pipeline aligns with the forward approach in traditional offline BBO, which typically encompasses training a scoring model followed by model-inner optimization, as discussed in Section 2.1.

### 3.1.1. STRING-BASED DATA REPRESENTATION

To solve multiple offline BBO problems with heterogeneous design spaces simultaneously, traditional methods that restrict the algorithm inside a fixed search scope are inapplicable, which calls for a more flexible representation for design-score pair data. Recently, Song et al. (2024a); Nguyen et al. (2024) have shown that string representation over $\mathbf{x}$ using

LLM is efficient and beneficial for BBO, enabling optimization in dynamic design spaces. Following (Nguyen et al., 2024), we represent each design $\mathbf{x}$ by a JSON dictionary-like format, e.g., a design $\mathbf{x} = (0, 1)^{\top}$ can be represented as $\{$`"x0":0,"x1":1`$\}$.

As for the score $y$, details of handling $y$ encompass token-based method and numeric-based method, which are determined by modeling methods of the regressor, which we will introduce in Section 3.1.3.

### 3.1.2. FORMULATION OF METADATA

While the mentioned string-based data representation exhibits flexibility, it alone is insufficient for effectively distinguishing between tasks and performing optimization in a multi-task suite. The integration of metadata into algorithms is quite essential for universal offline BBO, due to the following reasons: (1) from the NFL theorem (Wolpert & Macready, 1997), an algorithm tends to treat all possible tasks uniformly equal given only $(\mathbf{x}, y)$ data without any priors, which cannot perform well over all tasks. Thus, expert priors (Hvarfner et al., 2022; 2024) or user-defined task-specific metadata should be considered for optimization (Song et al., 2024b; Lindauer et al., 2024); (2) metadata enables training acceleration (Gao et al., 2025) and is crucial for task distinguishment when expert priors are limited (Chen et al., 2022b).

Thus, we extend each string $\mathbf{x}$ by inserting its associated metadata $m$ in the beginning. In order to maintain concision and preserve key information for optimization, we formulate the expressions of metadata $m$, consisting of three text-based task specifications: (1) **task name** to distinguish from other tasks; (2) **brief description** of the task with a concise natural language summary; and (3) detailed specification of the **optimization objective**. We provide all the metadata we used in our experiments in Appendix D. We tokenize the $(m, \mathbf{x})$ data using SentencePiece tokenizer (Kudo & Richardson, 2018) by default.

### 3.1.3. MULTI-TASK REGRESSOR TRAINING FOR UNIVERSAL OFFLINE BBO

The scoring model for offline BBO can be modeled by regression or ranking model. Here, we initiate the scoring model by a universal multi-task regressor for simplification. As shown in Fig. 2, we consider two variants of universal end-to-end regressors, UniSO-T and UniSO-N, which differ from how to deal with the objective score $y$.

**UniSO-T: Token-targeted regressor.** UniSO-T is a regressor with the objective of predicting the $y$ tokens. Specifically, it encodes the score $y$ into a list of tokens, trains a sequence-to-sequence auto-regressive model to predict the objective sequence, and then maps the sequence back to

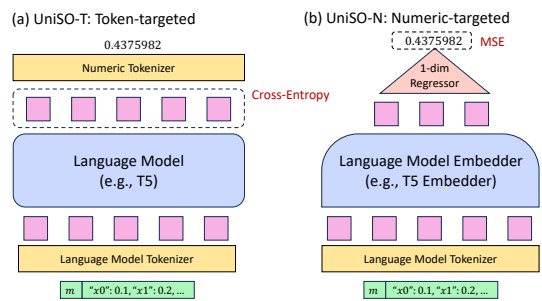

Figure 2. Model structure of UniSO-T (left) and UniSO-N (right).

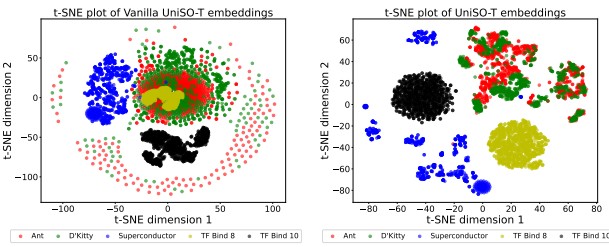

Figure 3. t-SNE plots comparing embedding distributions of vanilla UniSO-T and improved UniSO-T on five Design-Bench (Trabucco et al., 2022) tasks. The embedding distributions of vanilla UniSO-T show mixed and overlapping embeddings with a circular pattern, lacking clear task boundaries. Our improved method (right) achieves three improvements: (1) separating embeddings into distinct clusters for better task discrimination, (2) maintaining proximity between similar tasks (e.g., Ant and D'Kitty) to enable knowledge sharing between related domains, and (3) generating compact and smooth intra-task embedding distributions for stable representations.

### 3.1.4. STRING-BASED MODEL-INNER SEARCH

Upon completion of model training, the final designs are obtained through output maximization via model-inner search. Given the inherent challenges of computing exact gradients in discrete string space compared to continuous numerical space, we adopt black-box optimization strategies for this search process. Our implementation primarily employs Bayesian optimization (Garnett, 2023), which have demonstrated superior performance in gradient-free optimization scenarios. The technical specifications of our BO implementation, including acquisition function and code dependencies, can be found in Appendix B.2.

### 3.2. Issues of the Vanilla UniSO

Although the typical methods for UniSO-T and UniSO-N have demonstrated success in online BBO (Nguyen et al., 2024), there still exist limitations for offline BBO. Firstly, both the regressors in UniSO-T and UniSO-N cannot effectively distinguish different tasks without billions of data. To understand this, we visualize the t-SNE (Maaten & Hinton, 2008) plots of the embeddings of vanilla UniSO-T in the left sub-figure of Fig. 3, where the latent embeddings of vanilla UniSO-T show overlapping and circular pattern, lacking clear boundaries to distinguish tasks.

Besides, the embedding latent spaces for both UniSO-T and UniSO-N are not particularly designed for BBO, while properties of the latent spaces, like smoothness (Lee et al., 2023) or invariance (Qi et al., 2022), are usually important for downstream optimization.

Aiming at addressing these limitations, we introduce our proposed improvements in the following two subsections, which can be applied to both the modelings, and regulate

---

numerical values. A typical modeling of UniSO-T is Om-niPred (Song et al., 2024a) , which tokenizes $y$ by digits using P10 encoding [1] from (Charton, 2022) and trains a universal regressor based on next token prediction. Inspired by OmniPred, we tokenize $y$ using P10 encoding and implement a lightweight variant of the T5 model (Raffel et al., 2020). The model architecture incorporates Prefix-LM training (Liu et al., 2018) and is trained using cross-entropy loss on our custom dataset, as the training data of OmniPred follows a distinct organizational structure and remains unavailable.

**UniSO-N: Numeric-targeted regressor.** UniSO-N is another type of universal multi-task regressor, which first embeds the inputs into a unified latent space, and then trains a downstream regressor to map from embeddings to numerical objective scores. Inspired by (Nguyen et al., 2024), which uses a pre-trained T5 encoder (Raffel et al., 2020) and trains an in-context regressor for Bayesian optimization, we also employ a pre-trained T5-Small embedder [2], and train a MLP regressor mapping from embeddings to $y$ with mean squared error (MSE). We also normalize $y$ from different tasks using the same strategy in (Nguyen et al., 2024), which incorporates three steps: 1) utilize z-score normalization to shift the score of a single task; 2) implement a normal distribution fitting procedure on the subset of observations falling below the median value, utilizing a percentile-to-z-score transformation, to reduce sensitivity to bad outliers; 3) transform the scoring distribution through sequential application of min-max scaling $y \leftarrow \frac{y - y_{\min}}{y_{\max} - y_{\min}}$ and logarithmic transformation for re-scaling. Besides, although regressor training in offline BBO utilizes global z-score normalization, global statistics like mean value or standard variation of the embeddings are inaccessible, and thus batch normalization is applied to the embeddings before objective regression.

---

[1] For example, $y = 1.31 = 131 \times 10^{-2}$ is encoded into `<+><1><3><1><E-2>`

[2] https://huggingface.co/google-t5/t5-small

the embedding space based on two different perspectives, *metadata guidance* and *smoothness enhancement*.

### 3.3. Embedding Distribution Alignment via Metadata Guidance

As discussed above, the embeddings exhibit strange shape and cannot distinguish different tasks. Thus, we adopt a novel approach to align the distribution of input embeddings with their corresponding metadata embeddings via contrastive learning. The core idea is to guide the learning process by encouraging input embeddings to exhibit similarity patterns that mirror those observed in the metadata space.

Specifically, we first encode the input strings through the encoder to derive their embeddings (T5 encoder outputs for UniSO-T and embedder outputs for UniSO-N), while simultaneously employing a pre-trained expert language model embedder (T5-Small by default) to generate embeddings for metadata. After implementing mean pooling on both embeddings, we transform these intermediate representations through their respective nonlinear projection heads into a shared embedding space, yielding the final latent representations $\mathbf{z}^x \in \mathbb{R}^{n \times d}$ and $\mathbf{z}^m \in \mathbb{R}^{n \times d}$, where $n$ denotes the batch size and $d$ represents the dimension of the shared embedding space. Such a nonlinear projection is widely used in contrastive learning (Chen et al., 2020).

These projected representations are then utilized to compute the contrastive loss to align the distribution of input embeddings with their corresponding metadata embeddings:

$$\mathcal{L}\text{con} = -\frac{1}{N(N-1)} \sum_{1 \leq i < j \leq N} \hat{s}_{ij}^m \log\left(\frac{\exp(\frac{s_{ij}^x}{\tau})}{\sum_{k \neq i} \exp(\frac{s_{ik}^x}{\tau})}\right),$$

where: $s_{ij}^x = \frac{\mathbf{z}_i^{x\top} \mathbf{z}_j^x}{\|\mathbf{z}_i^x\| \cdot \|\mathbf{z}_j^x\|}$ represents the cosine similarity between input embeddings, and similarly, $s_{ij}^m$ is defined as the cosine similarity of the metadata embedding; $\hat{s}_{ij}^m = \frac{s_{ij}^m - \min_{i,j}(s_{ij}^m)}{\max_{i,j}(s_{ij}^m) - \min_{i,j}(s_{ij}^m)}$ is the normalized metadata similarity; $\tau$ is the temperature parameter; and $N$ is the data size.

The contrastive loss function enforces a dual objective by minimizing the KL divergence between the input embedding similarity distribution and the metadata-derived target distribution. An illustrative understanding of this loss function is that embeddings of the inputs with similar metadata remain proximate while dissimilar ones maintain distinct boundaries. As shown in the right sub-figure in Fig. 3, compared to the vanilla UniSO-T case, our proposed improvement is capable of distinguishing dissimilar tasks with the clearer clusters, while similar tasks like Ant and D'Kitty remain close, enabling structured information sharing for similar tasks.

However, this contrastive loss would raise non-smoothness [3] inside a task. The input embeddings from a same task (i.e., with the same metadata) can occupy arbitrary positions within a localized region of the high-dimensional space, forming either distinct clusters or non-uniform distributions, as long as they maintain adequate contrast (i.e., cosine similarity in the loss function) with respect to embeddings with dissimilar metadata. In Section 3.4, we further enhance the local smoothness of embeddings from a single task.

### 3.4. Local Embedding Smoothness Enhancement

Smoothness is crucial for neural network generalization (Nakkiran et al., 2019) and robustness (Weng et al., 2018), and smoothness of latent representation is also important for BBO (Zhang et al., 2020; Lee et al., 2023). For a given task $\mathcal{T}$, to enhance the local smoothness of embedding from $\mathcal{T}$, inspired by (Lee et al., 2023), we regulate the embedding space via the Lipschitz loss:

$$\mathcal{L}_{\text{lip}_{\mathcal{T}}} = \sum_{1 \leq i < j \leq N_{\mathcal{T}}} \max\left(0, \frac{|y_i - y_j|}{\|\mathbf{z}_i - \mathbf{z}_j\|_2} - L\right),$$

where $N_{\mathcal{T}}$ is the dataset size of task $\mathcal{T}$ and $L$ represents the local Lipschitz constant. We set $L$ as the median value of the Lipschitz matrix, i.e., $L = \text{median}_{1 \leq i < j \leq n_{\mathcal{T}}}(\frac{y_i - y_j}{\|\mathbf{z}_i - \mathbf{z}_j\|_2})$, by default following (Lee et al., 2023). The Lipschitz loss increases the correlation between the Euclidean distance of latent embeddings and the differences in their corresponding objective scores, enforcing the local embeddings to remain a local smoothness similar to the corresponding scores. Then, we compute the weighted sum of Lipschitz losses across all tasks, where the weights are designed to balance the impact of varying dataset sizes of different tasks:

$$\mathcal{L}_{\text{lip}} = \sum_{i=1}^{n_{\mathcal{T}}} \frac{\sum_{j=1}^{n_{\mathcal{T}}} N_{\mathcal{T}_j}}{N_{\mathcal{T}_i}} \mathcal{L}_{\text{lip}_{\mathcal{T}_i}},$$

where $n_{\mathcal{T}}$ denotes the number of tasks, and $\frac{\sum_{j=1}^{n_{\mathcal{T}}} N_{\mathcal{T}_j}}{N_{\mathcal{T}_i}}$ represents the weighting coefficient for task $\mathcal{T}_i$ to compensate for the disparity in dataset sizes across tasks. In our implementation, due to insufficient task-specific samples in individual batches of the shuffled training dataset, we utilize an unshuffled dataset that presents tasks sequentially for computing the Lipschitz loss.

### 3.5. Loss Balancing and Application to UniSO

With the main loss $\mathcal{L}_{\text{main}}$ to train the universal regressor (i.e., cross-entropy for UniSO-T or MSE for UniSO-N) and the mentioned regularization losses in hand, a naïve approach to handle these losses is to sum them up directly. However,

---

[3]In this work, smoothness refers to continuity in the embedding space.

simply summing them up would be effected by the scales of the losses, which focuses mainly on the loss with the largest scale during gradient backpropagation (Chen et al., 2018). Thus, a crucial problem is *how to balance the effects of these losses for training?*

To solve this, inspired by MetaBalance (He et al., 2022) which normalizes the gradients of different losses based on the gradient norm, we employ a similar yet simple technique to automatically balance the gradient of different losses based on the loss values. Specifically, for the main loss $\mathcal{L}_{\text{main}}$ and regularization losses $\mathcal{L}_{\text{con}}, \mathcal{L}_{\text{lip}}$, we calculate the gradient as:

$$\frac{\partial \mathcal{L}_{\text{total}}}{\partial \boldsymbol{\theta}} = \frac{\partial \mathcal{L}_{\text{main}}}{\partial \boldsymbol{\theta}} + \frac{\mathcal{L}_{\text{main}}}{\mathcal{L}_{\text{con}} + \delta} \cdot \frac{\partial \mathcal{L}_{\text{con}}}{\partial \boldsymbol{\theta}} + \frac{\mathcal{L}_{\text{main}}}{\mathcal{L}_{\text{lip}} + \delta} \cdot \frac{\partial \mathcal{L}_{\text{lip}}}{\partial \boldsymbol{\theta}}, \quad (1)$$

where $\boldsymbol{\theta}$ represents the parameters of the model and $\delta = 10^{-10}$ is a small constant added for numerical stability. This loss propagation mechanism adjusts the relative contributions of auxiliary losses by scaling them with respect to the main loss, maintaining task balance while preserving the principal optimization direction.

The balanced loss function is incorporated into both modeling frameworks as follows. For UniSO-T modeling, since an end-to-end universal regressor is trained from scratch, we update the parameters $\boldsymbol{\theta}$ with respect to Eq. (1) during training where we view the last hidden state of the encoder model as embeddings. For UniSO-N modeling, since the vanilla approach employs a pre-trained embedder without parameter updates during downstream numerical regressor training, we propose a two-stage approach: first fine-tuning the embedder via $\frac{\partial \mathcal{L}_{\text{total}}}{\partial \boldsymbol{\theta}} = \frac{\partial \mathcal{L}_{\text{con}}}{\partial \boldsymbol{\theta}} + \frac{\mathcal{L}_{\text{con}}}{\mathcal{L}_{\text{lip}} + \delta} \cdot \frac{\partial \mathcal{L}_{\text{lip}}}{\partial \boldsymbol{\theta}}$ similar to Eq. (1), and then training the regressor while keeping the embedder parameters frozen.

## 4. Experiment

In this section, we empirically study our proposed framework of universal string-based offline BBO on various tasks. We first introduce the experimental settings and the tasks in Section 4.1, and then show the performance of UniSO and answer several important research questions (RQs) in Section 4.2.

### 4.1. Experimental Settings and Tasks

**Detailed settings.** Following recent works in string-based LLM for BBO (Song et al., 2024a; Nguyen et al., 2024), we use a lightweight version of encoder-decoder T5 models (Raffel et al., 2020) for both the UniSO-T and UniSO-N variants. For UniSO-N series, we instantiate the downstream regressor as a MLP with two hidden layers of 2048 units

and train the model using AdamW optimizer (Loshchilov & Hutter, 2019) for 200 epochs with a batch size of 128. After the UniSO regressor is trained, we use BO as the default optimizer to search inside the model for 200 iterations, maintaining a population size that corresponds to the desired number of final solutions, which is set as 128 following prior works in offline BBO (Trabucco et al., 2022; Qian et al., 2025).

**Tasks.** We consider unconstrained tasks from two benchmarks for offline BBO, the popular Design-Bench (Trabucco et al., 2022) and the recently proposed benchmark SOO-Bench (Qian et al., 2025). Specifically, we select Ant Morphology (Brockman et al., 2016), D'Kitty Morphology (Ahn et al., 2020), Superconductor (Hamidieh, 2018), TF Bind 8 and TF Bind 10 (Barrera et al., 2016) from Design-Bench [4], and GTOPX 2,3,4,6 (Schlueter et al., 2021) from SOO-Bench. These tasks cover the `CONTINUOUS` and `CATEGORICAL` search spaces, and vary from low to high dimensional cases. Detailed description and properties of these tasks can be found in Appendix C.

### 4.2. Experimental Results

In this section, we present our experimental findings, aiming to answer the following RQs.

**RQ1: Is the string-based representation for designs versatile to offline BBO in the single task cases?** Previous works have demonstrated the potential of utilizing string-based representation for universal regression (Song et al., 2024a; Tang et al., 2025) and traditional BBO algorithms (Nguyen et al., 2024). However, its versatility to offline BBO is still unexplored. Thus, for a sanity check, we compare the vanilla variants of UniSO-T and UniSO-N to the expert model trained with numerical inputs within a single offline BBO task in this RQ. For a fair comparison, the experts are instantiated as the same structure of the objective regressor in UniSO-N, and we normalize the input designs using batch normalization, followed by model-inner search by both EAs and gradient ascent. Training details of the expert models can be found in Appendix B.1.

As shown in Table 1, we find that UniSO methods which utilize string representation inputs: (1) are capable of solving offline BBO, with most of the final scores exceeding the best score in offline dataset $\mathcal{D}(\text{best})$, except for Ant and D'Kitty in UniSO-N; (2) show competitive results against the numeric-input experts, where UniSO-T achieves the an average rank of 2.000 among the four methods, performing the best on 3 of the 10 tasks and being the runner-up on 4 of 10 tasks, while the best expert, BN + BO (i.e., batch nor-

---

[4]Following recent works in offline BBO (Tan et al., 2025; Yun et al., 2024), we exclude three tasks from Design-Bench, and provide detailed explanation in Appendix C.1.

*Table 1.* Un-normalized scores in unconstrained tasks from Design-Bench and SOO-Bench, where the best and runner-up results on each task are **Blue** and **Violet**, respectively. $\mathcal{D}$(best) denotes the best score in the offline dataset and BN represents batch normalization for numerical input designs. Both numeric-input experts and string-input UniSO methods are trained **within a single task**.

| Task | $\mathcal{D}$(best) | Numeric-input Experts | | String-input UniSO | |
|---|---|---|---|---|---|
| | | BN + BO | BN + Grad | UniSO-T | UniSO-N |
| Ant | 165.326 | **241.350 ± 288.922** | 229.462 ± 165.869 | **385.945 ± 95.402** | 103.669 ± 263.572 |
| D'Kitty | 199.363 | 102.972 ± 130.192 | **183.263 ± 62.436** | **243.428 ± 20.137** | -4.674 ± 9.298 |
| Superconductor | 74.000 | **83.884 ± 4.099** | **97.137 ± 6.113** | 79.783 ± 0.507 | 80.129 ± 3.320 |
| TF Bind 8 | 0.439 | 0.898 ± 0.088 | **0.959 ± 0.023** | **0.919 ± 0.033** | 0.502 ± 0.064 |
| TF Bind 10 | 0.005 | 0.454 ± 0.091 | **0.888 ± 0.229** | **0.776 ± 0.101** | 0.390 ± 0.227 |
| GTOPX 2 | -195.586 | -74.763 ± 8.577 | -128.310 ± 15.616 | **-73.484 ± 8.081** | **-72.981 ± 20.446** |
| GTOPX 3 | -151.190 | **-40.104 ± 2.748** | -151.190 ± 0.000 | **-41.952 ± 8.451** | -58.933 ± 14.143 |
| GTOPX 4 | -215.716 | **-87.976 ± 3.497** | -215.710 ± 0.000 | **-75.022 ± 12.215** | -113.715 ± 15.528 |
| GTOPX 6 | -112.599 | **-49.660 ± 6.170** | -112.599 ± 0.000 | -50.391 ± 4.048 | **-48.771 ± 3.725** |
| Avg. Rank | / | **2.333 ± 0.667** | 2.667 ± 1.333 | **2.000 ± 0.943** | 3.000 ± 1.155 |

malization for numerical input designs and using BO as the model-inner optimization algorithm), only ranks 2.333 on average. However, UniSO-N demonstrates inferior results compared to numeric-input experts. This may be because learning from LM embeddings to numerical objective space is relatively challenging. We also provide results of UniSO using evolutionary algorithms as the model-inner optimizer in Table 18 in Appendix E.6. Under the batch normalization setting, the experimental results demonstrate that employing evolutionary algorithms for search within expert models with numerical input outperforms gradient ascent.

**RQ2: How do the universal string-based offline BBO methods perform across a wide range of offline BBO tasks?** In Table 2, we report the results of our main experiments, where UniSO-T and UniSO-N are compared to $\mathcal{D}$(best) and single-task experts (numeric-input MLP), which are trained after batch normalization. The term "Improved" represents our improvements to regularize the latent space as introduced in Section 3. Among all the considered tasks, we find that: (1) most offline BBO tasks could benefit from UniSO methods, since the results of both the vanilla and improved versions of UniSO-T and UniSO-N exceed $\mathcal{D}$(best), except for three cases; (2) UniSO methods are competitive to single-task experts, and the improved UniSO-T achieves better empirical results than the experts; (3) the improved components (i.e., embedding distribution alignment and local smoothness enhancement) contribute a lot to both UniSO-T and UniSO-N, with generally better results and average ranks. Besides, we show the t-SNE results of improved UniSO-T in the right sub-figure of Fig. 3. Compared to the vanilla one, our improved method exhibits better distribution for task distinguishment while similar tasks remain close for information sharing, and shows smoother local cluster embeddings in the latent space.

**RQ3: Can UniSO generalize to unseen tasks?** As string-based universal online BBO methods show impressive generalization ability (Nguyen et al., 2024), we also examine the

zero-shot and few-shot generalization ability (Nguyen et al., 2023) of improved UniSO-T and UniSO-N on tasks that are unseen in the training data. Specifically, we choose three tasks RobotPush, Rover (Wang et al., 2018), and LunarLander (Brockman et al., 2016). Details of these tasks can be found in Appendix C. For zero-shot setting, we directly concatenate the metadata and the string designs suggested by model-inner optimizer and maximize the model's output. For few-shot setting, we first use the few-shot data to fine-tune the universal regressor using the main loss (i.e., cross-entropy for UniSO-T and MSE for UniSO-N) and SGD optimizer with a learning rate of $2 \times 10^{-5}$ for 5 epochs, and then search for final designs. We use the data provided by (Wang et al., 2024a) and the poorest 100 pairs of data to construct the few-shot dataset.

As shown in Fig. 4, both improved UniSO-T and improved UniSO-N show good generalization ability. The zero-shot results significantly exceed $\mathcal{D}$(best) and the few-shot results are even better than that of the single-task expert based on z-score normalization, which is a stronger expert than that based on batch normalization.

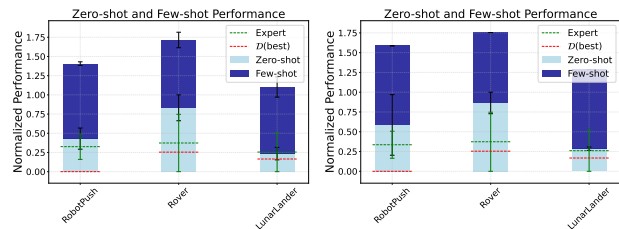

*Figure 4.* Normalized performance comparison between zero-shot inference and few-shot finetuning of UniSO-T (left) and UniSO-N (right) across three tasks (RobotPush, Rover, and LunarLander). The light blue bars represent zero-shot performance, the dark blue bars show the improved performance after few-shot finetuning. The red dashed line denotes the best score $\mathcal{D}$(best) in the few-shot offline dataset.

*Table 2.* Un-normalized scores in unconstrained tasks from Design-Bench and SOO-Bench, where the best and runner-up results on each task are **Blue** and **Violet**, respectively. $\mathcal{D}$(best) denotes the best score in the offline dataset and BN represents batch normalization for numerical input designs. Single-task experts are trained within one task, while UniSO-T and UniSO-N are done in a multi-task manner.

| Task | $\mathcal{D}$(best) | Single-task Numeric-input Experts | | UniSO-T | | UniSO-N | |
|---|---|---|---|---|---|---|---|
| | | BN + BO | BN + Grad | Vanilla | Improved | Vanilla | Improved |
| Ant | 165.326 | 241.350 ± 288.922 | 229.462 ± 165.869 | **446.283 ± 20.635** | **455.658 ± 39.188** | 260.058 ± 280.526 | 381.787 ± 88.907 |
| D'Kitty | 199.363 | 102.972 ± 130.192 | **183.263 ± 62.436** | 168.263 ± 104.068 | **222.007 ± 33.677** | 4.872 ± 13.718 | 42.215 ± 71.095 |
| Superconductor | 74.000 | **83.884 ± 4.099** | **97.137 ± 6.113** | 81.047 ± 6.476 | 82.642 ± 3.467 | 78.791 ± 2.981 | 82.033 ± 3.412 |
| TF Bind 8 | 0.439 | **0.898 ± 0.088** | **0.959 ± 0.023** | 0.870 ± 0.085 | 0.857 ± 0.069 | 0.739 ± 0.101 | 0.856 ± 0.043 |
| TF Bind 10 | 0.005 | 0.454 ± 0.091 | 0.888 ± 0.229 | **0.929 ± 0.802** | **0.944 ± 0.794** | 0.471 ± 0.150 | 0.528 ± 0.156 |
| GTOPX 2 | -195.586 | **-74.763 ± 8.577** | -128.310 ± 15.616 | **-68.526 ± 14.058** | -90.106 ± 8.223 | -94.656 ± 25.608 | -88.063 ± 29.409 |
| GTOPX 3 | -151.190 | **-40.104 ± 2.748** | -151.190 ± 0.000 | -47.440 ± 6.249 | **-44.427 ± 13.456** | -63.007 ± 12.559 | -48.848 ± 14.797 |
| GTOPX 4 | -215.716 | -87.976 ± 3.497 | -215.710 ± 0.000 | **-74.070 ± 13.131** | **-74.779 ± 11.032** | -111.437 ± 28.417 | -94.579 ± 11.591 |
| GTOPX 6 | -112.599 | -49.660 ± 6.170 | -112.599 ± 0.000 | -50.314 ± 6.481 | **-48.244 ± 5.766** | **-46.056 ± 11.547** | -52.672 ± 9.042 |
| Avg. Rank | / | 3.111 ± 1.523 | 4.111 ± 2.183 | **2.667 ± 1.247** | **2.222 ± 1.133** | 4.778 ± 1.474 | 4.111 ± 0.737 |

**RQ4: Is it necessary to train from pre-trained LM for UniSO-N?** As pre-trained language models are widely used in recent string-based optimization or regression (Nguyen et al., 2024; Tang et al., 2025), we investigate whether pre-trained initialization is necessary for UniSO-N by comparing UniSO-N trained from scratch with that from a pre-trained checkpoint. The results in the left subfigure of Fig. 5 show that UniSO-N trained from scratch consistently outperforms the pre-trained model. The OOD rank correlation denotes the rank correlation between between predicted and ground-truth objective scores in OOD regions, which is a crucial metric to evaluate the effectiveness of the surrogate model for offline BBO (Tan et al., 2025). Higher rank correlation indicates better surrogate model for offline BBO. The comparison in the right subfigure of Fig. 5 shows that UniSO-N trained from scratch consistently performs better than that from a pre-trained checkpoint. The pre-trained models initially demonstrate poor performance, and there is only little improvement later on. This may be due to pre-trained LMs' harmful biases for numerical optimization. We will elaborate on this in the next RQ.

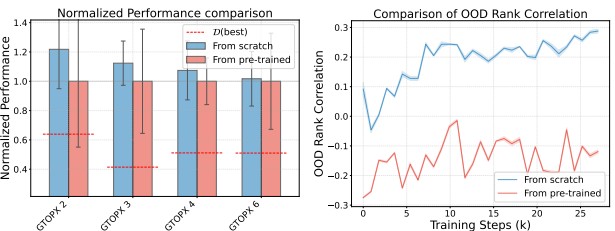

*Figure 5.* Left: Normalized performance comparison between UniSO-N trained from scratch (blue) and UniSO-N with pre-trained embedder (yellow) across four tasks in SOO-Bench. The red dashed line denotes the best score $\mathcal{D}$(best) in the offline dataset. Right: Comparison of OOD Spearman rank correlation on the Superconductor task between UniSO-N from pre-trained embedder (blue) and from-scratch (red). The OOD dataset is constructed following (Chen et al., 2023a). Higher correlation indicates better surrogate model for offline BBO (Tan et al., 2025).

**RQ5: Do LM priors indeed do harm to numeric optimization?** To better understand LMs' harmful priors for numerical optimization, we visualize the attention weight distribution (Allen-Zhu & Li, 2023) across input tokens. On the top two subfigures in Fig. 6, we visualize pre-trained T5 embedder's attention weight distribution on the TF Bind 10 task (detailed results of more tasks can be found in Appendix F). We can observe that pre-trained T5 embedder exhibits strong bias towards language grammar structural tokens (especially EOS tokens), while numerical tokens that are crucial for optimization are assigned with limited attention. In contrast, embedder trained from scratch shows a more balanced attention distribution with stronger focus on numerical tokens. This difference in attention assignment provides strong evidence for LMs' harmful biases for numeric optimization. To better understand how different LMs perform, in the next RQ, we will discuss the attention mechanisms of different LMs in detail.

**RQ6: How do different LMs assign attentions?** We further investigate whether the attention mechanism in T5 still exists in other pre-trained LMs. Specifically, we implement Qwen2.5-1.5B (Qwen Team et al., 2024) [5] and DeepSeek-R1-Distill-Qwen-1.5B (DeepSeek-AI et al., 2025) [6] on TF Bind 10 and GTOPX 6 tasks, which are shown in the bottom two subfigures of Fig. 6 and Appendix F, respectively.

The results reveal two key findings. Firstly, different from the T5 cases, Qwen2.5-1.5B and DeepSeek-R1-Distill-Qwen-1.5B demonstrate more balanced attention patterns on different input components. Specifically, these two models show higher attention weights on both the metadata and numerical tokens, capturing more useful information rather than mainly focusing on grammar tokens. Besides, compared to the Qwen base model, DeepSeek-R1 assigns more attention on numeric-related tokens, showing greater capa-

---

[5] https://huggingface.co/Qwen/Qwen2.5-1.5B
[6] https://huggingface.co/deepseek-ai/
DeepSeek-R1-Distill-Qwen-1.5B

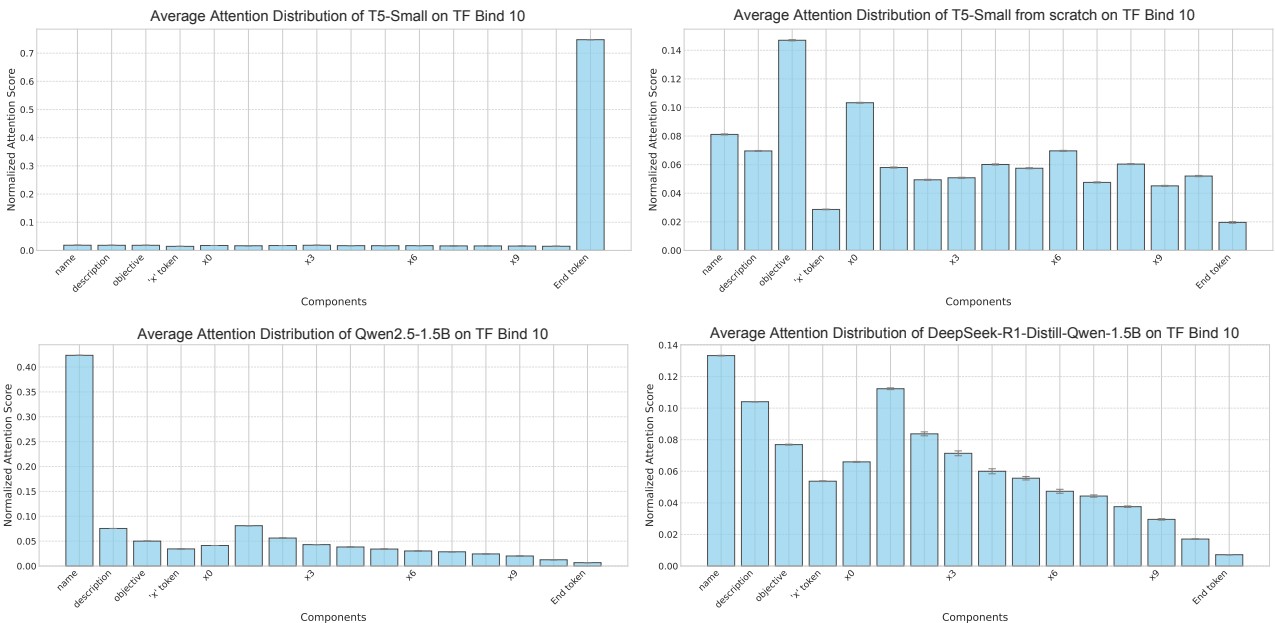

*Figure 6.* Attention weight distribution comparison of different models on the TF Bind 10 (Barrera et al., 2016) task: pre-trained T5-Small embedder (top left), T5-Small embedder trained from scratch (top right), pre-trained Qwen2.5-1.5B (bottom left), and pre-trained DeepSeek-R1-Distill-Qwen-1.5B (bottom right). More visualization results on other tasks are provided in Appendix F.

bility on optimization. This may come from the training data of DeepSeek-R1, which contains mathematical content aiming to enhance model's reasoning ability. The result suggests that LMs with stronger mathematical capabilities may be better for numerical optimization.

**RQ7: Is the best-performing UniSO variant comparable to the well-studied expert single-task offline BBO methods?** In Table 1 and Table 2, we use batch normalization for numerical inputs on single-task expert methods for a fair comparison. However, a more widely-used practice in the field of offline BBO is to employ a global z-score normalization for inputs. Thus, to better understand the performance of UniSO methods, in Table 8 in Appendix E.1, we compare improved UniSO-T, the best-performing universal offline BBO approach in Table 2, to a wide range of single-task expert offline BBO methods on Design-Bench. These methods are mainly based on z-score normalization for input solutions. Here the improved UniSO-T method is trained solely using Design-Bench tasks, and the detailed results of compared methods are referred from (Tan et al., 2025). According to the results in Table 8, UniSO-T achieves an average rank of 9.8 across 21 expert single-task offline BBO methods, outperforming some recently proposed methods, which are specifically designed for single-task offline BBO, and even achieving the best results on TF-Bind-10. Overall, there is still improvement room for UniSO methods, compared to state-of-the-art single-task offline BBO methods. Thus, how to propose better techniques and improve per-

formance for universal offline BBO is a crucially important future work.

**Additional analysis and ablation studies.** We provide additional results in Appendix E. Specifically, we provide the computational cost of UniSO in Appendix E.2, and explore the integration of in-context regressors, e.g., UniSO-N with Transformer neural processes (TNPs; Nguyen & Grover, 2022; Nguyen et al., 2023; 2024) in Appendix E.3. Furthermore, we discuss the influence of different model sizes in Appendix E.4. To further understand the contribution of each component of UniSO, in Appendix E.5, we conduct ablation studies on metadata quality, loss components, loss balancing strategies and model-inner search algorithms. Note that recently LLM-based automatic heuristics design shows impressive performance in many fields (Romera-Paredes et al., 2024; Novikov et al., 2025; Zheng et al., 2025). In Appendix E.7, we investigate the usage of LLM-based heuristics (Liu et al., 2024a;b;c) to further improve performance via metadata and appending auxiliary loss.

## 5. Conclusion

In this paper, we propose UniSO to overcome barriers in universal offline BBO by unifying string-based representation, latent space regularization, and metadata-guided learning. Extensive experimental results show the universality and effectiveness of UniSO. Future works can include training on more data and exploring in-context learning for enhanced performance in diverse real-world scenarios.

## Impact Statement

This paper presents work whose goal is to advance the field of large language models and black-box optimization. There are many potential societal consequences of our work, none which we feel must be specifically highlighted here.

## Acknowledgement

The authors thank anonymous reviewers for their insightful and valuable comments: ICML Reviewer 4Ajw for suggesting the usage of BO as model-inner optimizer, mechanism interpretable analysis and LLM-based heuristics, Reviewer GL3R for suggesting analysis of pre-trained checkpoint and model size, Reviewer TeSq for integration of in-context regressor, Reviewer F6Mo for comparison analysis between UniSO-T and UniSO-N, and ICLR FM-Wild workshop reviewers for constructive discussion on future works. This work was supported by the National Science and Technology Major Project (2022ZD0116600), the National Science Foundation of China (62276124, 624B1025, 624B2069), the Fundamental Research Funds for the Central Universities (14380020), and Young Elite Scientists Sponsorship Program by CAST for PhD Students. The authors want to acknowledge support from the Huawei Technology cooperation Project.

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

# A. Related Work

## A.1. Online BBO

An important setup for BBO tasks is online BBO, which obtains optimal solutions via iteratively querying the unknown black-box objective. Traditional online BBO algorithms are population-based search algorithms, e.g., evolutionary algorithm (EA; Bäck, 1996; Zhou et al., 2019), evolution strategies (ES; Hansen et al., 2015; Hansen, 2016), and particle swarm optimization (PSO; Kennedy & Eberhart, 1995; Gong et al., 2015). However, they find solutions via population-level evaluations and updates, which is unsuitable for expensive real-world BBO tasks. Bayesian optimization (BO; Garnett, 2023) is is a widely used sample-efficient method for expensive BBO problems. BO first fits a surrogate model, typically Gaussian processes (GP; Rasmussen & Williams, 2006), to approximate the objective function at each iteration, and then optimize a pre-defined acquisition function to sample the next candidate (Shahriari et al., 2016; Frazier, 2018). However, the limited efficiency of GP hinders the scalability to a large-scale scenario. Thus, recently deep learning techniques are well adopted to BO for surrogate modeling (Garnett et al., 2014; Wistuba & Grabocka, 2021; Wang et al., 2024b; Garnelo et al., 2018; Nguyen & Grover, 2022) or acquisition instantiation (Maraval et al., 2023). However, in many scientific and industrial scenarios, online evaluation is prohibited or cannot be used anymore (Gaulton et al., 2012; Kumar et al., 2022), showing an urgent need for offline BBO, which we will introduce in the next subsection.

## A.2. Offline BBO

Offline BBO (Trabucco et al., 2022; Kim et al., 2025) methods can be generally categorized into the following two types:

**Forward approach.** This category usually trains a scoring surrogate and maximizes the output to obtain the final design. However, these methods suffer from OOD issue such that errors made by the surrogate in OOD region would mislead the search procedure. To mitigate it, NEMO (Fu & Levine, 2021), COMs (Trabucco et al., 2021), IOM (Qi et al., 2022), RoMA (Yu et al., 2021), BOSS (Dao et al., 2024b) and IGNITE (Dao et al., 2024a) regulate the model from different perspectives, while Tri-Mentoring (Chen et al., 2023a) and ICT (Yuan et al., 2023) ensemble surrogates to enhance robustness. Besides, BDI (Chen et al., 2022a), MATCH-OPT (Hoang et al., 2024), and Cliqueformer (Kuba et al., 2024b;a) incorporate more information distilled from offline dataset, and ARCOO (Lu et al., 2023), PGS (Chemingui et al., 2024), GABO (Yao et al., 2024), and DEMO (Yuan et al., 2025) employ different techniques to guide the model-inner search. Recently, Tan et al. (2025) point out that regression is not suitable for offline BBO and call for a paradigm shift to rank the designs.

**Backward approach** typically fits a probabilistic model $p(\mathbf{x}|y)$, and samples promising designs from the model, which can be instantiated by prevalent generative models. For example, MINs (Kumar & Levine, 2020) uses GAN (Goodfellow et al., 2014), while DDOM (Krishnamoorthy et al., 2023), RGD (Chen et al., 2024), and DiffOPT (Kong et al., 2024) utilize diffusion model (Ho et al., 2020). Recent works also focus on learning from trajectories (Mashkaria et al., 2023; Yun et al., 2024), synthetic priors (Nguyen et al., 2023), or sampling-free latent variables (Yu et al., 2024).

## A.3. LLM for BBO

Recent progress in LLM has demonstrated the potential applicability of LLM to solve BBO problems (Song et al., 2024b). Works in this field can be generally categorized into two major branches: (1) leveraging general intelligence of LLM to design algorithms or heuristics **via natural language** (Romera-Paredes et al., 2024; Liu et al., 2024c; Novikov et al., 2025), and (2) utilizing the representation ability of LLM to improve learned BBO algorithms **via tokens or embeddings**.

For the first branch, works leverage LLM to design algorithm code. FunSearch (Romera-Paredes et al., 2024) proposes an iterative generation procedure for better code, which has been proven successful in many domains, e.g., symbolic regression (Shojaee et al., 2025) and scientific domains (Novikov et al., 2025). EoH (Liu et al., 2024a; Yao et al., 2025) incorporates crossover and mutation operators from evolutionary algorithms for better algorithm design, which also shows great potential in BBO (Liu et al., 2024c;b; Zheng et al., 2025). In Appendix E.7, we also discuss the potential of such methods to further improve the performance of UniSO. Besides, LLaMoCo (Ma et al., 2024) and LLMOPT (Jiang et al., 2025) train an optimization code generator via fine-tuning pre-trained LLM and training from scratch, respectively. Many of other works also employ LLM as components of BBO optimizers, e.g., LLAMBO (Liu et al., 2024d) simulates components of Bayesian optimization, such as regressor and acquisition function, with LLM. The success of this branch lies in whitening the problem nature via intelligence and prior knowledge conveyed by natural language. However, it is still limited by the training corpus (when it meets up with an unseen domain) and lacks interpretability.

For the second branch, various approaches have been proposed to leverage the representation power of LLM structure. The well-studied provable properties of these structure for sequential modeling enable its application for BBO, where optimization trajectory can also be viewed as a sequence. Given that **attention-based** Transformers (Vaswani et al., 2017) has superior scalability (Brown et al., 2020) and can learn in-context well (Garg et al., 2022), TNPs (Nguyen & Grover, 2022; Maraval et al., 2023) employs raw Transformers as in-context regressor for BBO. Note that Transformers trained with synthetic priors performs well on tabluar data (Hollmann et al., 2025) and Bayesian inference (Müller et al., 2022), works like PFNs4BO (Müller et al., 2023), LICO (Nguyen & Grover, 2025) and ExPT (Nguyen et al., 2023) utilize such priors for in-context optimization on both online and offline scenarios. Attention-based methods exhibit superior performance in BBO field, but they usually optimize in a fixed search space. **Token-based** methods offer another perspective by representing designs or parameters as tokens and learning from historical trajectories. For example, OptFormer (Chen et al., 2022b) views a parameter value as a token for hyper-parameter optimization and employs Transformer structure, combined with metadata, to represent an optimization trajectory, while Dery et al. (2022) consider a multi-step improvement for OptFormer and Song et al. (2025) plug a regret-to-go token into the algorithm history to solve BBO problems. Recently, **string-based** approaches have emerged as a promising direction. OmniPred (Song et al., 2024a) regresses in multi-task suite based on string-based representation for arbitrary designs, metadata and scores, while Nguyen et al. (2024); Tang et al. (2025) use LLM embeddings to pre-train an in-context regressor for BBO. Motivated by the superior performance of these methods, in this work, we use string-based representation to solve universal offline BBO, as discussed in Section 3.

## B. Experimental Settings

We adopted the T5X architecture (Raffel et al., 2020) as our base model. For embedder in UniSO-N and metadata embedder in UniSO-T, we directly leverage the pre-trained T5-small model. Details according to other components of the model and search algorithms can be found in the following subsections.

### B.1. Model Architecture Hyperparameters and Optimizer Configurations

For UniSO-T, we use the T5-based architecture. Pre-training details are listed in Table 3. We adopt T5's default tokenizer, i.e., SentencePiece (Kudo & Richardson, 2018), as the input tokenizer and P10 tokenizer (Charton, 2022) as the output tokenizer. The model is trained for 200 epochs with a batch size of 128. For few-shot fine-tuning, we fine-tune the model for 5 epochs using SGD. During inference, we set temperature to 0.7, apply top-$k$ sampling with $k = 20$ and nucleus sampling with $p = 0.95$ to balance output quality and diversity.

For UniSO-N, we use pre-trained T5-Small embedder to embed input strings, and then apply a 1-dimensional output MLP regressor for model instantiation. Model configurations are shown in Table 4. For pretraining, the regressor is trained for 200 epochs with a batch size of 128 before evaluation. For few-shot fine-tuning, we fine-tune the regressor for 5 epochs using SGD.

*Table 3.* Hyper-parameters and optimizer configuration for UniSO-T.

| Module | Hyper-parameters | Values |
|---|---|---|
| Encoder | Vocab size | 32128 |
| | Num layers | 6 |
| | Head dimension | 32 |
| | Embedding dimension | 384 |
| | MLP dimension | 512 |
| Decoder | Num layers | 6 |
| | Head dimension | 32 |
| | Embedding dimension | 384 |
| | MLP dimension | 512 |
| Optimizer | Name | AdamW |
| | Learning rate | $1 \times 10^{-4}$ |
| | Weight decay | 0.01 |
| | $(\beta_1, \beta_2)$ | $(0.9, 0.99)$ |
| LR scheduler | Name | CosineAnnealing |
| | Warmup steps | 1000 |

*Table 4.* Hyper-parameters and optimizer configuration for UniSO-N.

| Module | Hyper-parameters | Values |
|---|---|---|
| Regressor | Hidden layers | 2 |
| | Hidden dimension | 2048 |
| | Output dimension | 1 |
| | Activation | ReLU |
| Optimizer | Name | AdamW |
| | Learning rate | $1 \times 10^{-4}$ |
| | Weight decay | $1 \times 10^{-5}$ |
| | $(\beta_1, \beta_2)$ | $(0.9, 0.99)$ |
| LR Scheduler | Name | CosineAnnealing |
| | Warmup Steps | 1000 |

## B.2. Model-Inner Search Algorithms Details

For model-inner search algorithms, we consider three types of representative black-box optimizers, Bayesian optimization (BO; Garnett, 2023), evolutionary algorithms (EAs; Bäck, 1996; Zhou et al., 2019), and evolutionary strategies (ES; Hansen et al., 2015). Experimental comparison of these model-inner optimizers can be found in Appendix E.3. Here we present detailed hyper-parameter settings and other configurations of these optimizers.

For BO, we adopt BO-$q$EI for `CONTINUOUS` problems, which is implemented with BoTorch (Balandat et al., 2020)[7]. The hyper-parameters of BO-$q$EI implementation are shown in Table 5. For `CATEGORICAL` problems, we use a transformed overlap kernel from (Khan et al., 2023) as the Gaussian process (GP) kernel function, then maximize the upper confidence bound (UCB) acquisition of GP outputs using EAs to sample candidates in each iteration.

*Table 5.* Hyper-parameters of BO model-inner optimizer.

| Search space | Hyper-parameters | Value |
|---|---|---|
| | GP noise variance | 0.01 |
| | Initial training samples | 500 |
| | Batch size | 10 |
| | Number of restarts | 10 |
| `CONTINUOUS` | Initial random candidates | 128 |
| | Batch limit | 5 |
| | Max optimization iterations | 200 |
| | BO iterations | 100 |
| | MC samples | 128 |
| | GP noise variance | 0.01 |
| `CATEGORICAL` | EA population size | 32 |
| | EA generations | 200 |
| | UCB $\beta$ | 0.2 |

For EAs, we use `pymoo` (Blank & Deb, 2020)[8] for implementation. We initialize the population as the top-$k$ scoring designs in the dataset, where $k = 10$ is the population size. For continuous tasks, we use simulated binary crossover (SBX) and polynomial mutation (PM), which are the default genetic operators in `pymoo`, and we use the default hyper-parameters. For categorical tasks, we use uniform crossover and random replacement mutation, which are the default operators for `CATEGORICAL` search spaces in (Xue et al., 2024). We iterate 100 generations to search for the final solutions.

---

[7]https://botorch.org/docs/tutorials/closed_loop_botorch_only/
[8]https://pymoo.org/

For ES, we use CMA-ES (Hansen, 2016) implemented with `evosax` (Lange, 2023)[9]. We set the initial $\alpha = 0.5$, and search for 100 iterations.

## C. Details of Tasks and Datasets

In this section, we introduce details of different tasks and datasets we use in our experiments. We use the datasets and tasks from Design-Bench (Trabucco et al., 2022) and SOO-Bench (Qian et al., 2025), with 9 tasks and a dataset size of 90K in total. Detailed properties can be found in Table 6.

Table 6. Properties of tasks and datasets in Design-Bench and SOO-Bench.

| Benchmark Suite | Task | Dataset size | Variable type | # dimensions |
|---|---|---|---|---|
| Design-Bench | Ant | 10004 | CONTINUOUS | 56 |
| | D'Kitty | 10004 | CONTINUOUS | 60 |
| | Superconductor | 17010 | CONTINUOUS | 86 |
| | TF Bind 8 | 32898 | CATEGORICAL | 8 (3 categories) |
| | TF Bind 10 | 10000 | CATEGORICAL | 10 (3 categories) |
| SOO-Bench | GTOPX 2 | 22000 | CONTINUOUS | 22 |
| | GTOPX 3 | 18000 | CONTINUOUS | 18 |
| | GTOPX 4 | 26000 | CONTINUOUS | 26 |
| | GTOPX 6 | 22000 | CONTINUOUS | 22 |

### C.1. Design-Bench Tasks

Design-Bench (Trabucco et al., 2022)[10] is a famous benchmark suite for offline BBO. It includes various realistic tasks from real-world optimization problems, and each task corresponds to an oracle function for evaluation and a large static offline dataset. In this paper, we mainly consider 6 tasks in Design-Bench, and we directly use the open-sourced dataset of Design-Bench[11] as a part of the training data. Details of these tasks are as follows.

**Ant and D'Kitty Morphology**. These two tasks are both robot morphology optimization problems, and the goal is to optimize the morphological structure of two simulated robots: Ant from OpenAI Gym (Brockman et al., 2016) and D'Kitty from ROBEL (Ahn et al., 2020). Specifically, the objective of Ant Morphology is to make the Ant robot run quickly and that of D'Kitty Morphology is to let the D'Kitty robot move exactly to the predefined location. Both these tasks use a task-specific pre-trained Soft Actor Critic (Haarnoja et al., 2018) policy as controller, and simulate in the MuJoCo (Todorov et al., 2012) simulator with a timestep of 100. The design dimension of Ant Morphology is 56 and that of D'Kitty Morphology is 60, both of which include size, orientation, and location of the limbs. The dataset sizes of these two tasks are both 10004 with CONTINUOUS design spaces.

**Superconductor**. The objective of the Superconductor task (Hamidieh, 2018) is to maximize the critical temperature which derived from the chemical formula for a superconducting material. Although the actual critical temperature of a decoded material is inaccessible, which needs physical experiments, the evaluation is conducted using a pre-trained random forest regressor from (Fannjiang & Listgarten, 2020) as oracle function, which achieves a Spearman rank-correlation coefficient of 0.9210 on a held-out validation set. The design space consists of 86 CONTINUOUS variables, and the dataset size is 17010.

**TF Bind 8 and TF Bind 10**. The objective of TF Bind 8 and TF Bind 10 (Barrera et al., 2016) is to maximize the binding affinity score of the designed length 8 and length 10 DNA sequence with a prevalent human transcription factor SIX6_REF_R1. Scores are obtained via direct lookup from the exhaustive evaluation database established by (Barrera et al., 2016). The design dimension is 8 for TF Bind 8 and 10 for TF Bind 10, and the design spaces are both CATEGORICAL where the number of categories of all dimensions are 4. The TF Bind 8 dataset contains 32898 design-score pairs. Although the TF Bind 10 dataset in Design-Bench includes 4161482 design-score pairs, which is too large and beyond our computation budget, we sample a subset with size of 10000 following recent work in the field of offline BBO (Tan et al., 2025).

---

[9]https://github.com/RobertTLange/evosax
[10]https://github.com/brandontrabucco/design-bench
[11]https://huggingface.co/datasets/beckhamc/design_bench_data

We exclude two tasks in Design-Bench, following (Yu et al., 2024; Yun et al., 2024; Tan et al., 2025). Specifically, we exclude ChEMBL (Gaulton et al., 2012) because almost all methods produce the same oracle prediction results, as shown in (Krishnamoorthy et al., 2023; Mashkaria et al., 2023), which is not appropriate for comparison. We exclude synthetic NAS on CIFAR10 (Hinton et al., 2012) due to its high computation cost for exact evaluation over multiple seeds, which exceeds our computation resources. We also excluder Hopper Controller (Brockman et al., 2016) in Design-Bench since it has a normalization and de-normalization issue due to the stochastic policy (see `https://github.com/brandontrabucco/design-bench/issues/8#issuecomment-1086758113` for details) and recent works in offline BBO (Yu et al., 2024; Yun et al., 2024; Tan et al., 2025) do not benchmark on this task as well.

### C.2. SOO-Bench Tasks

SOO-Bench (Qian et al., 2025) is a recent benchmark for offline BBO, which not only provides more test problems but also proposes a novel method to evaluate the stability of the forward methods for offline BBO. We select the unconstraint tasks in SOO-Bench, GTOPX 2, GTOPX 3, GTOPX 4, and GTOPX 6 from GTOPX benchmark (Schlueter et al., 2021), except for hybrid 1 due to environment conflicts during installation. We use the open-source code of SOO-Bench[12] and generate task data following the default settings in the SOO-Bench paper. Specifically, for a given task, the dataset size is 1000 times the variable dimension and the data is uniformly drawn from the middle 50% of the overall distribution with respect to the score values. We set the random seed of the data generation procedure as 1 by default. Both these tasks are under `CONTINUOUS` design spaces and their evaluations are done through simulation library modules provided by (Schlueter et al., 2021)[13]. Detailed description of our selected tasks are as follows.

GTOPX encompasses a comprehensive suite of real-world space trajectory optimization problems (Izzo & Manuel López-Ibáñez, 2022; Izzo, 2010). Specifically, **GTOPX 2** encompasses the optimization of an intricate interplanetary trajectory for a Saturn rendezvous mission, featuring 22 decision variables with the objective of minimizing the total velocity increment $(\Delta V)$ required throughout the mission. **GTOPX 3** and **GTOPX 4** both address trajectory optimization for Mercury missions, where the primary objective is to minimize the total mission $\Delta V$. These problems are distinguished by their treatment of resonant flybys: GTOPX 3 explicitly excludes such maneuvers and operates in an 18-dimensional design space, while GTOPX 4 incorporates them within a 26-dimensional space. **GTOPX 6** focuses on the optimization of multi-gravity-assist trajectories targeting Comet 67P/Churyumov-Gerasimenko, employing 22 design variables to minimize the total $\Delta V$ requirements.

### C.3. Real-World Tasks

Following (Wang et al., 2024a), we evaluate our method on three real-world tasks, LunarLander, RobotPush, and Rover, and use the open-sourced dataset by (Wang et al., 2024a)[14]. Detailed information of these tasks are as follows.

**LunarLander**. LunarLander is a 12-dimensional `CONTINUOUS` task implemented in OpenAI Gym (Brockman et al., 2016)[15] that aims to learn the parameters of a controller for a lunar lander. The objective is to maximize the expected average return over 50 randomly generated environments.

**RobotPush**. RobotPush aims to minimize the distance between a designated target location and a pair of robotically-controlled objects, where there are 14 `CONTINUOUS` controllable variables, e.g., orientation and speed. The function is implemented in (Wang et al., 2018)[16] with a physics engine Box2D (Parberry, 2017).

**Rover**. Rover is a 2D trajectory optimization task that simulates a rover navigation task, which is defined by (Wang et al., 2018)[17]. The trajectory is optimized within a 60-dimensional `CONTINUOUS` unit hypercube. A cost function $c(\cdot)$ is defined in the hypercube to measure the trajectory quality, and the objective is to minimize the total cost.

---

[12]`https://github.com/zhuyiyi-123/SOO-Bench`
[13]`https://www.midaco-solver.com/index.php/about/benchmarks/gtopx`
[14]`https://drive.google.com/drive/folders/1hbxXdNM_CoON3EcfjcBGfUL21eKQdkiD?usp=sharing`
[15]`https://www.gymlibrary.dev/environments/box2d/lunar_lander`
[16]`https://github.com/zi-w/Ensemble-Bayesian-Optimization/blob/master/test_functions/push_function.py`
[17]`https://github.com/zi-w/Ensemble-Bayesian-Optimization/blob/master/test_functions/rover_function.py`

To generate diverse task trials, Wang et al. (2024a) employ a transformation from RIBBO (Song et al., 2025)[18] that introduces a scaling factor $s$ and a translation vector $\mathbf{t}$, where the transformed objective is computed as $y = s \cdot f(\mathbf{x} - \mathbf{t})$. These factors are deterministically generated based on random seeds, allowing different seeds to map to distinct task. We randomly choose one seed for one task for evaluation (100 for LunarLander, 100 for RobotPush, and 150 for Rover).

## D. Metadata in the Experiments

In Table 7, we deliver all the metadata we used in our experiments.

*Table 7.* Metadata illustration we used in the experiments.

| Task | Name | Description | Objective |
|---|---|---|---|
| Ant | Ant Morphology | a quadruped robot morphology optimization | to run as fast as possible |
| D'Kitty | D'Kitty Morphology | D'Kitty robot morphology optimization | to navigate the robot to a fixed location |
| Superconductor | Superconductor | critical temperature maximization | to design the chemical formula for a superconducting material that has a high critical temperature |
| TF Bind 8 | TF Bind 8 | DNA sequence optimization | to find the length-8 DNA sequence with maximum binding affinity with SIX6_REF_R1 transcription factor |
| TF Bind 10 | TF Bind 10 | DNA sequence optimization | to find the length-10 DNA sequence with maximum binding affinity with SIX6_REF_R1 transcription factor |
| GTOPX 2 | Cassini 2 | Complex interplanetary missions to Saturn | to achieve a rendezvous with Saturn, aiming to minimize the total velocity change |
| GTOPX 3 | Messenger (reduced) | Simulation of interplanetary missions to Mercury | to minimize the total velocity change over the course of the mission |
| GTOPX 4 | Messenger (full) | Interplanetary missions to Mercury, with resonant flybys of the planet | to minimize the total velocity change incurred throughout the mission |
| GTOPX 6 | Rosetta | Simulation of multi-gravity-assisted space missions to Comet 67P/Churyumov-Gerasimenko | to minimize the total velocity change required throughout the mission |
| LunarLander | LunarLander | Learn the parameters of a controller for a lunar lander | to maximize the mean terminal reward across a consistent batch of 50 randomly generated landscape |
| RobotPush | RobotPush | Control the robot to push items to a designated location | to minimize he distance between a predefined target location and two objects |
| Rover | Rover | 2D trajectories optimization for a rover | to design a reasonable trajectory to minimize the cost |

## E. Additional Experimental Results

In this section, we conduct additional experiments to further discover the effectiveness of UniSO. In this section, unless explicitly specified, we use EA as the default model-inner optimizer.

### E.1. Comparison to State-of-the-Art Single-Task Offline BBO Methods

To better understand the performance of UniSO methods, in this subsection, we compare improved UniSO-T, the best-performing universal offline BBO approach in Table 2, to a wide range of single-task offline BBO methods on Design-Bench (Trabucco et al., 2022). These methods are mainly based on z-score normalization for input solutions. Here the improved UniSO-T method is trained only on Design-Bench datasets and tasks, and the detailed results of compared methods are referred from (Tan et al., 2025). The objective score $y$ are normalized via global min-max normalization $y \leftarrow \frac{y - y_{\min}}{y_{\max} - y_{\min}}$, where $y_{\min}$ and $y_{\max}$ denote the lowest and highest scores in the full unobserved dataset from Design-Bench, respectively. Such a evaluation protocol under normalization is commonly adopted in offline BBO (Kumar & Levine, 2020; Trabucco et al., 2022). In Table 8, UniSO-T achieved an average rank of 9.8 across 21 expert single-task offline BBO methods, outperforming some recently proposed methods that are specifically designed for single-task offline BBO, and achieving the best results on TF-Bind-10. However, there is still improvement room for UniSO methods. Thus, how to propose better techniques and improve performance (e.g., training with more data) for universal offline BBO is a crucially important future work.

---

[18] https://github.com/songlei00/RIBBO/blob/dcfbed5326a411e8c285d226a6899d922317c7d6/problems/real_world_problem.py#L139

*Table 8.* Normalized scores in Design-Bench, where the best and runner-up results on each task are **Blue** and **Violet**. $\mathcal{D}$(best) denotes the best score in the offline dataset. All methods are trained within on task, while UniSO-T are done in a multi-task manner. Results of all compared methods are referred from (Tan et al., 2025).

| Method | Venue | Ant | D'Kitty | Superconductor | TF-Bind-8 | TF-Bind-10 | Avg. Rank |
|---|---|---|---|---|---|---|---|
| $\mathcal{D}$(best) | / | 0.565 | 0.884 | 0.400 | 0.439 | 0.467 | / |
| BO-$q$EI | | 0.812 ± 0.000 | 0.896 ± 0.000 | 0.382 ± 0.013 | 0.802 ± 0.081 | 0.628 ± 0.036 | 17.8 / 22 |
| CMA-ES | | **1.712 ± 0.754** | 0.725 ± 0.002 | 0.463 ± 0.042 | 0.944 ± 0.017 | 0.641 ± 0.036 | 11.4 / 22 |
| REINFORCE | Baselines | 0.248 ± 0.039 | 0.541 ± 0.196 | 0.478 ± 0.017 | 0.935 ± 0.049 | **0.673 ± 0.074** | 13.8 / 22 |
| Grad. Ascent | | 0.273 ± 0.023 | 0.853 ± 0.018 | 0.510 ± 0.028 | 0.969 ± 0.021 | 0.646 ± 0.037 | 11.2 / 22 |
| Grad. Ascent Mean | | 0.306 ± 0.053 | 0.875 ± 0.024 | 0.508 ± 0.019 | **0.985 ± 0.008** | 0.633 ± 0.030 | 10.6 / 22 |
| Grad. Ascent Min | | 0.282 ± 0.033 | 0.884 ± 0.018 | **0.514 ± 0.020** | 0.979 ± 0.014 | 0.632 ± 0.027 | 11.2 / 22 |
| CbAS | ICML'19 | 0.846 ± 0.032 | 0.896 ± 0.009 | 0.421 ± 0.049 | 0.921 ± 0.046 | 0.630 ± 0.039 | 15.6 / 22 |
| MINs | ICML'19 | 0.906 ± 0.024 | 0.939 ± 0.007 | 0.464 ± 0.023 | 0.910 ± 0.051 | 0.633 ± 0.034 | 12.6 / 22 |
| DDOM | ICML'23 | 0.908 ± 0.024 | 0.930 ± 0.005 | 0.452 ± 0.028 | 0.913 ± 0.047 | 0.616 ± 0.018 | 14.2 / 22 |
| BONET | ICML'23 | 0.921 ± 0.031 | 0.949 ± 0.016 | 0.390 ± 0.022 | 0.798 ± 0.123 | 0.575 ± 0.039 | 14.6 / 22 |
| GTG | NeurIPS'24 | 0.855 ± 0.044 | 0.942 ± 0.017 | 0.480 ± 0.055 | 0.910 ± 0.040 | 0.619 ± 0.029 | 13.6 / 22 |
| COMs | ICML'21 | 0.916 ± 0.026 | 0.949 ± 0.016 | 0.460 ± 0.040 | 0.953 ± 0.038 | 0.644 ± 0.052 | 9.0 / 22 |
| RoMA | ICML'21 | 0.430 ± 0.048 | 0.767 ± 0.031 | 0.494 ± 0.025 | 0.665 ± 0.000 | 0.553 ± 0.000 | 18.0 / 22 |
| IOM | NeurIPS'22 | 0.889 ± 0.034 | 0.928 ± 0.008 | 0.491 ± 0.034 | 0.925 ± 0.054 | 0.628 ± 0.036 | 12.6 / 22 |
| BDI | NeurIPS'22 | **0.963 ± 0.000** | 0.941 ± 0.000 | 0.508 ± 0.013 | 0.973 ± 0.000 | 0.658 ± 0.000 | 5.4 / 22 |
| ICT | NeurIPS'23 | 0.915 ± 0.024 | 0.947 ± 0.009 | 0.494 ± 0.026 | 0.897 ± 0.050 | 0.659 ± 0.024 | 8.8 / 22 |
| Tri-Mentoring | NeurIPS'23 | 0.891 ± 0.011 | 0.947 ± 0.005 | 0.503 ± 0.013 | 0.956 ± 0.000 | 0.662 ± 0.012 | 7.0 / 22 |
| PGS | AAAI'24 | 0.715 ± 0.046 | **0.954 ± 0.022** | 0.444 ± 0.020 | 0.889 ± 0.061 | 0.634 ± 0.040 | 13.4 / 22 |
| FGM | AISTATS'24 | 0.923 ± 0.023 | 0.944 ± 0.014 | 0.481 ± 0.024 | 0.811 ± 0.079 | 0.611 ± 0.008 | 12.8 / 22 |
| MATCH-OPT | ICML'24 | 0.933 ± 0.016 | 0.952 ± 0.008 | 0.504 ± 0.021 | 0.824 ± 0.067 | 0.655 ± 0.050 | 7.6 / 22 |
| RaM-ListNet | ICLR'25 | 0.949 ± 0.025 | **0.962 ± 0.015** | **0.517 ± 0.029** | **0.981 ± 0.012** | **0.670 ± 0.035** | **2.0 / 22** |
| **UniSO-T (Ours)** | / | 0.850 ± 0.062 | 0.915 ± 0.015 | 0.489 ± 0.062 | 0.947 ± 0.036 | **0.673 ± 0.136** | 9.8 / 22 |

### E.2. Computational Cost

In this subsection, we provide the computational resource and time for individual tasks. We conduct all our experiments a system with 4 GPUs (total computing power ~188 TFLOPS) and a 128-core CPU. Detailed computational resources are listed in Table 9. Time distribution over all tasks is provided in Fig. 7, and detailed computational time budgets can be found in Table 10.

*Table 9.* Computational resources comparison between improved UniSO-N and UniSO-T. All measurements are conducted on a system with 4 GPUs (total computing power ~188 TFLOPS) and a 128-core CPU.

| | Training | GPU memory |
|---|---|---|
| UniSO-N | 22530s | 21G |
| UniSO-T | 71225s | 89G |

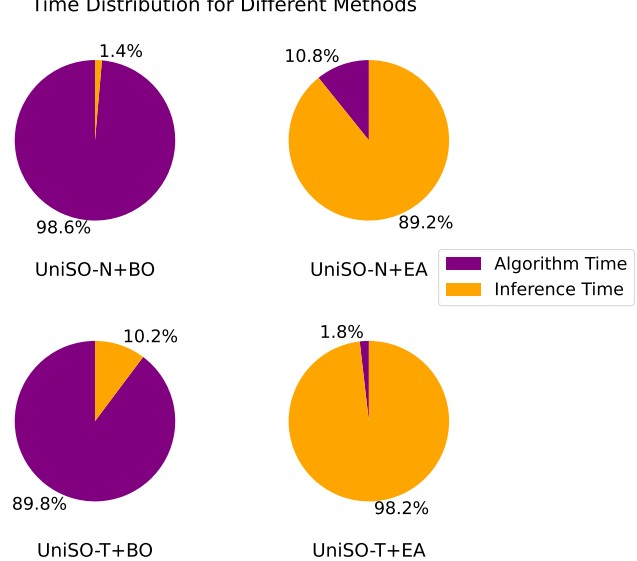

*Figure 7.* Time distribution comparison between algorithm time and model inference time for different UniSO models (UniSO-N, UniSO-T) and optimization methods (BO-$q$EI, EA). The pie charts show the percentage breakdown between algorithm time (purple) and inference time (orange) for each model. The distribution is computed over all tasks.

*Table 10.* Computational budgets of improved UniSO-N and UniSO-T with different black-box optimizers (BO-$q$EI, EA) on unconstrained tasks from Design-Bench and SOO-Bench. Each cell shows algorithm time / model inference time (in seconds). Here, BO-$q$EI and EA are allowed for an evaluation budget of 1000 (i.e., the optimizer can access the model's outputs for upmost 1000 times). EA-25600 denotes the EA with an extended evaluation budget of 25600, implemented with a population size of 128 over 200 generations, which is our default optimizer setting in the submitted version.

| Model | UniSO-N | | | UniSO-T | | |
|---|---|---|---|---|---|---|
| Search method | BO-$q$EI | EA | EA-25600 | BO-$q$EI | EA | EA-25600 |
| Ant | 74.44±0.26 / 1.87±0.00 | 0.25±0.00 / 1.94±0.00 | 2.75±0.02 / 38.16±0.00 | 74.06±0.27 / 14.81±0.00 | 0.26±0.00 / 14.55±0.00 | 2.62±0.00 / 202.05±0.06 |
| D'Kitty | 78.22±1.03 / 1.72±0.00 | 0.23±0.00 / 2.00±0.00 | 2.59±0.00 / 37.75±0.00 | 77.14±0.98 / 13.75±0.01 | 0.26±0.00 / 13.94±0.01 | 2.59±0.00 / 196.90±0.02 |
| Superconductor | 73.48±0.00 / 1.89±0.00 | 0.23±0.00 / 1.92±0.00 | 2.68±0.00 / 40.60±0.03 | 73.78±0.00 / 14.86±0.01 | 0.27±0.00 / 14.74±0.01 | 2.80±0.01 / 214.94±0.01 |
| TF Bind 8 | 59.67±0.00 / 1.58±0.00 | 0.27±0.00 / 1.77±0.00 | 3.08±0.01 / 35.02±0.00 | 60.47±0.00 / 13.02±0.06 | 0.26±0.00 / 13.17±0.07 | 2.46±0.01 / 170.63±0.00 |
| TF Bind 10 | 62.57±0.02 / 1.59±0.00 | 0.21±0.00 / 1.76±0.00 | 2.76±0.01 / 35.28±0.05 | 62.08±0.02 / 13.03±0.03 | 0.25±0.00 / 13.01±0.03 | 2.33±0.00 / 170.98±0.15 |
| GTOPX 2 | 181.67±8.44 / 1.71±0.00 | 0.21±0.00 / 1.94±0.00 | 2.34±0.00 / 35.20±0.01 | 182.48±8.26 / 13.46±0.00 | 0.25±0.00 / 13.66±0.00 | 2.49±0.02 / 182.35±0.48 |
| GTOPX 3 | 179.39±6.39 / 1.58±0.00 | 0.22±0.00 / 1.79±0.00 | 2.53±0.00 / 34.82±0.00 | 179.07±6.20 / 13.23±0.00 | 0.24±0.00 / 13.08±0.00 | 2.45±0.02 / 178.54±0.07 |
| GTOPX 4 | 189.18±39.94 / 1.62±0.00 | 0.21±0.00 / 1.84±0.00 | 2.37±0.00 / 35.75±0.01 | 189.88±40.37 / 13.04±0.00 | 0.25±0.00 / 13.28±0.00 | 2.49±0.01 / 184.38±0.00 |
| GTOPX 6 | 180.05±17.91 / 1.60±0.00 | 0.22±0.00 / 1.92±0.00 | 2.37±0.00 / 35.98±0.00 | 176.91±18.75 / 13.50±0.00 | 0.25±0.00 / 13.26±0.00 | 2.56±0.00 / 182.37±0.10 |

### E.3. UniSO-N with In-Context Transformer Neural Processes

In this subsection, we investigate integration with in-context regressors based on Transformer neural processes (TNP; Nguyen & Grover, 2022) with UniSO-N. We treat offline BBO as a single-epoch online optimization process, where top-scoring solutions in the offline dataset serve as contexts. We implement two TNP variants: (1) UniSO-N + TNP-UCB that maximizes the UCB acquisition function for one-epoch, following (Nguyen et al., 2024); (2) UniSO-N + TNP-ED that directly regresses objective scores using a single-layer MLP on TNP's hidden state outputs, following the architecture design from Section 3.2.2 of ExPT (Nguyen et al., 2023).

Table 11 demonstrates that TNP-ED outperforms TNP-UCB on 6 out of 9 tasks from Design-Bench and SOO-Bench. This shows that direct regression and score maximization is more effective than modeling the distribution and maximizing the acquisition function. We also conduct few-shot learning experiments on unseen tasks (RobotPush, Rover, and LunarLander). Table 12 shows that both TNP variants successfully exceed the best scores in the offline datasets, showing great generalization potential.

However, UniSO-N with MLP consistently achieves better performance than both TNP variants. This performance gap between MLP and TNP variants can be attributed primarily to optimization constraints in our string-based setting. While previous TNP applications leverage gradient-based optimization on model outputs or acquisition functions, the non-differentiable nature of tokenization in string-based optimization necessitates usage of black-box optimizers (Nguyen et al., 2024). This limits the optimization efficiency and the reduce the effectiveness of TNP in our paper, compared to its success in other scenarios. Future work includes exploring incorporating gradient information through techniques e.g., soft-prompt methods (Lester et al., 2021), for gradient approximation, or developing encoder-decoder architectures for legal solution string reconstruction.

*Table 11.* Comparison of improved UniSO-N with MLP and with in-context TNP (Nguyen & Grover, 2022) on unconstrained tasks from Design-Bench and SOO-Bench, where the best and runner-up results on each task are **Blue** and **Violet**. $\mathcal{D}$(best) denotes the best score in the offline dataset. Here, UniSO-N + TNP-UCB maximizes the UCB acquisition function for one epoch on trained TNP to obtain final candidates, following (Nguyen et al., 2024), and UniSO-N + TNP-ED directly regresses the score values and maximizes the model output, following Section 3.2.2 in (Nguyen et al., 2023).

| Task | $\mathcal{D}$(best) | UniSO-N + MLP | UniSO-N + TNP-UCB | UniSO-N + TNP-ED |
|---|---|---|---|---|
| Ant | 165.326 | **269.691 ± 77.425** | 110.143 ± 229.571 | **292.098 ± 229.171** |
| D'Kitty | **199.363** | **173.911 ± 46.662** | -226.316 ± 317.276 | 129.005 ± 41.107 |
| Superconductor | **74.000** | **67.333 ± 10.838** | 58.653 ± 20.171 | 64.544 ± 2.888 |
| TF Bind 8 | 0.439 | **0.833 ± 0.005** | 0.638 ± 0.000 | **0.713 ± 0.000** |
| TF Bind 10 | 0.005 | **0.959 ± 0.115** | **0.674 ± 0.000** | 0.397 ± 0.000 |
| GTOPX 2 | -195.586 | **-124.995 ± 56.170** | -183.413 ± 81.862 | **-181.144 ± 11.685** |
| GTOPX 3 | -151.190 | **-62.622 ± 22.261** | -180.053 ± 93.678 | **-99.735 ± 2.637** |
| GTOPX 4 | -215.716 | **-110.284 ± 17.559** | **-130.988 ± 38.336** | -178.791 ± 79.623 |
| GTOPX 6 | -112.599 | **-57.435 ± 18.832** | **-108.859 ± 25.493** | -195.184 ± 40.626 |
| Avg. Rank | N/A | **1.111 ± 0.314** | 2.667 ± 0.471 | **2.222 ± 0.629** |

*Table 12.* Few-shot experimental results of improved UniSO-N with MLP and with in-context TNP (Nguyen & Grover, 2022) on RobotPush, Rover, and LunarLander, where the best and runner-up results on each task are **Blue** and **Violet**. $\mathcal{D}$(best) denotes the best score in the offline dataset. Here, UniSO-N + TNP-UCB maximizes the UCB acquisition function for one epoch on trained TNP to obtain final candidates, following (Nguyen et al., 2024), and UniSO-N + TNP-ED directly regresses the score values and maximizes the model output, following Section 3.2.2 in (Nguyen et al., 2023). UniSO-N + MLP utilizes few-shot data to fine-tune MLP regressor, while UniSO-N + TNP-UCB and UniSO-N + TNP-ED directly view the few-shot data as context points.

| Task | $\mathcal{D}$(best) | UniSO-N + MLP | UniSO-N + TNP-UCB | UniSO-N + TNP-ED |
|---|---|---|---|---|
| RobotPush | 0.102 | **7.014 ± 0.000** | **3.769 ± 1.969** | 0.877 ± 1.041 |
| Rover | -16.148 | **-8.488 ± 0.003** | -12.593 ± 1.891 | **-12.038 ± 1.142** |
| LunarLander | **7.038** | **287.038 ± 0.000** | -113.493 ± 129.223 | -79.885 ± 91.442 |
| Avg. Rank | N/A | **1.000 ± 0.000** | 2.667 ± 0.471 | **2.333 ± 0.471** |

## E.4. Affects of Different Model Size

In this subsection, we compare UniSO-N with pre-trained embedders of different sizes. In Table 13, we find that UniSO-N with bigger pre-trained embedder (T5-Base) performs even worse than that with T5-Small across most tasks. Such inferior results are also reflected in training loss curve in Fig. 8, where T5-Base not only starts with higher loss but also drops limitedly throughout training, while T5-Small demonstrates consistent optimization progress.

This performance difference aligns with our observation discussed in Section 4.2 that LMs' priors may do harm to numerical optimization. Since a larger pre-trained LM embedder incorporates more prior, there may introduce more unfavourable biases for numerical optimization. Thus, how to mitigate this gap and align different modalities of natural language and numerical representation well is critical future work (Song et al., 2024b; Van Breugel & Van Der Schaar, 2024).

*Table 13.* Comparison of UniSO-N equipped with embedders of different scales (e.g., T5-Small and T5-Base) on unconstrained tasks from Design-Bench and SOO-Bench, where the better one is **Bold**. $\mathcal{D}$(best) denotes the best score in the offline dataset.

| Task | $\mathcal{D}$(best) | UniSO-N + T5-Small | UniSO-N + T5-Base |
|---|---|---|---|
| Ant | 165.326 | **269.691 ± 77.425** | 255.336 ± 209.169 |
| D'Kitty | **199.363** | 173.911 ± 46.662 | 199.186 ± 0.000 |
| Superconductor | 74.000 | 67.333 ± 10.838 | **93.816 ± 11.763** |
| TF Bind 8 | 0.439 | **0.833 ± 0.005** | 0.597 ± 0.081 |
| TF Bind 10 | 0.005 | **0.959 ± 0.115** | 0.394 ± 0.140 |
| GTOPX 2 | -195.586 | **-124.995 ± 56.170** | -163.866 ± 74.375 |
| GTOPX 3 | -151.190 | **-62.622 ± 22.261** | -71.990 ± 11.158 |
| GTOPX 4 | -215.716 | -110.284 ± 17.559 | **-96.400 ± 39.728** |
| GTOPX 6 | -112.599 | **-57.435 ± 18.832** | -88.123 ± 20.223 |
| Avg. Rank | / | **1.333 ± 0.471** | 1.667 ± 0.471 |

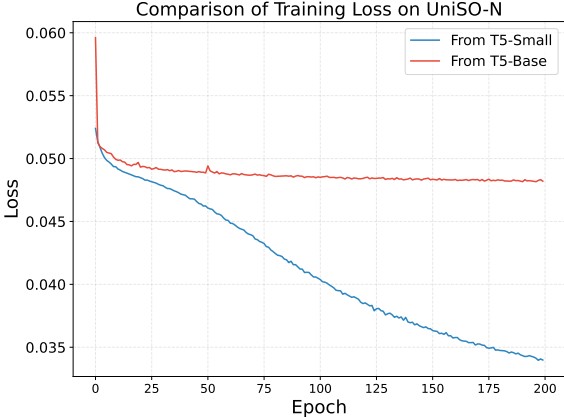

*Figure 8.* Training loss curves comparing UniSO-N with T5-Small versus T5-Base embedders during training.

## E.5. Ablation Studies on Each Component

To thoroughly evaluate the effectiveness of each component our method, we conduct ablation studies on metadata quality, loss components, loss balancing strategies, and model-inner optimizers.

For metadata quality, we examine the effectiveness different metadata components (name, description, and objective) by removing each component. As shown in Table 14, experiments across in-distribution tasks and both zero-shot and few-shot scenarios on unseen tasks demonstrate the contribution of each metadata component to overall performance. This shows the importance of metadata quality.

For loss components, we study them by removing each loss term individually. The results in Table 15 show that both loss components contribute significantly to UniSO-T's performance, with the complete loss achieving superior results compared to variants with individual losses removed.

*Table 14.* Ablation studies on different metadata components (name, description, and objective) on various tasks. All experiments are conducted based on improved UniSO-T. The first part includes tasks appear in training dataset, i.e., unconstrained tasks from Design-Bench and SOO-Bench. The remaining two parts contain unseen tasks during training, where we compare them under both zero-shot and few-shot settings. The best and runner-up results on each task are **Blue** and **Violet**. $\mathcal{D}$(best) denotes the best score in the offline dataset.

| Task | $\mathcal{D}$(best) | UniSO-T | w/o name | w/o desc. | w/o obj. | w/o metadata |
|---|---|---|---|---|---|---|
| Ant | 165.326 | **374.665 ± 56.057** | 345.369 ± 40.675 | 362.842 ± 33.008 | **363.812 ± 59.380** | 358.379 ± 64.211 |
| D'Kitty | 199.363 | 225.752 ± 8.521 | **235.715 ± 11.617** | 226.479 ± 15.311 | **245.135 ± 15.571** | 227.169 ± 12.278 |
| Superconductor | 74.000 | 92.200 ± 15.209 | 85.207 ± 6.261 | **99.113 ± 12.742** | **97.610 ± 10.297** | 90.871 ± 10.611 |
| TF Bind 8 | 0.439 | 0.903 ± 0.041 | **0.954 ± 0.025** | 0.935 ± 0.041 | 0.937 ± 0.012 | **0.950 ± 0.025** |
| TF Bind 10 | 0.005 | **0.823 ± 0.542** | **0.696 ± 0.126** | 0.664 ± 0.141 | 0.596 ± 0.148 | 0.651 ± 0.121 |
| GTOPX 2 | -195.586 | **-72.848 ± 9.576** | -97.806 ± 40.646 | **-63.670 ± 20.381** | -80.789 ± 8.908 | -79.864 ± 13.338 |
| GTOPX 3 | -151.190 | **-45.602 ± 8.433** | -50.788 ± 8.706 | **-45.981 ± 4.211** | -50.660 ± 10.713 | -48.178 ± 12.638 |
| GTOPX 4 | -215.716 | -84.271 ± 8.307 | -84.962 ± 11.300 | -92.163 ± 9.529 | **-75.233 ± 5.734** | **-79.887 ± 14.729** |
| GTOPX 6 | -112.599 | -47.794 ± 11.943 | **-42.181 ± 11.671** | **-45.591 ± 12.310** | -48.050 ± 13.901 | -45.764 ± 7.685 |
| RobotPush (zero-shot) | 0.102 | **3.171 ± 0.984** | 2.747 ± 1.455 | **3.416 ± 1.455** | 2.634 ± 0.953 | 2.517 ± 1.640 |
| Rover (zero-shot) | -16.148 | **-8.888 ± 2.119** | -11.009 ± 0.598 | -10.854 ± 0.859 | **-9.099 ± 2.202** | -9.089 ± 3.070 |
| LunarLander (zero-shot) | 7.038 | **31.186 ± 27.971** | 30.105 ± 57.577 | 30.892 ± 54.657 | **52.108 ± 47.941** | 6.251 ± 53.042 |
| RobotPush (few-shot) | 0.102 | **7.067 ± 0.169** | 7.026 ± 0.219 | **7.129 ± 0.486** | 6.310 ± 1.677 | 6.155 ± 1.495 |
| Rover (few-shot) | -16.148 | **-8.239 ± 1.270** | -8.850 ± 0.703 | **-8.084 ± 0.569** | -8.342 ± 1.573 | -10.511 ± 2.070 |
| LunarLander (few-shot) | 7.038 | **248.573 ± 45.386** | 226.726 ± 65.244 | **244.252 ± 38.329** | 233.169 ± 51.037 | 233.919 ± 60.467 |
| Avg. Rank | / | **2.333 ± 1.350** | 3.600 ± 1.451 | **2.467 ± 1.310** | 3.067 ± 1.340 | 3.533 ± 1.087 |

*Table 15.* Ablation studies of each component of the improved losses in UniSO-T few-shot performance on RobotPush, Rover, and LunarLander.

| Task | $\mathcal{D}$(best) | UniSO-T | UniSO-T w/o $\mathcal{L}_{\mathrm{lip}}$ | UniSO-T w/o $\mathcal{L}_{\mathrm{con}}$ | UniSO-T w/o $\{\mathcal{L}_{\mathrm{lip}}, \mathcal{L}_{\mathrm{con}}\}$ |
|---|---|---|---|---|---|
| Ant | 165.326 | **374.665 ± 56.057** | **292.644 ± 65.145** | 232.399 ± 66.642 | 275.216 ± 90.820 |
| D'Kitty | 199.363 | **225.752 ± 8.521** | 175.796 ± 64.079 | **224.668 ± 23.942** | 216.070 ± 23.209 |
| Superconductor | 74.000 | **92.200 ± 15.209** | **90.910 ± 6.538** | 80.910 ± 14.995 | 86.795 ± 13.466 |
| TF Bind 8 | 0.439 | 0.903 ± 0.041 | 0.916 ± 0.044 | **0.945 ± 0.049** | **0.940 ± 0.027** |
| TF Bind 10 | 0.005 | **0.823 ± 0.542** | 0.623 ± 0.062 | **0.823 ± 0.542** | **0.830 ± 0.539** |
| GTOPX 2 | -195.586 | **-72.848 ± 9.576** | -112.357 ± 39.361 | **-87.305 ± 24.428** | -132.023 ± 63.084 |
| GTOPX 3 | -151.190 | **-45.602 ± 8.433** | **-50.294 ± 6.184** | -56.067 ± 11.332 | -60.941 ± 17.235 |
| GTOPX 4 | -215.716 | **-84.271 ± 8.307** | -96.550 ± 12.745 | **-84.152 ± 15.571** | -100.943 ± 15.044 |
| GTOPX 6 | -112.599 | -47.794 ± 11.943 | -67.276 ± 24.305 | **-43.334 ± 8.402** | -71.749 ± 28.497 |
| Avg. Rank | / | **1.667 ± 0.943** | 2.889 ± 0.737 | **2.333 ± 1.155** | 3.111 ± 0.994 |

Additionally, we study the loss balancing strategy by comparing it with algorithm that remove this strategy. The comparative results in Table 16 validates the importance of loss balancing strategy over static weight assignments.

*Table 16.* Ablation studies of loss balancing strategy in improved UniSO-T on unconstrained tasks from Design-Bench and SOO-Bench, where the better one is **Bold**. $\mathcal{D}$(best) denotes the best score in the offline dataset.

| Task | $\mathcal{D}$(best) | UniSO-T w/ balance | UniSO-T w/o balance |
|---|---|---|---|
| Ant | 165.326 | **374.665 ± 56.057** | 185.597 ± 165.661 |
| D'Kitty | 199.363 | **225.752 ± 8.521** | 203.553 ± 32.507 |
| Superconductor | 74.000 | **92.200 ± 15.209** | 79.635 ± 5.777 |
| TF Bind 8 | 0.439 | 0.903 ± 0.041 | **0.929 ± 0.049** |
| TF Bind 10 | 0.005 | **0.823 ± 0.542** | 0.696 ± 0.126 |
| GTOPX 2 | -195.586 | **-72.848 ± 9.576** | -175.327 ± 73.053 |
| GTOPX 3 | -151.190 | **-45.602 ± 8.433** | -56.221 ± 18.342 |
| GTOPX 4 | -215.716 | **-84.271 ± 8.307** | -122.291 ± 54.913 |
| GTOPX 6 | -112.599 | **-47.794 ± 11.943** | -70.352 ± 25.870 |
| Avg. Rank | / | **1.111 ± 0.314** | 1.889 ± 0.314 |

In offline BBO, gradient-based model-inner optimizers are commonly used. However, they are challenging to apply in string spaces. Therefore, we explore several alternative black-box search algorithms, including EAs (Bäck, 1996; Zhou et al., 2019), BO-$q$EI (Garnett, 2023), and CMA-ES (Hansen, 2016). Implementation details are provided in Appendix B.2. We fix the evaluation budget as 1000. Since CMA-ES cannot operate in `CATEGORICAL` search space, we compare these optimizers on SOO-Bench tasks. As shown in Table 17, BO-$q$EI achieves the best performance, while EA is the runner-up.

*Table 17.* Comparison of different model-inner BBO optimizers (BO-$q$EI, CMA-ES, and EA) in improved UniSO-T. As CMA-ES cannot operate in categorical space, we conduct experimental comparison on continuous tasks from SOO-Bench, where the best and runner-up results on each task are **Blue** and **Violet**. $\mathcal{D}$(best) denotes the best score in the offline dataset. Here all optimizers are allowed for a evaluation budget of 1000 (i.e., the optimizer can access the model's outputs for upmost 1000 times) for fair comparison.

| Task | $\mathcal{D}$(best) | BO-$q$EI | CMA-ES | EA |
|---|---|---|---|---|
| GTOPX 2 | -195.586 | **-80.220 ± 11.852** | -138.716 ± 45.523 | **-99.778 ± 18.512** |
| GTOPX 3 | -151.190 | **-48.493 ± 3.745** | -81.524 ± 19.054 | **-69.278 ± 20.009** |
| GTOPX 4 | -215.716 | **-80.232 ± 13.582** | -177.559 ± 29.664 | **-132.543 ± 38.303** |
| GTOPX 6 | -112.599 | **-73.306 ± 12.892** | -101.525 ± 16.986 | **-62.024 ± 23.828** |
| Avg. Rank | / | **1.250 ± 0.433** | 3.000 ± 0.000 | **1.750 ± 0.433** |

To better understand the search behaviors of BO and EA, we visualize the optimization trajectories of both BO and EA in Fig. 9. BO demonstrates focused exploration guided by the Gaussian Process model (which captures the spectral cluster property of embeddings), while EA shows more random search behavior. The convergence curves (Fig. 10) further validates BO's superior optimization efficiency compared to EA's tendency toward local optima.

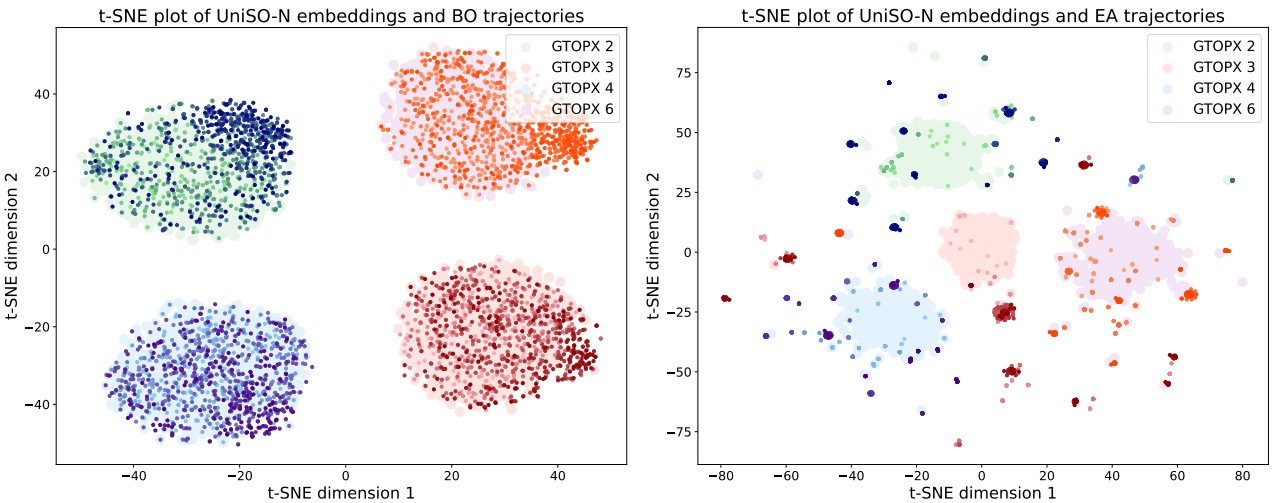

*Figure 9.* t-SNE plots of UniSO-N embedder comparing BO-$q$EI (left) and EA (right) search trajectories. Each part represents data from a single task, and small scatters represent the search trajectories of these two optimizers. Different colors represent different tasks, with darker colors indicating later stages of the search.

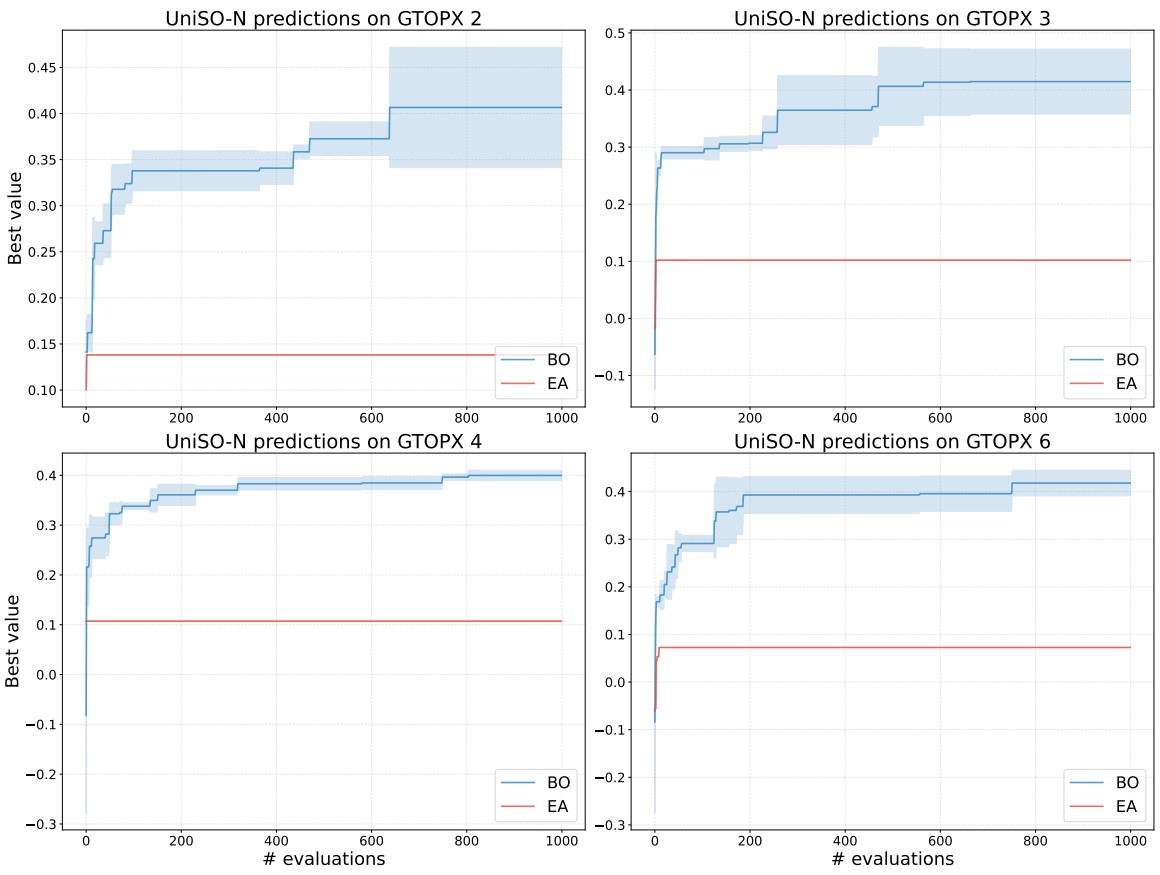

*Figure 10.* The historical best change prediction value curves of UniSO-N, with BO-$q$EI and EA as optimizers.

## E.6. UniSO + EA Results

In this subsection, we provide detailed experimental results of UniSO + EA in Tables 18 and 19.

*Table 18.* Un-normalized scores in unconstrained tasks from Design-Bench and SOO-Bench, where the best and runner-up results on each task are **Blue** and **Violet**, respectively. $\mathcal{D}$(best) denotes the best score in the offline dataset and BN represents batch normalization for numerical input designs. Both numeric-input experts and string-input UniSO methods are trained **within a single task**.

| Task | $\mathcal{D}$(best) | Numeric-input Experts | | String-input UniSO | |
|---|---|---|---|---|---|
| | | BN + EAs | BN + Grad | UniSO-T | UniSO-N |
| Ant | 165.326 | 118.877 ± 127.688 | 229.462 ± 165.869 | **245.212 ± 165.083** | **254.720 ± 102.151** |
| D'Kitty | 199.363 | 111.205 ± 66.986 | 183.263 ± 62.436 | **229.114 ± 25.766** | **188.642 ± 26.306** |
| Superconductor | 74.000 | **93.951 ± 7.039** | **97.137 ± 6.113** | 88.720 ± 9.884 | 63.930 ± 6.748 |
| TF Bind 8 | 0.439 | **0.984 ± 0.007** | **0.959 ± 0.023** | 0.948 ± 0.028 | 0.949 ± 0.000 |
| TF Bind 10 | 0.005 | **0.905 ± 0.326** | **0.888 ± 0.229** | 0.623 ± 0.115 | 0.600 ± 0.006 |
| GTOPX 2 | -195.586 | **-88.054 ± 20.878** | -128.310 ± 15.616 | **-76.479 ± 4.815** | -117.022 ± 51.671 |
| GTOPX 3 | -151.190 | **-64.028 ± 22.678** | -151.190 ± 0.000 | **-46.526 ± 13.434** | -71.784 ± 16.665 |
| GTOPX 4 | -215.716 | **-96.432 ± 10.868** | -215.716 ± 0.000 | **-87.714 ± 8.795** | -101.304 ± 18.492 |
| GTOPX 6 | -112.599 | **-64.217 ± 14.602** | -112.599 ± 0.000 | **-48.186 ± 9.486** | -80.391 ± 12.469 |
| Avg. Rank | / | **2.222 ± 1.030** | 3.000 ± 1.054 | **1.889 ± 1.100** | 2.889 ± 0.875 |

*Table 19.* Un-normalized scores in unconstrained tasks from Design-Bench and SOO-Bench, where the best and runner-up results on each task are **Blue** and **Violet**, respectively. $\mathcal{D}$(best) denotes the best score in the offline dataset and BN represents batch normalization for numerical input designs. Single-task experts are trained within one task, while UniSO-T and UniSO-N are done in a multi-task manner.

| Task | $\mathcal{D}$(best) | Single-task Experts | | UniSO-T + EA | | UniSO-N + EA | |
|---|---|---|---|---|---|---|---|
| | | BN + EAs | BN + Grad | Vanilla | Improved | Vanilla | Improved |
| Ant | 165.326 | 118.877 ± 127.688 | 229.462 ± 165.869 | **275.216 ± 90.820** | **374.665 ± 56.057** | 268.399 ± 82.858 | 269.691 ± 77.425 |
| D'Kitty | 199.363 | 111.205 ± 66.986 | 183.263 ± 62.436 | **216.070 ± 23.209** | **225.752 ± 8.521** | 130.655 ± 83.106 | 173.911 ± 46.662 |
| Superconductor | 74.000 | **93.951 ± 7.039** | **97.137 ± 6.113** | 86.795 ± 13.466 | 92.200 ± 15.209 | 81.266 ± 16.073 | 67.333 ± 16.838 |
| TF Bind 8 | 0.439 | **0.984 ± 0.007** | **0.959 ± 0.023** | 0.940 ± 0.027 | 0.903 ± 0.041 | 0.944 ± 0.016 | 0.833 ± 0.005 |
| TF Bind 10 | 0.005 | **0.905 ± 0.326** | 0.888 ± 0.229 | 0.830 ± 0.539 | 0.823 ± 0.542 | 0.603 ± 0.005 | **0.959 ± 0.115** |
| GTOPX 2 | -195.586 | **-88.054 ± 20.878** | -128.310 ± 15.616 | -132.023 ± 63.084 | **-72.848 ± 9.576** | -117.022 ± 51.671 | -124.995 ± 56.170 |
| GTOPX 3 | -151.190 | -64.028 ± 22.678 | -151.190 ± 0.000 | **-60.941 ± 17.235** | **-45.602 ± 8.433** | -88.601 ± 31.865 | -62.622 ± 22.261 |
| GTOPX 4 | -215.716 | **-96.432 ± 10.868** | -215.716 ± 0.000 | -100.943 ± 15.044 | **-84.271 ± 8.307** | -99.834 ± 20.837 | -110.284 ± 17.559 |
| GTOPX 6 | -112.599 | -64.217 ± 14.602 | -112.599 ± 0.000 | -71.749 ± 28.497 | **-47.794 ± 11.943** | -71.174 ± 12.932 | **-57.435 ± 18.832** |
| Avg. Rank | / | **3.111 ± 1.728** | 4.111 ± 1.792 | 3.667 ± 1.333 | **2.111 ± 1.663** | 4.222 ± 1.030 | 3.778 ± 1.618 |

### E.7. Integration of LLM-based Heuristics

Note that recently LLM-based automatic heuristics design shows impressive performance in many fields (Romera-Paredes et al., 2024; Novikov et al., 2025; Zheng et al., 2025), in Appendix E.7, we investigate the integration of LLM-based heuristics (Liu et al., 2024a;b;c) on UniSO-N to further improve performance. We adopt the approach in EoH (Liu et al., 2024a). Our implementation incorporates EoH from three key perspectives: (1): EoH-M: Summarizing better metadata; (2): EoH-R: Designing regularization trick to fine-tune the pre-trained LM embedder; (3): EoH-A: Implementing auxiliary loss based our improvement techniques to further enhance the model's performance.

Using Claude-3.5-Sonnet (Anthropic, 2024) as the backbone LLM, we design prompt templates following the EoH methodology. Results are provided in Fig. 11 and Table 20. The prompts are shown in Figures 16 to 24. The best individual solutions found by EoH-M, EoH-R, EoH-A are provided in Figures 25 to 27, respectively.

The experimental results show that EoH-M captures more comprehensive metadata, leading to improved overall performance. Although EoH-R underperforms, EoH-A further enhances UniSO-N's capabilities, showing the potential of LLM-based heuristics on improving the performance optimization algorithms.

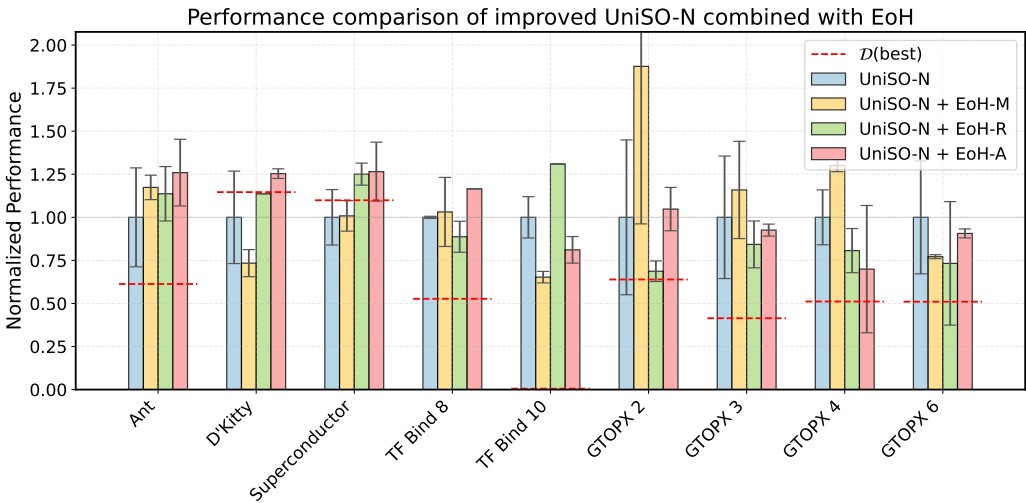

*Figure 11.* Performance comparison of improved UniSO-N combined with different EoH (Liu et al., 2024a) strategies. The bar plot shows the normalized performance of the baseline UniSO-N model and three EoH-enhanced variants (EoH-M: metadata summarization, EoH-R: regularization-based fine-tuning, EoH-A: auxiliary loss implementation) across nine optimization tasks. The red dashed lines indicate the best score $\mathcal{D}$(best) in offline dataset for each task. Error bars represent standard deviations. The data is normalized by scaling UniSO-N's performance to 1 for each task.

*Table 20.* Un-normalized scores of improved UniSO-N and its variants with different EoH (Liu et al., 2024a) strategies (EoH-M: metadata summarization, EoH-R: regularization-based fine-tuning, EoH-A: auxiliary loss implementation) in unconstrained tasks from Design-Bench and SOO-Bench, where the best and runner-up results on each task are **Blue** and **Violet**, respectively. $\mathcal{D}$(best) denotes the best score in the offline dataset.

| Task | $\mathcal{D}$(best) | UniSO-N | UniSO-N + EoH-M | UniSO-N + EoH-R | UniSO-N + EoH-A |
|---|---|---|---|---|---|
| Ant | 165.326 | 269.691 ± 77.425 | **316.515 ± 19.141** | 306.577 ± 42.614 | **339.697 ± 52.210** |
| D'Kitty | **199.363** | 173.911 ± 46.662 | 127.644 ± 13.676 | 197.777 ± 0.000 | **218.026 ± 4.858** |
| Superconductor | 74.000 | 67.333 ± 10.838 | 67.894 ± 5.964 | **84.215 ± 4.306** | **85.186 ± 11.518** |
| TF Bind 8 | 0.439 | 0.833 ± 0.005 | **0.859 ± 0.167** | 0.739 ± 0.075 | **0.971 ± 0.000** |
| TF Bind 10 | 0.005 | **0.959 ± 0.115** | 0.626 ± 0.032 | **1.256 ± 0.000** | 0.778 ± 0.074 |
| GTOPX 2 | -195.586 | -124.995 ± 56.170 | **-66.611 ± 32.466** | -181.919 ± 15.818 | **-119.284 ± 14.325** |
| GTOPX 3 | -151.190 | **-62.622 ± 22.261** | **-54.032 ± 13.158** | -74.310 ± 11.999 | -67.640 ± 2.542 |
| GTOPX 4 | -215.716 | **-110.284 ± 17.559** | **-84.778 ± 2.349** | -136.679 ± 21.707 | -157.717 ± 83.337 |
| GTOPX 6 | -112.599 | **-57.435 ± 18.832** | -74.435 ± 1.096 | -78.385 ± 38.336 | **-63.365 ± 1.828** |
| Avg. Rank | / | 2.667 ± 0.943 | **2.333 ± 1.155** | 3.000 ± 1.054 | **2.000 ± 1.054** |

# F. Attention Weights Distribution Visualization

In this section, we provide the average attention weights distribution on all tasks that we use in this paper. Fig. 12 and Fig. 13 show attention distribution of T5-Small from pre-trained and from scratch, respectively, which further validate our observation of LM's harmful priors.

In Fig. 14 and Fig. 15, we compare different pre-trained LMs (T5-Small, T5-Base, Qwen2.5-1.5B, and DeepSeek-R1-Distill-Qwen-1.5B) on GTOPX 6 and TF Bind 10 task, respectively. We find from the results that LMs that perform well on mathematical tasks shown great potential on numerical optimization tasks.

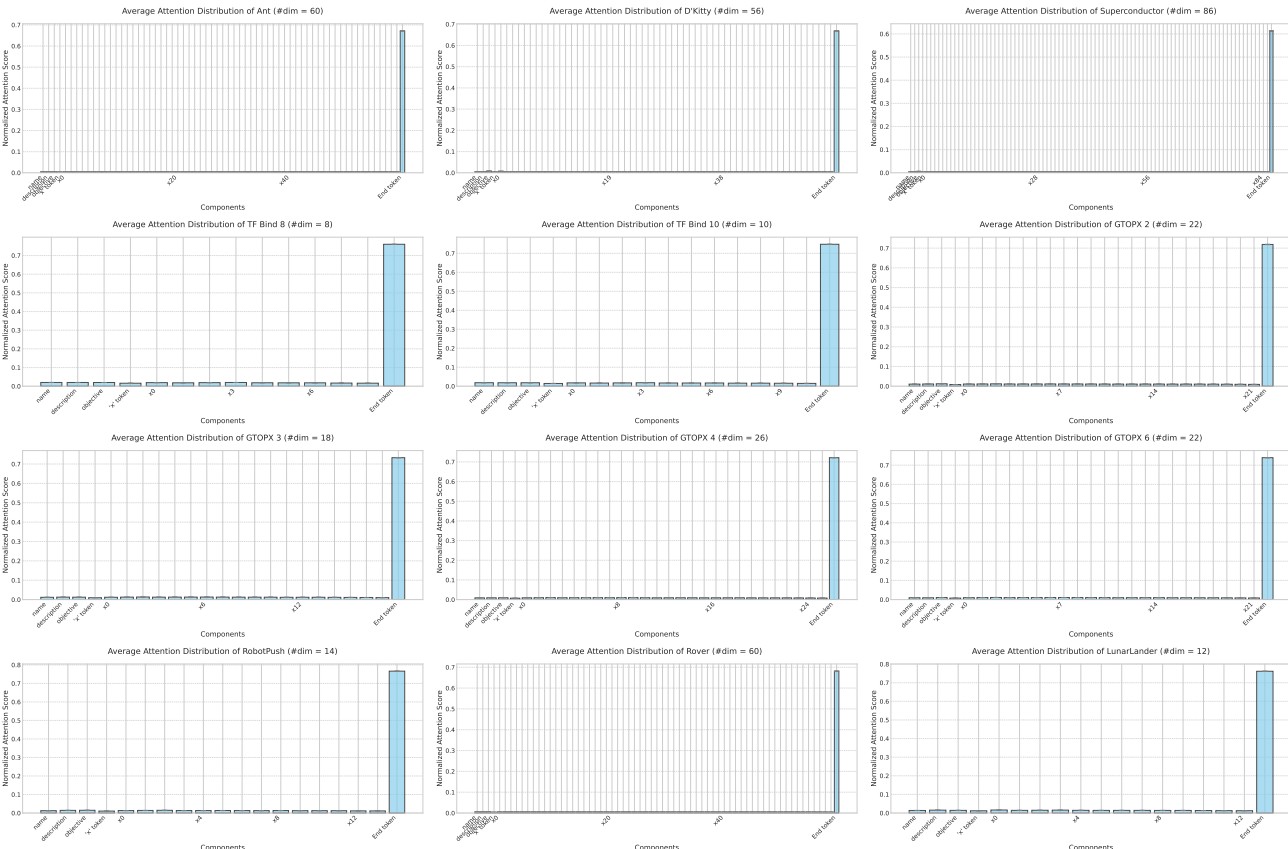

*Figure 12.* Average attention distribution of pre-trained T5-small embedder on data from different tasks. The plots present attention patterns for 12 distinct tasks: Ant, D'Kitty, Superconductor, TF Bind 8, TF Bind 10, GTOPX 2, GTOPX 3 , GTOPX 4 , GTOPX 6, RobotPush, Rover, and LunarLander, which consistently show that the pre-trained model exhibits strong attention bias towards the `EOS` token across all tasks, demonstrating the model's focus on structural elements rather than numeric-solution-relevant components.

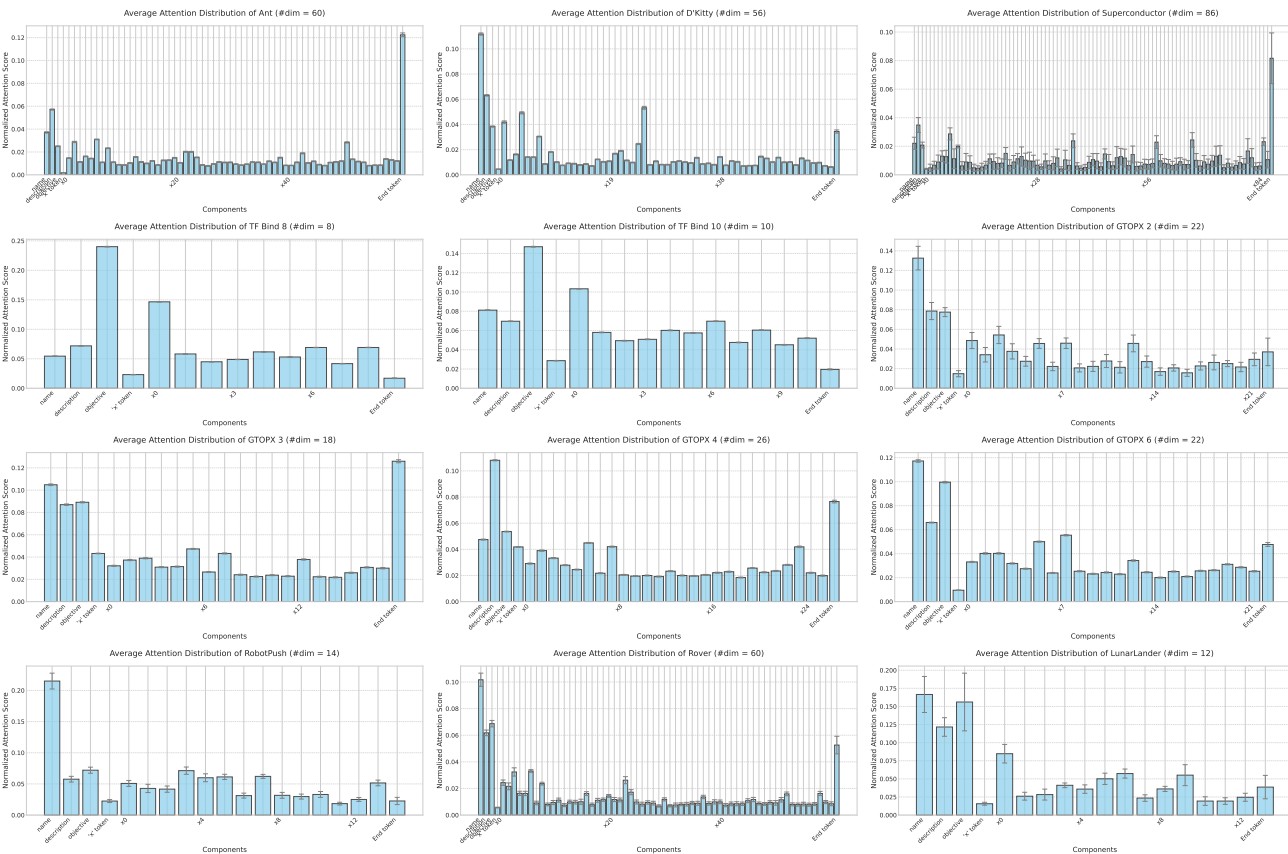

*Figure 13.* Average attention distribution of T5-small embedder trained from scratch on different tasks. The plots show attention patterns across various components for 12 different tasks: Ant, D'Kitty, Superconductor, TF Bind 8, TF Bind 10, GTOPX 2, GTOPX 3, GTOPX 4, GTOPX 6, RobotPush, Rover, and LunarLander. The consistent pattern across tasks shows more balanced attention distribution on numeric-solution-relevant components, contrasting with pre-trained models' biases towards language-based structural tokens.

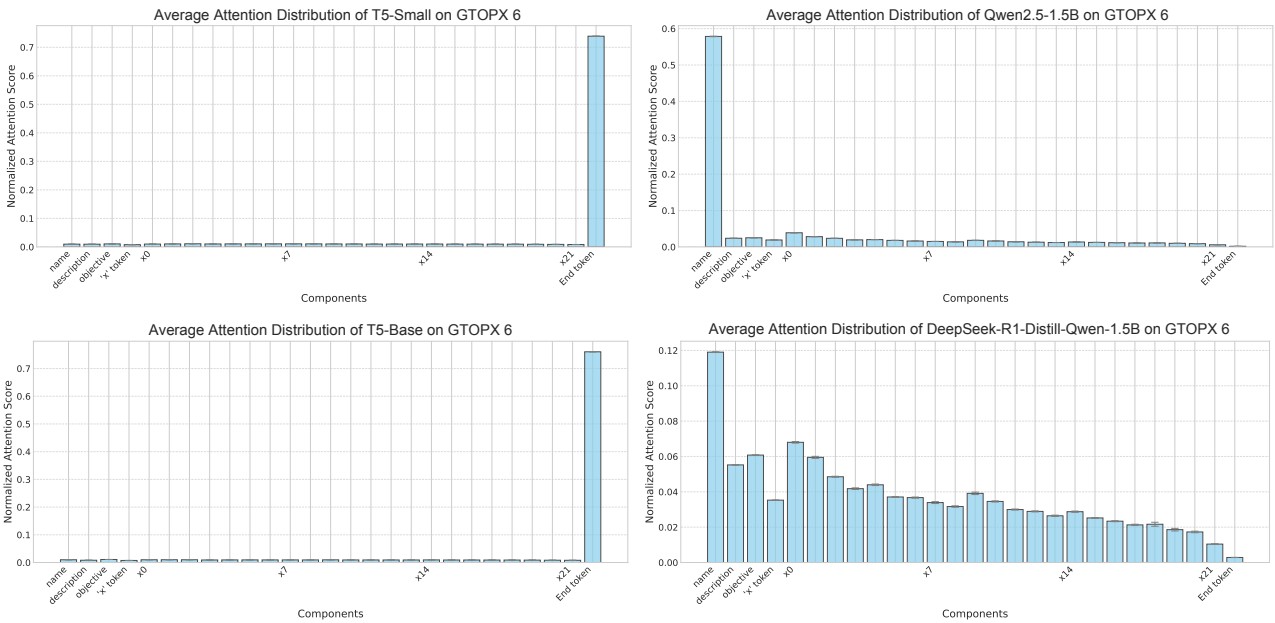

*Figure 14.* Different language models' average attention distribution on GTOPX 6 task. The bar plots compare the normalized attention scores across different components for four models: T5-Small (top left), Qwen2.5-1.5B (top right), T5-Base (bottom left), and DeepSeek-R1-Distil-Qwen-1.5B (bottom right). Here, pre-trained T5 models exhibit strong attention bias towards the `EOS` token, while the DeepSeek-R1-Distill-Qwen-1.5B shows a more balanced attention distribution across numeric-solution-relevant components compared to Qwen2.5-1.5B, demonstrating the effectiveness of knowledge transfer in optimization tasks.

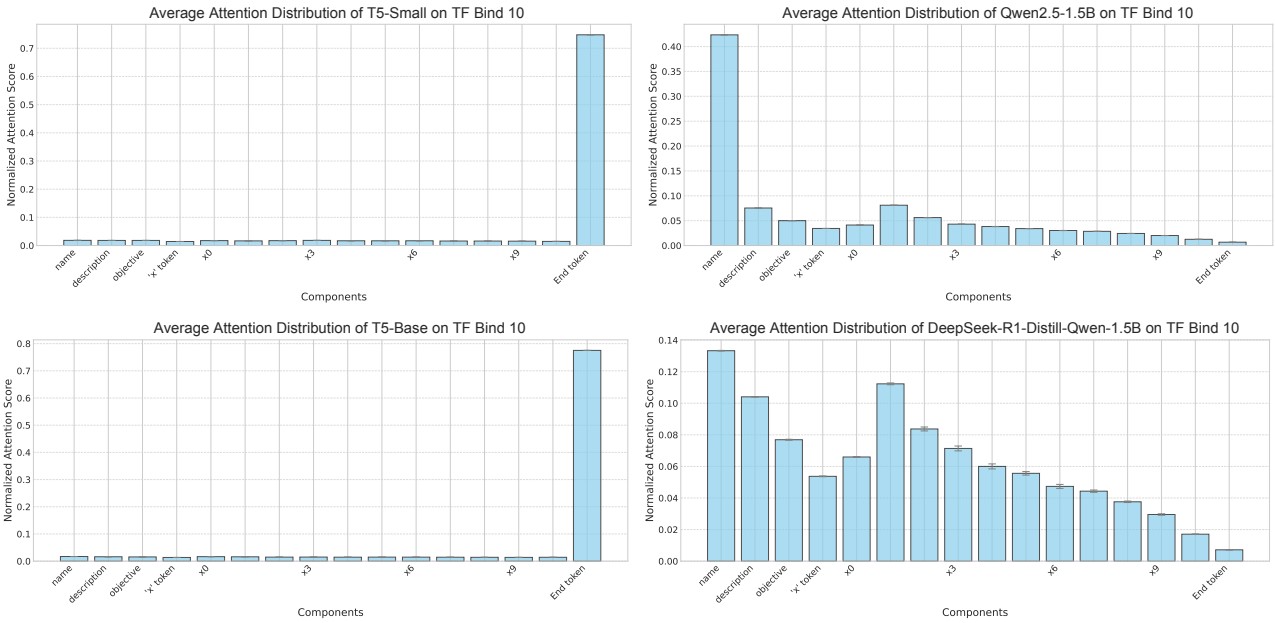

*Figure 15.* Different language models' average attention distribution on GTOPX 6 task. The bar plots compare the normalized attention scores across different components for four models: T5-Small (top left), Qwen2.5-1.5B (top right), T5-Base (bottom left), and DeepSeek-R1-Distil-Qwen-1.5B (bottom right). Similar to that in GTOPX 6, pre-trained T5 models exhibit strong attention bias towards the `EOS` token, while the DeepSeek-R1-Distill-Qwen-1.5B shows a more balanced attention distribution across numeric-solution-relevant components compared to Qwen2.5-1.5B, demonstrating the effectiveness of knowledge transfer in optimization tasks.

## G. EoH Prompts and Solutions

In this section, we provide prompts and best individual solutions for EoH (Liu et al., 2024a) integration experiments in Appendix E.7.

---

**Prompt for Initialization**

I am solving universal offline black-box optimization (BBO) problem, i.e., solving multiple offline BBO task simultaneously. My approach is based on string-based representation of x by "{x1:**,x2:**,...}". I use LM embedder to map the input string to an embedding space, and then apply an MLP head for regressing the objective. I need help generating metadata for 9 optimization tasks involving: robot morphology (Ant Morphology and D'Kitty Morphology), material science (Superconductor), DNA sequence optimization (TF Bind 8 and 10), and space mission trajectory optimization (GTOPX 2 3 4 6).

tasks information:
Ant Morphology and D'Kitty Morphology: robot morphology optimization. We created these two tasks to optimize the morphological structure of two simulated robots: Ant from OpenAI Gym and D'Kitty from ROBEL. For Ant Morphology, the goal is to optimize the morphology of an ant-shaped robot, to run as fast as possible, with a pre-trained neural network controller. For DKitty Morphology, the goal is to optimize the morphology of D'Kitty, a quadrupedal robot, such that a pre-trained neural network controller can navigate the robot to a fixed location. In this fashion, the goal for both tasks is to recover a morphology with which the pre-trained controller is compatible. The variable morphology parameters of both robots include size, orientation, and location of the limbs, giving us 60 continuous values in total for Ant and 56 for D'Kitty. To evaluate the ground-truth value for a given design, we run robotic simulation in MuJoCo for 100 timesteps, averaging 16 independent trials. These parameters are chosen to reduce stochasticity and allow the simulator to run in a minimal amount of time.
...

For each task, you need to generate 3 versions of metadata, where each metadata must be preceded by the task name. Do not generate additional explanations.

---

**Prompt for E2**

I am solving universal offline black-box optimization (BBO) problem, i.e., solving multiple offline BBO task simultaneously. My approach is based on string-based representation of x by "{x1:**,x2:**,...}". I use LM embedder to map the input string to an embedding space, and then apply an MLP head for regressing the objective. I need help generating metadata for 9 optimization tasks involving: robot morphology (AntMorphology-v0 and DKittyMorphology-v0), material science (Superconductor-v0), DNA sequence optimization (TF Bind 8 and 10), and space mission trajectory optimization (GTOPX 2 3 4 6).

tasks information:
...

I have five existing metadatas as follows:
No.1 :
Code:
...
No.3 :
Code:

Please help me design a new algorithm that is different from the given ones but can be motivated by them.
**Firstly,** identify the common backbone idea in the provided metadatas.
**Secondly,** based on the backbone idea create your new metadatas.
You need to generate 9 metadata for the 9 tasks above respectively, each metadata must be preceded by the task name, and do not generate additional explanations.

---

*Figure 16.* Two examples of prompt engineering used in initialization and E2 strategy for EoH-M.

**Prompt for E1**

I am solving universal offline black-box optimization (BBO) problem, i.e., solving multiple offline BBO task simultaneously. My approach is based on string-based representation of x by "{x1:**,x2:**,...}". I use LM embedder to map the input string to an embedding space, and then apply an MLP head for regressing the objective. I need help generating metadata for 9 optimization tasks involving: robot morphology (Ant Morphology and D'Kitty Morphology), material science (Superconductor), DNA sequence optimization (TF Bind 8 and 10), and space mission trajectory optimization (GTOPX 2 3 4 6).

tasks information:
...
I have 3 existing metadatas as follows:
...

Please help me create new metadatas that has a totally different form from the given ones. You need to generate 9 metadata for the 9 tasks above respectively, each metadata must be preceded by the task name, and do not generate additional explanations.

**Prompt for M1**

I am solving universal offline black-box optimization (BBO) problem, i.e., solving multiple offline BBO task simultaneously. My approach is based on string-based representation of x by "{x1:**,x2:**,... }". I use LM embedder to map the input string to an embedding space, and then apply an MLP head for regressing the objective. I need help generating metadata for 9 optimization tasks involving: robot morphology (Ant Morphology and D'Kitty Morphology), material science (Superconductor), DNA sequence optimization (TF Bind 8 and 10), and space mission trajectory optimization (GTOPX 2 3 4 6).

tasks information:
...

I have metadatas as follows:
No.1 :
Code:
...
No.3 :
Code:

Please assist me in creating new metadatas that has a different form but can be a modified version of the metadatas provided.
You need to generate 9 metadata for the 9 tasks above respectively, each metadata must be preceded by the task name, and do not generate additional explanations.

*Figure 17.* Two examples of prompt engineering used in E1 and M1 strategy for EoH-M.

---

**Prompt for M3**

I am solving universal offline black-box optimization (BBO) problem, i.e., solving multiple offline BBO task simultaneously. My approach is based on string-based representation of x by "{x1:**,x2:**,...}". I use LM embedder to map the input string to an embedding space, and then apply an MLP head for regressing the objective. I need help generating metadata for 9 optimization tasks involving: robot morphology (Ant Morphology and D'Kitty Morphology), material science (Superconductor), DNA sequence optimization (TF Bind 8 and 10), and space mission trajectory optimization (GTOPX 2 3 4 6).

tasks information:
...

First, you need to identify the main components in the metadatas below:
No.1 :
Code:
...
No.3 :
Code:

**Next,** analyze whether any of these components can be overfit to the in-distribution instances.
**Then,** based on your analysis, simplify the components to enhance the generalization to potential out-of-distribution instances.
**Finally,** provide the revised metadata,, you need to generate 9 metadata for the 9 tasks above respectively, each metadata must be preceded by the task name, and do not generate additional explanations.

---

*Figure 18.* Two examples of prompt engineering used in M3 strategy for EoH-M. Note that we do not have an M2 strategy, since M2 in EoH (Liu et al., 2024a) represents hyper-parameter adaptation while metadata do not have any hyper-parameters.

**Prompt for Initialization**

I am solving universal offline black-box optimization (BBO) problem, i.e., solving multiple offline BBO task simultaneously. My approach is based on string-based representation of x by "{x1:**,x2:**,...}". I use a language model embedder to map the input string to an embedding space, and then apply an MLP head for regressing the objective. However, the LM embedder contains improper bias for regression. I need to design a novel loss function to optimize the embedding representations of different tasks in the language model's latent space. The goal is to form a clear distributional structure of embeddings in the latent space while maintaining distinguishability between tasks.

I need to design a novel loss function to optimize the embedding representations of different tasks in the language model's latent space. The goal is to form a clear distributional structure of embeddings in the latent space while maintaining distinguishability between tasks.
**Firstly,** describe your new algorithm and main steps in one sentence. The description must be inside a brace.
**Next,** implement it in Python as a function named latent_loss. This function should accept two input(s): 'x_embedding', 'meta_embedding'. The function should return one output(s): loss. 'x_embedding' represents a batch of embeddings to be optimized with shape B*D (batch size * embedding dimension). 'meta_embedding' represents metadata information also provided in a batch embedding format with shape B*D. The output 'loss' is the designed loss value. Note that both 'x_embedding' and 'meta_embedding' are torch tensors with matching batch sizes. The novel loss function should be sufficiently complex to achieve effective embedding optimization while maintaining computational stability. It is important to ensure the loss computation is mathematically sound and properly scaled.

All inputs and outputs are torch.Tensor. Do not give additional explanations.

**Prompt for E2**

I am solving universal offline black-box optimization (BBO) problem, i.e., solving multiple offline BBO task simultaneously. My approach is based on string-based representation of x by "{x1:**,x2:**,...}". I use a language model embedder to map the input string to an embedding space, and then apply an MLP head for regressing the objective. However, the LM embedder contains improper bias for regression. I need to design a novel loss function to optimize the embedding representations of different tasks in the language model's latent space. The goal is to form a clear distributional structure of embeddings in the latent space while maintaining distinguishability between tasks.

I have five existing algorithms with their codes as follows:
No.1 algorithms description:
Code:
...
No.3 algorithms description:
Code:

Please help me design a new algorithm that is different from the given ones but can be motivated by them.
**Firstly,** identify the common backbone idea in the provided algorithms.
**Secondly,** based on the backbone idea describe your new algorithm in one sentence.
**Thirdly,** implement it in Python as a function named 'latent_loss'. This function should accept two inputs: 'x_embedding', 'meta_embedding . The function should return one output: 'loss'. 'x_embedding' represents a batch of embeddings to be optimized with shape B*D (batch size * embedding dimension). 'meta_embedding' represents metadata information also provided in a batch embedding format with shape B*D.The output 'loss' is the designed loss value. Note that both 'x_embedding' and 'meta_embedding' are torch tensors with matching batch sizes. The novel loss function should be sufficiently complex to achieve effective embedding optimization while maintaining computational stability. It is important to ensure the loss computation is mathematically sound and properly scaled.

All inputs and outputs are torch.Tensor. Do not give additional explanations.

*Figure 19.* Two examples of prompt engineering used in initialization and E2 strategy for EoH-R.

**Prompt for E1**

I am solving universal offline black-box optimization (BBO) problem, i.e., solving multiple offline BBO task simultaneously. My approach is based on string-based representation of x by "{x1:**,x2:**,...}". I use a language model embedder to map the input string to an embedding space, and then apply an MLP head for regressing the objective. However, the LM embedder contains improper bias for regression. I need to design a novel loss function to optimize the embedding representations of different tasks in the language model's latent space. The goal is to form a clear distributional structure of embeddings in the latent space while maintaining distinguishability between tasks.

I need to design an additional loss function to further fine-tune the embedder checkpoint. The goal is to obtain better performances of the final solutions over all tasks.

Please help me create a new algorithm that has a totally different form from the given ones.

**Firstly,** describe your new algorithm and main steps in one sentence. The description must be inside a brace.
**Next,** implement it in Python as a function named additional_loss. This function should accept two input(s): 'x_embedding', 'meta_embedding'. The function should return one output(s): loss. 'x_embedding' represents a batch of embeddings to be optimized with shape B*D (batch size * embedding dimension). 'meta_embedding' represents metadata information also provided in a batch embedding format with shape B*D. The output 'loss' is the designed loss value. Note that both 'x_embedding' and 'meta_embedding' are torch tensors with matching batch sizes. The novel loss function should be sufficiently complex to achieve effective embedding optimization while maintaining computational stability. It is important to ensure the loss computation is mathematically sound and properly scaled.

All inputs and outputs are torch.Tensor. Do not give additional explanations.

**Prompt for M1**

I am solving universal offline black-box optimization (BBO) problem, i.e., solving multiple offline BBO task simultaneously. My approach is based on string-based representation of x by "{x1:**,x2:**,...}". I use a language model embedder to map the input string to an embedding space, and then apply an MLP head for regressing the objective. However, the LM embedder contains improper bias for regression. I need to design a novel loss function to optimize the embedding representations of different tasks in the language model's latent space. The goal is to form a clear distributional structure of embeddings in the latent space while maintaining distinguishability between tasks.

I have one algorithm with its code as follows.
algorithms description:
Code:

Please assist me in creating a new algorithm that has a different form but can be a modified version of the algorithm provided.
**Firstly,** describe your new algorithm and main steps in one sentence. The description must be inside a brace.
**Next,** implement it in Python as a function named additionalloss. This function should accept two input(s): 'x_embedding', 'meta_embedding'. The function should return one output(s): loss. 'x_embedding' represents a batch of embeddings to be optimized with shape B*D (batch size * embedding dimension). 'meta_embedding' represents metadata information also provided in a batch embedding format with shape B*D. The output 'loss' is the designed loss value. Note that both 'x_embedding' and 'meta_embedding' are torch tensors with matching batch sizes. The novel loss function should be sufficiently complex to achieve effective embedding optimization while maintaining computational stability. It is important to ensure the loss computation is mathematically sound and properly scaled.

All inputs and outputs are torch.Tensor. Do not give additional explanations.

*Figure 20.* Two examples of prompt engineering used in E1 and M1 strategy for EoH-R.

**Prompt for M2**

I am solving universal offline black-box optimization (BBO) problem, i.e., solving multiple offline BBO task simultaneously. My approach is based on string-based representation of x by "{x1:**,x2:**,...}". I use a language model embedder to map the input string to an embedding space, and then apply an MLP head for regressing the objective. However, the LM embedder contains improper bias for regression. I need to design a novel loss function to optimize the embedding representations of different tasks in the language model's latent space. The goal is to form a clear distributional structure of embeddings in the latent space while maintaining distinguishability between tasks.

I need to design an additional loss function to further fine-tune the embedder checkpoint. The goal is to obtain better performances of the final solutions over all tasks.

Please identify the main algorithm parameters and assist me in creating a new algorithm that has a different parameter settings of the score function provided.

**Firstly,** describe your new algorithm and main steps in one sentence. The description must be inside a brace.
**Next,** implement it in Python as a function named latent_loss. This function should accept two input(s): 'x_embedding', 'meta_embedding'. The function should return one output(s): loss. 'x_embedding' represents a batch of embeddings to be optimized with shape B*D (batch size * embedding dimension). 'meta_embedding' represents metadata information also provided in a batch embedding format with shape B*D. The output 'loss' is the designed loss value. Note that both 'x_embedding' and 'meta_embedding' are torch tensors with matching batch sizes. The novel loss function should be sufficiently complex to achieve effective embedding optimization while maintaining computational stability. It is important to ensure the loss computation is mathematically sound and properly scaled.

All inputs and outputs are torch.Tensor. Do not give additional explanations.

**Prompt for M3**

I am solving universal offline black-box optimization (BBO) problem, i.e., solving multiple offline BBO task simultaneously. My approach is based on string-based representation of x by "{x1:**,x2:**,...}". I use a language model embedder to map the input string to an embedding space, and then apply an MLP head for regressing the objective. However, the LM embedder contains improper bias for regression. I need to design a novel loss function to optimize the embedding representations of different tasks in the language model's latent space. The goal is to form a clear distributional structure of embeddings in the latent space while maintaining distinguishability between tasks.

First, you need to identify the main components in the algorithm below:
algorithms description:
Code:

**Next,** analyze whether any of these components can be overfit to the in-distribution instances. **Then**, based on your analysis, simplify the components to enhance the generalization to potential out-of-distribution instances. **Finally**, provide the revised code, keeping the function name, inputs, and outputs unchanged.
Note that both 'x_embedding' and 'meta_embedding' are torch tensors with matching batch sizes. The novel loss function should be sufficiently complex to achieve effective embedding optimization while maintaining computational stability. It is important to ensure the loss computation is mathematically sound and properly scaled.

All inputs and outputs are torch.Tensor. Do not give additional explanations.

*Figure 21.* Two examples of prompt engineering used in M2 and M3 strategy for EoH-R.

**Prompt for Initialization**

I am solving universal offline black-box optimization (BBO) problem, i.e., solving multiple offline BBO task simultaneously. My approach is based on string-based representation of x by "{x1:**,x2:**,...}". I use a language model embedder to map the input string to an embedding space, and then apply an MLP head for regressing the objective. However, the LM embedder contains improper bias for regression. I have fine-tuned a T5-small embedder checkpoint that forms a clear distributional structure of embeddings in the latent space while maintaining distinguishability between tasks via the following implementation:

Loss code in our implementation...

I need to design an additional loss function to further fine-tune the embedder checkpoint. The goal is to obtain better performances of the final solutions over all tasks.
**Firstly,** describe your new algorithm and main steps in one sentence. The description must be inside a brace.
**Next,** implement it in Python as a function named additional_loss. This function should accept two input(s): 'x_embedding', 'meta_embedding'. The function should return one output(s): loss. 'x_embedding' represents a batch of embeddings to be optimized with shape B*D (batch size * embedding dimension). 'meta_embedding' represents metadata information also provided in a batch embedding format with shape B*D. The output 'loss' is the designed loss value. Note that both 'x_embedding' and 'meta_embedding' are torch tensors with matching batch sizes. The novel loss function should be sufficiently complex to achieve effective embedding optimization while maintaining computational stability. It is important to ensure the loss computation is mathematically sound and properly scaled.

All inputs and outputs are torch.Tensor. Do not give additional explanations.

---

**Prompt for E2**

I am solving universal offline black-box optimization (BBO) problem, i.e., solving multiple offline BBO task simultaneously. My approach is based on string-based representation of x by "{x1:**,x2:**,...}". I use a language model embedder to map the input string to an embedding space, and then apply an MLP head for regressing the objective. However, the LM embedder contains improper bias for regression. I have fine-tuned a T5-small embedder checkpoint that forms a clear distributional structure of embeddings in the latent space while maintaining distinguishability between tasks via the following implementation:

Loss code in our implementation...

I have five existing algorithms with their codes as follows:
No.1 algorithms description:
Code:
...
No.3 algorithms description:
Code:

Please help me design a new algorithm that is different from the given ones but can be motivated by them.
**Firstly,** identify the common backbone idea in the provided algorithms.
**Secondly,** based on the backbone idea describe your new algorithm in one sentence.
**Thirdly,** implement it in Python as a function named 'additional_loss'. This function should accept two inputs: 'x_embedding', 'meta_embedding . The function should return one output: 'loss'. 'x_embedding' represents a batch of embeddings to be optimized with shape B*D (batch size * embedding dimension). 'meta_embedding' represents metadata information also provided in a batch embedding format with shape B*D.The output 'loss' is the designed loss value. Note that both 'x_embedding' and 'meta_embedding' are torch tensors with matching batch sizes. The novel loss function should be sufficiently complex to achieve effective embedding optimization while maintaining computational stability. It is important to ensure the loss computation is mathematically sound and properly scaled.

All inputs and outputs are torch.Tensor. Do not give additional explanations.

*Figure 22.* Two examples of prompt engineering used in initialization and E2 strategy for EoH-A.

**Prompt for E1**

I am solving universal offline black-box optimization (BBO) problem, i.e., solving multiple offline BBO task simultaneously. My approach is based on string-based representation of x by "{x1:**,x2:**,...}". I use a language model embedder to map the input string to an embedding space, and then apply an MLP head for regressing the objective. However, the LM embedder contains improper bias for regression. I have fine-tuned a T5-small embedder checkpoint that forms a clear distributional structure of embeddings in the latent space while maintaining distinguishability between tasks via the following implementation:

Loss code in our implementation...

I need to design an additional loss function to further fine-tune the embedder checkpoint. The goal is to obtain better performances of the final solutions over all tasks.

Please help me create a new algorithm that has a totally different form from the given ones.

**Firstly,** describe your new algorithm and main steps in one sentence. The description must be inside a brace.
**Next,** implement it in Python as a function named additional_loss. This function should accept two input(s): 'x_embedding', 'meta_embedding'. The function should return one output(s): loss. 'x_embedding' represents a batch of embeddings to be optimized with shape B*D (batch size * embedding dimension). 'meta_embedding' represents metadata information also provided in a batch embedding format with shape B*D. The output 'loss' is the designed loss value. Note that both 'x_embedding' and 'meta_embedding' are torch tensors with matching batch sizes. The novel loss function should be sufficiently complex to achieve effective embedding optimization while maintaining computational stability. It is important to ensure the loss computation is mathematically sound and properly scaled.

All inputs and outputs are torch.Tensor. Do not give additional explanations.

**Prompt for M1**

I am solving universal offline black-box optimization (BBO) problem, i.e., solving multiple offline BBO task simultaneously. My approach is based on string-based representation of x by "{x1:**,x2:**,...}". I use a language model embedder to map the input string to an embedding space, and then apply an MLP head for regressing the objective. However, the LM embedder contains improper bias for regression. I have fine-tuned a T5-small embedder checkpoint that forms a clear distributional structure of embeddings in the latent space while maintaining distinguishability between tasks via the following implementation:

Loss code in our implementation...

I have one algorithm with its code as follows.
algorithms description:
Code:

Please assist me in creating a new algorithm that has a different form but can be a modified version of the algorithm provided.
**Firstly,** describe your new algorithm and main steps in one sentence. The description must be inside a brace.
**Next,** implement it in Python as a function named additional_loss. This function should accept two input(s): 'x_embedding', 'meta_embedding'. The function should return one output(s): loss. 'x_embedding' represents a batch of embeddings to be optimized with shape B*D (batch size * embedding dimension). 'meta_embedding' represents metadata information also provided in a batch embedding format with shape B*D. The output 'loss' is the designed loss value. Note that both 'x_embedding' and 'meta_embedding' are torch tensors with matching batch sizes. The novel loss function should be sufficiently complex to achieve effective embedding optimization while maintaining computational stability. It is important to ensure the loss computation is mathematically sound and properly scaled.

All inputs and outputs are torch.Tensor. Do not give additional explanations.

*Figure 23.* Two examples of prompt engineering used in E1 and M1 strategy for EoH-A.

**Prompt for M2**

I am solving universal offline black-box optimization (BBO) problem, i.e., solving multiple offline BBO task simultaneously. My approach is based on string-based representation of x by "{x1:**,x2:**,...}". I use a language model embedder to map the input string to an embedding space, and then apply an MLP head for regressing the objective. However, the LM embedder contains improper bias for regression. I have fine-tuned a T5-small embedder checkpoint that forms a clear distributional structure of embeddings in the latent space while maintaining distinguishability between tasks via the following implementation:

Loss code in our implementation...

I need to design an additional loss function to further fine-tune the embedder checkpoint. The goal is to obtain better performances of the final solutions over all tasks.

Please identify the main algorithm parameters and assist me in creating a new algorithm that has a different parameter settings of the score function provided.

**Firstly,** describe your new algorithm and main steps in one sentence. The description must be inside a brace.
**Next,** implement it in Python as a function named additional_loss. This function should accept two input(s): 'x_embedding', 'meta_embedding'. The function should return one output(s): loss. 'x_embedding' represents a batch of embeddings to be optimized with shape B*D (batch size * embedding dimension). 'meta_embedding' represents metadata information also provided in a batch embedding format with shape B*D. The output 'loss' is the designed loss value. Note that both 'x_embedding' and 'meta_embedding' are torch tensors with matching batch sizes. The novel loss function should be sufficiently complex to achieve effective embedding optimization while maintaining computational stability. It is important to ensure the loss computation is mathematically sound and properly scaled.

All inputs and outputs are torch.Tensor. Do not give additional explanations.

**Prompt for M3**

I am solving universal offline black-box optimization (BBO) problem, i.e., solving multiple offline BBO task simultaneously. My approach is based on string-based representation of x by "{x1:**,x2:**,...}". I use a language model embedder to map the input string to an embedding space, and then apply an MLP head for regressing the objective. However, the LM embedder contains improper bias for regression. I have fine-tuned a T5-small embedder checkpoint that forms a clear distributional structure of embeddings in the latent space while maintaining distinguishability between tasks via the following implementation:

Loss code in our implementation...

First, you need to identify the main components in the algorithm below:
algorithms description:
Code:

**Next,** analyze whether any of these components can be overfit to the in-distribution instances. **Then,** based on your analysis, simplify the components to enhance the generalization to potential out-of-distribution instances. **Finally,** provide the revised code, keeping the function name, inputs, and outputs unchanged.
Note that both 'x_embedding' and 'meta_embedding' are torch tensors with matching batch sizes. The novel loss function should be sufficiently complex to achieve effective embedding optimization while maintaining computational stability. It is important to ensure the loss computation is mathematically sound and properly scaled.

All inputs and outputs are torch.Tensor. Do not give additional explanations.

*Figure 24.* Two examples of prompt engineering used in M2 and M3 strategy for EoH-A.

```
# Ant Morphology
{"type": "robot_design", "dimensions": 60, "input_range": [-1.0, 1.0], "output_range":
   [0, inf], "optimization_target": "running_speed", "simulation_engine": "MuJoCo", "
   evaluation_trials": 16, "simulation_length": 100}

# D'Kitty Morphology
{"type": "robot_design", "dimensions": 56, "input_range": [-1.0, 1.0], "output_range":
   [0, inf], "optimization_target": "target_reaching", "simulation_engine": "MuJoCo", "
   evaluation_trials": 16, "simulation_length": 100}

# Superconductor
{"type": "materials_science", "dimensions": 81, "input_range": [0.0, 1.0], "output_range"
   : [0, inf], "optimization_target": "critical_temperature", "model_type": "
   random_forest", "dataset_examples": 21263}

# TF Bind 8
{"type": "dna_design", "sequence_length": 8, "alphabet": ["A", "C", "G", "T"], "
   output_range": [0, 1], "optimization_target": "binding_affinity", "search_space":
   65792, "training_examples": 32898}

# TF Bind 10
{"type": "dna_design", "sequence_length": 10, "alphabet": ["A", "C", "G", "T"], "
   output_range": [0, 1], "optimization_target": "binding_affinity", "search_space":
   1048576, "training_examples": 50000}

# GTOPX 2
{"type": "spacecraft_trajectory", "dimensions": 22, "input_range": [-1.0, 1.0], "
   output_range": [0, inf], "optimization_target": "delta_v", "mission_type": "
   saturn_rendezvous", "maneuver": "dsm"}

# GTOPX 3
{"type": "spacecraft_trajectory", "dimensions": 18, "input_range": [-1.0, 1.0], "
   output_range": [0, inf], "optimization_target": "delta_v", "mission_type": "
   mercury_transfer", "flyby_type": "non_resonant"}

# GTOPX 4
{"type": "spacecraft_trajectory", "dimensions": 26, "input_range": [-1.0, 1.0], "
   output_range": [0, inf], "optimization_target": "delta_v", "mission_type": "
   mercury_transfer", "flyby_type": "resonant"}

# GTOPX 6
{"type": "spacecraft_trajectory", "dimensions": 22, "input_range": [-1.0, 1.0], "
   output_range": [0, inf], "optimization_target": "delta_v", "mission_type": "
   comet_rendezvous", "maneuver": "multi_gravity_assist"}
```

*Figure 25.* Best metadata found by EoH-M.

```python
import torch
import torch.nn.functional as F

def latent_loss(x_embedding: torch.Tensor, meta_embedding: torch.Tensor) -> torch.Tensor:
    batch_size = x_embedding.shape[0]
    # Multi-scale kernel density estimation
    scales = torch.tensor([0.1, 0.5, 1.0], device=x_embedding.device)
    dist_matrix = torch.cdist(x_embedding, x_embedding)

    density_loss = 0
    for scale in scales:
        kernel = torch.exp(-dist_matrix / (2 * scale * scale))
        density = kernel.sum(dim=1) / batch_size
        density_loss += -torch.log(density + 1e-6).mean()

    # Diffusion distance mapping
    transition = kernel / kernel.sum(dim=1, keepdim=True)
    diffused_x = torch.matmul(transition, x_embedding)
    diffusion_loss = torch.norm(diffused_x - x_embedding, dim=1).mean()

    # Task-specific repulsion
    x_norm = F.normalize(x_embedding, dim=1)
    meta_norm = F.normalize(meta_embedding, dim=1)
    task_sim = torch.matmul(x_norm, meta_norm.t())
    task_probs = F.softmax(task_sim / 0.1, dim=1)

    task_repulsion = torch.matmul(task_probs, task_probs.t())
    repulsion_loss = torch.mean(task_repulsion * torch.exp(-dist_matrix))

    # Smooth boundary transitions
    boundary_kernel = torch.exp(-task_sim)
    boundary_density = torch.matmul(boundary_kernel, task_probs)
    smoothness_loss = torch.mean((boundary_density - task_probs)**2)

    # Combined loss with balanced weights
    loss = (0.3 * density_loss +
            0.3 * diffusion_loss +
            0.2 * repulsion_loss +
            0.2 * smoothness_loss)

    return loss
```

*Figure 26.* Best code found by EoH-R. Description by the model: the algorithm extends the local structure preservation concept by incorporating adaptive kernel-based density estimation with multi-scale diffusion distances, while utilizing task-specific repulsion forces and smooth boundary transitions through probabilistic assignment matrices.

```
1  import torch
2
3  def additional_loss(x_embedding: torch.Tensor, meta_embedding: torch.Tensor) -> torch.
       Tensor:
4      x_dist = torch.cdist(x_embedding, x_embedding)
5      m_dist = torch.cdist(meta_embedding, meta_embedding)
6
7      x_rank = torch.argsort(torch.argsort(x_dist, dim=1), dim=1).float()
8      m_rank = torch.argsort(torch.argsort(m_dist, dim=1), dim=1).float()
9
10     rank_corr = torch.mean((x_rank - m_rank)**2)
11
12     dist_ratio = x_dist / (m_dist + 1e-8)
13     consistency = torch.mean(torch.abs(dist_ratio - torch.median(dist_ratio)))
14
15     return rank_corr + 0.2 * consistency
```

*Figure 27.* Best code of EoH-A. Description by the model: balance pairwise distances in both embedding and metadata spaces while enforcing consistency through rank correlation.

