# OpenReview forum: "Towards Universal Offline Black-Box Optimization via Learning Language Model Embeddings"
_ICML.cc/2025/Conference — ICML 2025 poster_

### Official Review · Reviewer_F6Mo · 2025-02-24

**Overall Recommendation:** 5

**Summary:**

This paper explores universal offline black-box optimization (BBO) by leveraging language model-based string embeddings. Prior LLM work has two flaws: it doesn't detail paradigms to clarify their strengths and compatibility, and neglects offline BBO's unique need to prevent overfitting to limited historical data. The paper proposes a approach, UniSO, which introduces a string-based representation to unify diverse numerical spaces and enables cross-domain optimization. UniSO employs metadata-guided learning and embedding distribution alignment to enhance task discrimination and stability.

**Claims And Evidence:**

yes

**Essential References Not Discussed:**

yes

**Experimental Designs Or Analyses:**

yes

**Methods And Evaluation Criteria:**

yes

**Other Comments Or Suggestions:**

None

**Other Strengths And Weaknesses:**

### *Strength:*

1. It introduces a novel approach by leveraging **string-based representations** for universal offline black-box optimization.
2. The paper proposes a framework, UniSO, which is capable of **distinguishing dissimilar tasks** with clearer clusters, demonstrating more distinct boundaries in results compared to previous methods.
3. The proposed embedding distribution alignment via metadata and smoothness enhancement help model demonstrates **strong cross-task generalization capability**.
4. The paper conducts extensive experiments on various tasks from different benchmarks.

### *Weakness:*

1. **Computational Complexity** The paper provides limited consideration of efficiency and runtime performance. The proposed method relies on large-scale language models and evolutionary algorithms, which may introduce high computational costs. The computational cost and resource requirements may increase significantly as the problem size grows.
2. The metadata includes a brief description of the task with a concise natural language summary. Is this part manually written or generated by a model? Could this lead to **instability in metadata quality**, potentially affecting the final results?
3. The approach projects input embeddings into a shared latent space, but it does not discuss how this scales with increasing dimensionality or dataset size.

**Questions For Authors:**

### *Questions:*

1. Referring to **Weakness 1**, is there any discussion on efficiency that can be provided as feedback?
2. Referring to **Weakness 2**, is there any discussion on the details of metadata generation and its impact on performance?
3. Is there an experimental comparison of the effectiveness of the **loss balancing strategy in Section 3.5**? In the ablation study, after removing certain losses(e.g. lip, con), what form does the balanced loss take?
4. Is there any discussion on why the performance improvement of UniSO-N with the strategies proposed in the article is less than that of UniSO-T?

**Relation To Broader Scientific Literature:**

Related to some extent.

**Theoretical Claims:**

yes

---

> ### Author Rebuttal · Authors · 2025-04-01
>
> Thanks for your positive and valuable comments! Below please find our response. Corresponding experimental results can be found at https://anonymous.4open.science/api/repo/UniSO/file/F6Mo.pdf.
>
> ## R1 Computational cost
>
> Thanks for your suggestions. We have compared the computational cost in Table S1. Both the training data and the computational budget are available and affordable for academic research. We will revise to include this discussion. Thank you.
>
> ## R2 Instability in metadata quality
>
> Thanks for your insightful question! The metadata in our experiments is a manually crafted summary of the tasks rather than generated by a model, which is concluded from Design-Bench [1] and SOO-Bench [2]. Besides, to assess model performance under varying metadata quality, we conduct experiments with ablation studies on different metadata components (i.e., name, description, and objective).
>
> Table S2 shows the results on in-distribution tasks, zero- and few-shot results on unseen tasks, revealing the effectiveness of each component in metadata. We will add the detailed discussions to the revised paper. Thank you.
>
> ## R3 The approach projects input embeddings into a shared latent space, but it does not discuss how this scales with increasing dimensionality or dataset size
>
> Thanks for your valuable comment. In this work, we represent inputs of varying dimensionalities using a string format `{"x1":val1,"x2":val2,...}`. These strings are tokenized and processed through an embedder to obtain embeddings of a fixed dimension. After that, we employ a nonlinear projection to map the embeddings to the shard latent space, which is a common practice [3]. Thus, our approach is capable of handling inputs with different dimensionalities.
>
> While the approach is flexible, it may face information compression challenges when dealing with extremely high-dimensional inputs or large datasets, as the embedded size is fixed. Investigating potential scaling laws in LLMs for universal offline BBO remains an interesting direction for future research. We will add a discussion about this in the revised paper. Thank you.
>
> ## R4 Loss balancing strategy
>
> Thanks for your valuable suggestion! We conduct experiments by removing this strategy. Results in Table S3 clearly demonstrate that the strategy is beneficial. We use a similar strategy after removing certain loss (e.g., $L_{lip}$): $\frac{\partial L_{total}}{\partial\boldsymbol{\theta}}=\frac{\partial L_{main}}{\partial\boldsymbol{\theta}}+\frac{L_{main}}{L_{con}+\delta}\cdot\frac{\partial L_{con}}{\partial \boldsymbol{\theta}}$. We will revise to make it clear. Thank you.
>
> ## R5 UniSO-T vs. UniSO-N
>
> Thanks for your insightful question! Their main difference is whether the initial model is pre-trained or not:
>
> - UniSO-N uses a pre-trained T5-small encoder to encode the input strings, following [4-5];
> - Since the token representation of numeric y has a different token vocabulary from pre-trained LLM [6], UniSO-T is trained from scratch.
>
> The inferior performance of UniSO-N may stem from the inconsistent bias from different modalities between natural language and numeric representation [7]. Besides, [8] theoretically investigates the issues of different modalities.
>
> To further investigate if training from scratch would improve the performance of UniSO-N, we compare UniSO-N both from scratch and from pre-trained in Table S4, where UniSO-N trained from scratch performs better than that trained from the pre-trained encoder.
>
> To better understand this phenomenon, we analyze the rank correlation between the ground-truth scores and model predictions in the out-of-distribution (OOD) regions during the training, which is an important metric for offline BBO [9]. The results in Figure S1 show that pre-trained models initially show poor performance, and there is only a small improvement later on. How to align different modalities well is a critical future work.
>
> We will add these discussions in our revised paper. Thanks again for your insightful comments!
>
> ---
>
> **Thank you again for your encouraging comments! We hope that our response has addressed your concerns, but if we missed anything please let us know.**
>
> ### References
>
> [1] Design-Bench: Benchmarks for data-driven offline model-based optimization. ICML, 2022.
>
> [2] SOO-Bench: Benchmarks for evaluating the stability of offline black-box optimization. ICLR, 2025.
>
> [3] A simple framework for contrastive learning of visual representations. ICML, 2020.
>
> [4] Understanding LLM embeddings for regression. TMLR, 2025.
>
> [5] Predicting from strings: Language model embeddings for Bayesian optimization. arXiv, 2024.
>
> [6] OmniPred: Language models as universal regressors. TMLR, 2024.
>
> [7] Position: Why tabular foundation models should be a research priority. ICML, 2024.
>
> [8] A theory of multimodal learning. NeurIPS, 2023.
>
> [9] Offline model-based optimization by learning to rank. ICLR, 2025.

---

> > ### Comment · Reviewer_F6Mo · 2025-04-02
> >
> > Thanks the authors for very informative response. My concerns have been addressed. After reading the other reviews and responses, I raise my score to 5 to support accepting this paper.

---

> > > ### Author Response · Authors · 2025-04-08
> > >
> > > Thanks for your encouraging feedback! We are glad to hear that your concerns have been addressed. We sincerely appreciate the time and effort you have dedicated to reviewing our paper and providing thoughtful and valuable comments.
> > >
> > > Best regards
> > >
> > > Authors

---

### Official Review · Reviewer_GL3R · 2025-03-08

**Overall Recommendation:** 5

**Summary:**

This paper proposes to use recent works in LLMs as regressors for offline blackbox optimization. The key contributions are:
  * Organizing the same LLM bodies with different regression heads as UniSO-T (token output) and UniSO-N (numeric output).
  * Proposing two regularization methods, i.e. (1) a contrastive loss to make inputs from the same task clustered and (2) smoothness alignment to make sure the embeddings form a reasonable objective landscape
  * Comprehensive experiments on DesignBench to show:
    * UniSO-T generally is better than UniSO-N across all scenarios.
    * UniSO-T performs competitively in apples-to-apples comparisons in single-task settings.
    * Adding the proposed regularization makes UniSO-T even better.
    * If UniSO-T is allowed to pretrain on multiple tasks, it does even better, and showing task metadata also helps.
    * Among multiple different works in offline blackbox optimization, UniSO-T performs decently, but there is room for improvement.

**Claims And Evidence:**

Yes, I could follow all of the conclusions made by the experiments - they are very clean (i.e. A/B tests via Tables to show one setting is better than another, making everything else equal).

**Essential References Not Discussed:**

No issues - The paper's references are comprehensive.

**Experimental Designs Or Analyses:**

Yes, I checked all details. No issues here.

**Methods And Evaluation Criteria:**

Yes, DesignBench is a well known benchmark for offline blackbox optimization.

**Other Comments Or Suggestions:**

### Experimental Result Presentation
This is a minor point, but I would suggest that the authors try to make the results more aesthetic. While I can follow the tables and their implied conclusions, there are too many tables - perhaps consider balancing between using histograms, bar plots, tables, scatter plots, etc. for better presentation - it might help the average reader unfamiliar with the field (who might get bored of too many tables).
  * I understand for example, different tasks have different scales, which makes it hard to put all of them in the same figure. In this case, consider grouping, e.g. (TF-Binds in one subplot, GTOPX's in another...)

### Additional Conclusions
It's interesting how UniSO-T consistently beats UniSO-N. Is there a reason why for this (e.g. UniSO-T perhaps is more end-to-end, which is better for the model)? In any case, I suggest also raising this as an important conclusion of this work.

**Other Strengths And Weaknesses:**

### Strengths
* The regularization methods make sense, but are non-obvious (i.e. the average blackbox optimization researcher is unlikely to be aware of them), and introducing them is a strong contribution for shaping LLM-embeddings better for objective functions. These regularization methods are also fairly model-agnostic, and can be applied to any LLM model and any optimization problem.
* This paper is very clear on its experimental conclusions - I could follow every single research conclusion and agree on based on the evidence. I learned a lot from this paper as a whole.

**Questions For Authors:**

### Clarification Questions
* As mentioned - what exactly is the "expert" mentioned in Table 1? Is it just a MLP as a regression model?
* When pretraining UniSO-N over multiple tasks - how was y-normalization handled? Is it possible that requiring y-normalization on every task confuses UniSO-N?

### Research Questions
* Is it necessary to start off with a pretrained checkpoint, or does random initialization work fine too, for the T5-small model?
* Does model size matter (e.g. what happens if T5-large is used?)

**Relation To Broader Scientific Literature:**

The paper contributes well to the broader issue of "universal" regression methods for blackbox optimization. It demonstrates that recent works in LLM regressors can be used directly in optimization scenarios, and the addition of the general regularization methods improves optimization results significantly.

There are many interesting avenues of future work, e.g.
  * Pretraining such regressors over a large amount of data (rather than just the data from DesignBench)
  * Applying such techniques for more exotic search spaces (e.g. code)

**Theoretical Claims:**

N/A - there are no theoretical claims, and that's fine.

---

> ### Author Rebuttal · Authors · 2025-04-01
>
> Thanks for your encouraging and valuable comments! Below please find our response. Corresponding experimental results can be found at https://anonymous.4open.science/api/repo/UniSO/file/GL3R.pdf.
>
> ## R1 Is "expert" in Table 1 a basic MLP regressor?
>
> Yes. It refers to a numeric-input MLP regressor. We will revise to make it clear. Thank you.
>
> ## R2  Future work like more data or more exotic search spaces
>
> Thanks for your valuable comments! We wholeheartedly agree with your ideas. There are many interesting directions for future work:
>
> - Training with more data with balanced ratios from different trials;
> - Applying different modalities as metadata, instead of solely upon human-summarized text;
> - Designing efficient numeric representation or tokenization.
>
> We will keep focusing on discovering better paradigms for universal BBO. Thanks for your encouragement!
>
> ## R3  Advice for better visualization
>
> We agree that adding more styles of presentations will be better for understanding. According to your suggestions, we additionally show the generalization capability of improved UniSO-T method on unseen tasks in Figure S1, which can better show the generalization and effectiveness of few-shot finetuning. We will change other experimental presentations in our paper in the revised version. Thank you very much!
>
> ## R4 UniSO-T vs. UniSO-N
>
> Please refer to R5 of Reviewer F6Mo due to space limitations.
>
> ## R5  y-normalization
>
> We normalize y from multiple tasks following [1], as discussed in Section 3.3:
> - Apply z-score normalization inside a task to shift the score to a standard normal distribution;
> - Perform a normal distribution fitting process on the subset of data points whose values are below the median. To decrease the influence of extreme outliers, we use a transformation from percentiles to z-scores.
> - Transform the scoring distribution through sequential application of min-max scaling $y\leftarrow\frac{y-y_{min}}{y_{max}-y_{min}}$ and logarithmic transformation for re-scaling.
>
> It is quite essential for UniSO-N to normalize y, since an MLP regressor requires y normalization to prevent gradient explosion or vanishment. But objective normalization can be problematic when encountering outlier feedback values at test-time [2] (especially when few-shot evaluations are allowed). How to eliminate such issues (e.g., modeling of UniSO-T does not require y-normalization) for BBO is a promising future work.
>
> ## R6  Is it necessary to start off with a pre-trained checkpoint, or does random initialization work fine too, for the T5-small model?
>
> Thanks for your constructive question! We conduct a more detailed analysis and find that it is not necessary to start off with a pre-trained checkpoint. Pre-trained language model checkpoints contain priors from natural language, which, however, would introduce harmful bias for offline BBO.
>
> From the final performance in Table S1 and OOD rank correlation (an important metric for offline BBO [3]) curve in Figure S2, we find that UniSO-N trained from scratch consistently performs better than that from a pre-trained checkpoint. The pre-trained models initially demonstrate poor performance, and there is only little improvement later on. These biases may come from different modalities between natural language and numeric representation [4], and how to mitigate this gap and align different modalities well is critical future work in our community.
>
> We will add this discussion to our revised version. Thank you again for the insightful suggestion!
>
> ## R7  T5 model size
>
> Thanks for your insightful question! Due to computational resource constraints, we can only employ models of similar size to T5-Small. As a follow-up experiment, we further train a larger model of UniSO-N, which utilizes a T5-Base embedder. However, the performance results in Table S2 and the loss curve in Figure S3 show that equipped with a larger T5-Base embedder, UniSO-N performs even worse. Since a larger pre-trained model incorporates more prior, there may introduce more unfavourable biases for offline BBO, which also aligns with our observation in our response in R6. As for UniSO-T, we fail to train with a larger model size due to limited computational resources (since UniSO-T occupies more GPU memory than UniSO-N).
>
> Thanks for your great question! We will include more a detailed discussion on this point.
>
> ---
>
> **Thank you again for your encouraging comments! We hope that our response has addressed your concerns, but if we missed anything please let us know.**
>
> ### References
>
> [1] Predicting from strings: Language model embeddings for Bayesian optimization. arXiv, 2024.
>
> [2] OmniPred: Language models as universal regressors. TMLR, 2024.
>
> [3] Offline model-based optimization by learning to rank. ICLR, 2025.
>
> [4] Position: Why tabular foundation models should be a research priority. ICML, 2024.

---

### Official Review · Reviewer_4Ajw · 2025-03-09

**Overall Recommendation:** 5

**Summary:**

This paper introduces UniSO, a novel framework for universal offline BBO that leverages string embedding spaces to address limitations of traditional methods in heterogeneous search spaces and cross-task generalization. Using LM embeddings, it converts numerical parameters into strings and proposes two variants: UniSO-T (seq-to-seq token predictor) and UniSO-N (embedding + MLP regressor). Enhancements via metadata-guided embedding alignment (heuristic projection tricks to encourage embedding to be well-clustered) and local smoothness (heuristic tricks enforcing local embeddings to be smoothed as their objective scores) improve performance.
## update after rebuttal
### *After Discussion*

*I very much appreciate the author’s respect on my ideas in improving the work, as well as the respect for my research taste on offline BBO in ICML. Also, I appreciate the author’s hard work during the rebuttal.*

1. ***Respect and realization of my proposed cluster-utilized BBO and replacement of EA**. I again appreciate the authors’ attempts in adopting methods beyond their original EA: BO-qEI and CMA-ES, and found BO-qEI’s effectiveness. I thank the authors’ recognition of my explanations to the reason (inherent cluster-utilized GP nature), and provide visualizations to aid the claims.*
2. ***Respect and realization of my proposed mechanic interpretation and EoH-based improvement**. I highly value the author’s improvements in these trends. I believe the true power of LLM in assisting offline BBO is in these regards currently: (i) whitening problem-method’s black-box nature as possible by implicit spectral clustering properties; (ii) using LLM-based heuristics that could be better than human design. I especially appreciate the average attention visuals and the three approaches the author added, which show interpretability and effectiveness in certain regards.*
3. ***Extended Experiments on other LLMs and computational budget**. I appreciate the author’s attempts in adopting other LLM, such as Deepseek and Qwen, and provide mechanic interpretative visuals to validate the evidence. I appreciate the extended time budgets studies after my replies to the rebuttal.*
4. ***Promise in revision and code availability**. I appreciate the author’s promise in adding an algorithm block w.r.t. BO-qEI and Figure 1 in the main body, and the details provided in the appendices. I appreciate the author’s willing to incorporate so many improvements in the revised paper, and their willingness to make the code available, which would greatly contribute to the BBO community.*

*Therefore, despite requiring a big revision, the revised paper warrants ICML’s attention and I would increase my score from 2 to 5 to strongly support the paper. I'm waiting to see more details in these improvements in the revised version. I would appreciate if the authors could clearly acknowledge us anonymous reviewers in the Acknowledge Section to indicate which ideas and improvements are based on the reviews of anonymous reviewers during the ICML rebuttal period. It’s my honor to have the authors’ respect and I wish good luck for the author’s future journey in offline BBO.*

**Claims And Evidence:**

- **Claim 1**: String embeddings unify heterogeneous search spaces for universal offline BBO.
    - **Evidence**: UniSO excels on multi-task datasets (e.g., Ant Morphology, TF Bind 8), outperforming baselines (Tables 1 & 2).
- **Claim 2**: Metadata guidance and geometric regularization boost optimization stability.
    - **Evidence**: t-SNE plots (Fig. 3) show improved embedding separation; enhanced UniSO-T outperforms vanilla versions (Table 2).
- **Claim 3**: UniSO generalizes across tasks and to unseen scenarios.
    - **Evidence**: Zero/few-shot results on unseen tasks (e.g., RobotPush) surpass dataset bests and single-task experts (Table 3).

**Essential References Not Discussed:**

N/A

**Experimental Designs Or Analyses:**

- **Design**:
    - Datasets from Design-Bench and SOO-Bench (DOUBLE, CATEGORICAL spaces).
    - Baselines: single-task experts (BN+EAs, BN+Grad).
    - T5-based training; zero/few-shot tests on unseen tasks.
- **Analysis**:
    - t-SNE validates embedding improvements.
    - Ablation studies (Tables 4-7) assess metadata/loss contributions.
    - Average ranks compare method performance.

**Methods And Evaluation Criteria:**

- **Methods**:
    - String representation (e.g., JSON format).
    - UniSO-T (seq-to-seq token predictor) and UniSO-N (embedding + MLP regressor).
    - Improvements: metadata-guided alignment (heuristic projection tricks to encourage embedding to be well-clustered), embedding smoothness (heuristic tricks enforcing local embeddings to be smoothed as their objective scores).
    - Model-inner search via EAs.
- **Evaluation**:
    - Un-normalized scores compared to dataset bests and single-task experts.
    - Tasks from Design-Bench, SOO-Bench, and real-world scenarios.
    - RQs assess versatility, multi-task performance, and generalization.

**Other Comments Or Suggestions:**

I suggest including detailed EAs for each tasks, hyperparameter, training time details in the appendix.

**Other Strengths And Weaknesses:**

**Strengths**

1. Innovative string-based approach breaks single-task, fixed-dimension barriers.
2. Comprehensive experiments span multiple scenarios with good-illustrated results.
3. Code availability promised, aiding reproducibility.

**Weaknesses**

1. EA-based BOO method is a topic that doesn’t warrant ICML community’s much attention, and your methods utilize LM embeddings in a very heuristic manner. Especially, your tricks in 3.3 and 3.4 are empirical based on prior work, which is without serious provable guarantee and might be suboptimal. Nowadays, ML community either prefer provable algorithms (with assumptions over problem structure) or using LLM-based heuristics (EoH in Liu et al. (2024)) to design heuristic tricks, since human-designed heuristics are usually sub-optimal and easily defeated by LLM-based heuristics.
2. Computational cost of training/search not detailed, potentially limiting scalability.
3. The author didn’t show their evolutionary algorithms procedures in the main body, which serve as a crucial rule.
4. Since t-SNE can enable clear cluster structures, why don’t you adopt more cluster-based Learning methods or corresponding Bayesian OPT for BOO insteads of EAs?

**Questions For Authors:**

1. What explains UniSO-T vs. UniSO-N performance differences across task types (continuous vs. discrete)?
2. Have you evaluated the individual contributions of metadata components (name, description, objective)?
3. See Weakness section above.

*Reference*

Liu et al. (2024) Evolution of heuristics: Towards efficient automatic algorithm design using large language model.

**Relation To Broader Scientific Literature:**

N/A

**Theoretical Claims:**

N/A

---

> ### Author Rebuttal · Authors · 2025-04-01
>
> Thanks for your valuable comments. Below please find our response. Corresponding experimental results and references can be found at https://anonymous.4open.science/api/repo/UniSO/file/4Ajw.pdf.
>
> ## R1 EA-based BBO method doesn’t warrant ICML community’s much attention; why don’t you adopt more effective optimizers (e.g., BO) instead of EAs?
>
> We understand your concerns, but there is a misunderstanding. Our work **is not** an EA-based BBO method. We focus on offline BBO, a recent popular topic in ML community (A paper list can be found in Table S1). However, none of these studies considers the universal settings, which is a long-standing challenge in BBO.
>
> In offline BBO, multiple optimizers (e.g., gradient ascent, EAs) can be applied to search in the model. Since gradients are difficult to compute in string space, we use EA instead, which is a common practice in offline BBO. In fact, any BBO algorithms are applicable. According to your comments, we further add two methods: BO-$q$EI (based on BoTorch) and CMA-ES (based on evosax). The results in Table S2 show that BO-$q$EI performs best and the default EA used in our experiments is still a competitive method.
>
> We will add these discussions to our revised paper. Thank you.
>
> ## R2 The preference of ML community: Either prefer provable algorithms or using LLM-based heuristics
>
> Thanks for your comments. However, we beg to differ with you on this matter. A position paper at ICML’24 [1] comprehensively discusses LLM for BBO. In our opinion, there are two major branches in LLM for BBO:
>
> - Leveraging general intelligence (e.g., reasoning, coding) of LLM to design algorithms or heuristics via **natural language** [2-3]. As LLM shows impressive general intelligence, several works utilize LLM to generate optimization code and heuristics by prompt engineering based on natural language. Representative works such as FunSearch [4] and EoH [5] iteratively search for better algorithms and heuristics, and GENIUS [6] employs LLM to find better neural architecture code.
> - Leveraging the representation ability of LLM to improve the ability of learned BBO algorithms via **tokens or embeddings** [1].  This branch mainly focuses on how to efficiently utilize the inner embeddings of LLM. For example, [7] shows that when numeric $y$ is tokenized into a sequence, LLM is capable of universal regression via pre-training at scale; [8-9] show the superiority of LLM embedders for universal regression and BBO tasks, while [10-11] represent input values as tokens and thus learn in-context via LLM to solve BBO.
>
> Our work falls into the second category, especially extending the scope to universal offline BBO for the first time. We deliver a unified representation that enables universal offline BBO possible and propose two effective advancements to overcome the barriers. The first branch you mentioned could help further enhance the performance of our approach. Thanks to the recent rapid development of frameworks [3], we can conveniently test whether this idea works in the future.
>
> Additionally, the performance of our proposed method is not entirely heuristic; it is based on some theoretical foundations.
>
> - $L_{con}$ is used for embedding distribution alignment, which is actually isomorphic to the relaxed variant of InfoNCE, a popular loss in contrastive learning that has been well studied [12].
> - $L_{lip}$ aims to enhance smoothness. [13] has proven that it supports the lower bound on the correlation of the embedding space and the objective space. Besides, Figure 4 in [8] shows the importance of the Lipschitz continuity for regression at different model scales and problem dimensions.
>
> Overall, we believe that our work contributes substantially to the LLM for BBO community. We will revise to add these discussions. Thank you very much.
>
> ## R3 Computational cost
>
> Please refer to R1 of Reviewer F6Mo due to space limitations.
>
> ## R4 Details of EA
>
> Thank you for pointing this out! We are sorry for not making it clear. We initialize the population as the top-$k$ scoring designs in the dataset, where $k$ is the population size. For continuous tasks, we use simulated binary crossover and polynomial mutation; For categorical tasks, we use uniform crossover and random replacement mutation. These operators are the default operators in [14]. We will add these to our revised paper. Thank you.
>
> ## R5 UniSO-T vs. UniSO-N
>
> Please refer to R5 of Reviewer F6Mo due to space limitations.
>
> ## R6 Influence of metadata components
>
> Thanks to your suggestion, we have conducted experiments to verify the individual contributions of metadata components. The results in Table S3 show the effectiveness of each component in metadata. We will add more detailed discussions to our paper. Thank you.
>
> ---
>
> **We hope that our response has addressed your concerns, but if we missed anything please let us know.**
>
> References are provided in Table S4.

---

> > ### Comment · Reviewer_4Ajw · 2025-04-04
> >
> > I appreciate the authors’ detailed response, especially the new experiments and transparency in showing BO-qEI’s superiority, which aligns with my expectations.
> >
> > > R1.1
> > >
> > >
> > > I value the BO-qEI and CMA-ES experiments. BO-qEI’s Gaussian Process (GP) surrogate can model spatial correlations, implicitly exploiting embedding’s implicit clusters for principled optimization. For ICML (unlike PPSN or CEC), I expect a principled pipeline over black-box heuristics. Your 2024 references highlight diffusion models (DMs), reflecting community focus on LLMs and DMs. Zhang et al. (2024) connect DMs to EAs, suggesting BO-qEI or even DMs could replace EAs, aligning with ICML’s interests. In a word, I suggest replacing EC with BO-qEI, and discussed whether BO’s GP nature might help utilize the spectral cluster property of embedding—which potentially make your method more principled (the details are worth discussing).
> > >
> >
> > > R1.2
> > >
> > >
> > > Your “token embeddings” branch fits my “provable algorithm” criterion. Though not fully precise, token embeddings extract regularity via neural networks’ spectral bias, supported by *mechanistic interpretability* (e.g., Allen-Zhu & Li, 2024), and contrastive learning theory (e.g., Wen & Li, 2021; Oko et al., 2025) might help.
> > >
> > > 1. Your first branch overlaps with the second. LLM4AD’s strength lies in token embeddings over strings that even unstructured, refined via ICL-based EC techniques for task-specific updates. This adaptability (e.g., for even non-language CVRP, DNA design) relies on ICL’s contextual shifts, not prior knowledge—contradicting your R3 response to Reviewer TeSq. Also, your claim in R2 that “[10-11]… learn in-context via LLM to solve BBO” seems to be contradicted with your R3 to Reviewer TeSq?
> > > 2. Your R6 response to Reviewer GL3R—that LM prior knowledge harms due to modality differences—*undermines your abstract’s story’s conveyed message*, which tout LMs capturing unifying latent relationships. If so, I recommend you leverage the EoH approach to iteratively update LM’s token embedding, which might diminish the improper part of the priors. Also, LM’s understanding of language based on its hierarchical structure (see Allen-Zhu and Li’s Physics of LLM 1), which might be very different from BBO’s strings. A discussion over these require more studies of *mechanistic interpretability*, which is currently missing.
> > > 3. Limiting experiments to T5 (even for metadata) is puzzling when stronger models like Gemma or DeepSeek exist, despite BBO’s lightweight T5 preference. Beyond [10, 12, 13], you overlook *mechanistic interpretability* works (e.g., Allen-Zhu & Li, 2024) on LM’s properties. Besides, per Wei et al. (2023), in ICL it seems that **LLM, compared with LM**, do ICL in a way that could **forget historical knowledge better** (their Flipped-Label ICL experiments). I suggest comparing different LLM/LMs’ spectral clustering (via CLIP loss probing) by their t-SNE visuals and BBO experiments to identify optimal LLMs/LMs (e.g., could UniSO-N outperform UniSO-T with a different LM, despite modality bias?). Adopting frontier models and interpretability techniques—beyond T5 and t-SNE—could strengthen your work. LLMs’ value in offline BBO lies in whitening black-box problems for principled synergy with cluster-based BO. Performance is secondary; LLM heuristics (e.g., your Eq. 1, crafted metadata) has potential to surpass human designs—I appreciate your intention to try it.
> >
> > > R3
> > >
> > >
> > > Your caption lacks detail. Is “one task” an average? What’s the variance? Which components drive time/bit complexity—can bounds be estimated? Tables without algorithmic analysis don’t address method’s scalability, especially absent task-specific BBO comparisons.
> > >
> >
> > > R4
> > >
> > >
> > > EA details should have been in Section 3.1.4 or an appendix with discussion, as do BO-qEI and CMA-ES, while it’s currently absent. Figure 1’s pipeline needs an algorithm block detailing these methods (especially BO-qEI’s block for inner-search) for clarity—but it’s currently missing.
> > >
> >
> > > R6
> > >
> > >
> > > I appreciate the table. UniSO-N experiments might deepen your discussions about the harmful bias you proposed in your R6 to Reviewer GL3R. Token embedding *mechanistic interpretability* studies might modestly clarify the broader picture.
> > >
> >
> > I would appreciate if you see the 1-3 points in my R1.2 and show me whether collaborating with ICL of LLM in forming your metadata-supervised embedding, could help your UniSO-N (new) overcome the harmful bias you proposed.
> >
> > *References*
> >
> > - Zhang et al. (2024). *Diffusion Models are Evolutionary Algorithms.*
> > - Allen-Zhu & Li (2024). *Physics of Language Models: Parts 1-3.3.*
> > - Wen & Li (2021). *Toward Understanding the Feature Learning Process of Self-supervised Contrastive Learning.*
> > - Oko et al. (2025). *A Statistical Theory of Contrastive Pre-training and Multimodal Generative AI.*
> > - Wei et al. (2023). *Larger language models do in-context learning differently.*

---

> > > ### Author Response · Authors · 2025-04-09
> > >
> > > Thanks for your feedback! We are glad to hear that many of your concerns have been addressed in our first response. Below please find our response to your remaining concerns. Corresponding experimental results can be found at https://anonymous.4open.science/api/repo/UniSO/file/4Ajw-2.pdf.
> > >
> > > ## R1 Whether BO helps utilize the spectral cluster property
> > >
> > > We are glad to hear your appreciation! Following your suggestions, we further visualize the model-inner search trajectories of BO-$q$EI and EAs to provide a more comprehensive discussion.
> > >
> > > The results are provided in Figure S1, where different colors represent different tasks, with darker colors indicating later stages of the search. Thanks to the modeling of GP, BO's search is concentrated around task-specific areas. In contrast, EA appears randomly across the search space, resulting in lower search efficiency. We also demonstrate the best model prediction value of BO and EA during the search process. Figure S2 shows that BO’s GP nature facilitates a better performance, which is consistent with its better final results. Meanwhile, EA quickly converges to a local optimum that is difficult to improve upon. We will replace EC with BO and add these discussions in our revised paper. Thank you.
> > >
> > > ## R1.2-1 Conflicts with R3 response to Reviewer TeSq
> > >
> > > We understand your concerns, but there are some misunderstandings. R3 is intended to explain no free lunch in BBO; that is, using **only** the historical data of "x-y" **without any** task-specific information for BBO does not work. Through this claim, we aim to emphasize the importance of metadata and task information, which we believe that we are in agreement on this point: ICL you mentioned has provided information, LLM4AD requires users to describe the task, and [10-11] also use metadata and prior. We will add a detailed discussion in the revised paper to make it clear. Thank you.
> > >
> > > ## R1.2-2
> > >
> > > Thanks for your suggestion! We leverage the capabilities of LLM to achieve universal optimization. Although the models pre-trained on other tasks have some negative impacts, the overall benefits still outweigh the drawbacks.
> > >
> > > Your suggestion to leverage EoH to iteratively update is highly valuable. After careful consideration, **we incorporate EoH from three perspectives**: (1) summarizing better metadata (denoted as EoH-M); (2) designing regularization trick to fine-tune the pre-trained LM embedder (denoted as EoH-R); (3) implementing auxiliary loss based our improvement techniques to further enhance the model’s performance (denoted as EoH-A). Following the EoH paper, we design the prompt templates. We use ``Claude-3.5-Sonnet`` as the backbone LLM. Results are provided in Figure S3 and Table S1. The prompts are shown in Figures S4-12.
> > >
> > > We can conclude that: (1) EoH-M obtain more comprehensive metadata (Figure S13), leading to better overall performance; (2) EoH-A can further improve UniSO-N’s performance based on our proposed method, demonstrating the potential of LLM-based heuristics. We also provide the best results of EoH-R and EoH-A in Figures S14 and S15, respectively. Due to the space limitations of response, we will add a detailed analysis in the revised paper. Thank you very much!
> > >
> > > ## R1.2-3 & R6 Mechanistic interpretability
> > >
> > > Thanks for your insightful suggestion! To further investigate whether UniSO models focus more on numerical structure rather than component tokens in BBO strings (i.e., ‘x’ token or EOS token), we visualize the average attention weights (averaged over all heads of all layers)  from different models, following (Allen-Zhu and Li, 2023).
> > >
> > > Results in Figures S16-17 clearly demonstrate that the pre-trained LM embedder focuses more on grammar structure tokens, especially on EOS tokens, while the embedders trained from scratch focus more on solution tokens, which are important for BBO tasks. These results provide better interpretable evidence for the LLM’s harmful bias for numeric optimization. We will add these discussions in the revised paper. Thank you very much!
> > >
> > > ## R1.2-3 Other LLM
> > >
> > > Good points! According to your suggestions, we further apply the pre-trained ``Qwen2.5-1.5B`` and ``Deepseek-R1-Distill-Qwen-1.5B``. Results in Figures S18-19 show that the ability to capture BBO solution tokens is ranked as follows: DeepSeek-R1 > Qwen > T5.  Due to computational cost and time limitations, we will include more results in the revised paper. Thank you.
> > >
> > > ## R3 Detailed computational time
> > >
> > > One task means the search time for a single task. We provide detailed results as you suggested in Figure S18 and Table S2. We will add these results to the revised paper. Thank you.
> > >
> > > ## R4 Details of optimizers
> > > We are not allowed to upload the revised paper during the response period due to the ICML rules. We will add these details in the revised paper. Thank you.
> > >
> > > **We hope that our response has addressed your concerns. We will include all the added results and discussion in the final version. Thank you.**

---

### Official Review · Reviewer_TeSq · 2025-03-13

**Overall Recommendation:** 3

**Summary:**

The paper explored using LLM embeddings as the universal representation space for offline black-box optimization across different domains. The paper discussed two approaches, next-token prediction and numeric prediction, and proposed several ways to constrain the learned latent space for effective offline BBO.

**Claims And Evidence:**

The paper made three claims: (1) string-based latent space enables universal offline BBO, (2) geometric regularization of the latent space improves optimization, and (3) metadata helps generalization to unseen tasks. The ablation studies empirically support claims (2) and (3). Claim (1) is true to some extent, since they showed a single model can generalize to many optimization tasks. However, the performance of the universal model is inferior to existing task-specific approaches.

**Essential References Not Discussed:**

I did not find any missing references.

**Experimental Designs Or Analyses:**

Yes, they followed the same experimental protocol as the existing works in the literature. I did not find any issues.

**Methods And Evaluation Criteria:**

Yes, they evaluated their proposed method on commonly used datasets and tasks in offline BBO.

**Other Comments Or Suggestions:**

- Typo:
  - Line 355, The term “Improved” represents the our improvements => The term “Improved” represents our improvements
  - Line 356: regular => regularize
 - In table 2, are single-task experts the same as numeric-input experts? If this is true, please use the same name to avoid confusion.

**Other Strengths And Weaknesses:**

The main weakness is the inferior performance of the method compared to existing task-specific approaches (Table 8). The promise of having a universal model for black-box optimization is that we can train the model on massive data and exploit the relations between tasks to enhance performance on individual tasks. However, this is not sufficiently shown in the paper, as the proposed method underperforms recently published works.

**Questions For Authors:**

- In 3.1.2, the authors said "(1) utilizing a single algorithm to solve multiple offline BBO problems simultaneously solely on general design-score data is impossible due to the no free lunch theorem". Can you elaborate? LLMs can simultaneously perform many tasks via in-context learning without any metadata, why can't BBO?
- For UniSO-N, why didn't the authors train an in-context regressor similar to the Predicting from Strings paper? Training an MLP means the model has to rely on the metadata alone to distinguish between tasks.
- In 3.3, the authors said "we first encode the input strings through the encoder to derive their embeddings, while simultaneously
employing a pre-trained expert language model embedder (t5-small by default) to generate embeddings for metadata". What is the encoder? Is it not the same as the t5-small model used to encode metadata strings?
- Per the last paragraph of 3.5, in the first stage, don't you have to train the regressor too, because only so you have the main loss?

**Relation To Broader Scientific Literature:**

The paper is strongly related to prior works in universal Bayesian optimization, which rely on LLMs to serve as a universal surrogate model across different optimization tasks. This paper extended the idea to the offline BBO setting, and proposed several modifications to make the latent space suitable for offline optimization.

**Theoretical Claims:**

The paper does not have theoretical claims.

---

> ### Author Rebuttal · Authors · 2025-04-01
>
> Thanks for your valuable comments. Below please find our response.  Corresponding experimental results can be found at https://anonymous.4open.science/api/repo/UniSO/file/TeSq.pdf.
>
> ## R1 Some typos
>
> Thanks for pointing these out. We have revised them and checked them thoroughly. Thank you very much.
>
> ## R2 Are single-task experts the same as numeric-input experts?
>
> Yes, the single-task experts are numeric-input experts. We will revise it to make it clear. Thank you.
>
> ## R3 More elaboration on why LLM cannot perform well on BBO via ICL
>
> Thanks for your insightful comments. The no free lunch (NFL) theorem [1-2] states that no single algorithm can universally outperform others on average over a uniform distribution of learning problems (i.e., all problems are equally likely, with no inherent structure or bias). Crucially, LLM’s success in ICL inherently violates this uniformity assumption [3]: real-world tasks (e.g., language, vision) occupy a highly structured subspace (e.g., grammar rules, spatial locality), which LLM exploits through pre-training on non-uniform, human-aligned data distributions. In contrast, general BBO problems—when stripped of metadata—align with the NFL assumption, because the lack of prior knowledge about problem distributions (e.g., smoothness) forces algorithms to treat the space as uniform [1], rendering cross-problem generalization impossible. Thus, we introduce metadata during training, which is an attempt to explicitly break this uniformity by restricting the problem space to a structured, domain-specific subspace, thereby enabling algorithms to leverage these useful biases. We will revise it to make it clear. Thank you.
>
> ## R4 Why not train an in-context regressor similar to Predicting from Strings paper?
>
> Thanks for the valuable question. Unlike online BBO, traditional offline BBO only allows evaluation of solutions in **one batch** and does not utilize the in-context information. Following the common practice in offline BBO [4], we use MLP as the regressor. We agree that it is valuable to test the performance of an in-context regressor. According to your suggestion, we consider offline BBO as a one-epoch online BBO and implement the in-context regressor using TNP [5], treating the top-scoring offline solutions as the contexts. Following Predicting from Strings [6], we maximize the UCB acquisition through EA to obtain final candidates.
>
> Results in Table S1 show that UniSO-N + TNP performs worse than UniSO + MLP. This suggests a potential mismatch between ICL and the offline BBO setting. Extending these approaches to universal offline BBO remains an interesting future work. We will include this discussion in our revised paper. Thank you.
>
> ## R5 What is the encoder in Section 3.3?
>
> UniSO-N: encoding input strings by a pre-trained T5-small encoder; UniSO-T: training a T5 encoder-decoder from scratch and using its encoder to encode input strings; The encoders for metadata of two approaches are the pre-trained T5-small encoder. We will revise it to make it clear. Thank you.
>
> ## R6 Don’t UniSO-N train with the regressor during finetuning?
>
> Thanks for pointing this out! We are sorry for not making it clear. In fact, during finetuning, we update the embedder via $\frac{\partial L_{total}}{\partial\boldsymbol{\theta}}=\frac{\partial L_{con}}{\partial\boldsymbol{\theta}}+\frac{L_{con}}{L_{lip}+\delta}\cdot\frac{\partial L_{lip}}{\partial \boldsymbol{\theta}}$ without incorporating the main loss. We will revise it. Thank you very much.
>
> ---
>
> **We hope that our response has addressed your concerns, but if we missed anything please let us know.**
>
> ### References
>
> [1] No free lunch theorems for optimization. IEEE TEvC, 1997.
>
> [2] The supervised learning no-free-lunch theorems. Soft computing and industry: Recent applications, 2002.
>
> [3] Position: The no free lunch theorem, Kolmogorov complexity, and the role of inductive biases in machine learning. ICML, 2024.
>
> [4] Offline model-based optimization: Comprehensive review. arXiv, 2025.
>
> [5] Transformer neural processes: Uncertainty-aware meta learning via sequence modeling. ICML, 2022.
>
> [6] Predicting from strings: Language Model embeddings for Bayesian optimization. arXiv, 2024.

---

> > ### Comment · Reviewer_TeSq · 2025-04-05
> >
> > I thank the authors for the response and additional experiments. I will retain my borderline score because even though the paper proposes some interesting ideas, the empirical performance is still much worse than state-of-the-art methods.
> >
> > Regarding using an in-context regressor for offline optimization, there is a cleaner approach than what you did. You train the TNP model over many different tasks in the offline data by asking the model to regress the score value of the target points given the context points. After training, you can adapt this model to a new task by using a few labeled data points from this task as context, and perform gradient ascent/descent on the input of the target point.

---

> > > ### Author Response · Authors · 2025-04-08
> > >
> > > Thank you for your positive ratings and for finding our ideas interesting!
> > >
> > > Our work introduces a universal string-based offline BBO framework, paving the way for general-purpose BBO algorithms. UniSO can simultaneously address multiple offline BBO tasks and efficiently transfer to unseen tasks, whereas previous offline methods lack such capability. Those well-designed methods focus on solving a single problem and do not consider how to scale to heterogeneous tasks. In our current implementation, we only use data from Design-Bench and SOO-Bench, whose task diversity and dataset size are relatively limited compared to current practices in language and vision; this may prevent the LLM-based architecture from reaching its full potential. Therefore, as the first universal offline BBO work, although we outperform $\mathcal{D}$(best) and multiple recent methods, surpassing the current SOTA methods on tasks for which they were specifically trained remains challenging. Incorporating more diverse data into the pre-training phase is an important future work.
> > >
> > > As for the UniSO-N + TNP experiment, we agree that directly regressing and maximizing the score value is cleaner than modeling the distribution and maximizing the acquisition function. Following TNP-ED outlined in Sec. 3.2.2 of ExPT paper [1], which also aims to directly regress score values, we apply a 1-layer MLP on TNP’s outputs to map the values and use MSE to train the model. After training, we maximize the model’s output to obtain candidate solutions. We use the default parameters as in the TNP paper for training.
> > >
> > > Table S1 at https://anonymous.4open.science/api/repo/UniSO/file/TeSq_final_response.pdf shows that UniSO-N + TNP-ED outperforms UniSO-N + TNP-UCB (i.e., vanilla TNP with maximizing UCB acquisition in our first response) in 6/9 tasks appeared in training data. Besides, few-shot experiments on unseen tasks in Table S2 show that UniSO-N + MLP and UniSO-N + TNP exceed $\mathcal{D}$(best), demonstrating that UniSO-N is sufficient to distinguish tasks. However, there is still improvement room for UniSO-N + TNP compared to UniSO-N + MLP. This performance of using TNP is not as good as that in other scenarios [1, 2], probably due to the employed different optimizers. In [1, 2], the authors obtained the next candidates via applying gradient search on the model output or acquisition function. However, string-based universal optimization restricts the utilization of gradient, as tokenization is non-differentiable [3]. Thus, following [3], we use black-box optimizer to maximize the model’s outputs, which is convenient but lacks efficiency compared to gradient ascent. We will consider incorporating gradient information in future work, including leveraging soft-prompt techniques [4] to approximate the gradients or training an encoder-decoder structure to reconstruct a legal solution string from embedding.
> > >
> > > **We will add this discussion in the revised version. Thank you again for your appreciation and constructive suggestions!**
> > >
> > > ---
> > >
> > > ### References
> > >
> > > [1] ExPT: Synthetic pretraining for few-shot experimental design. NeurIPS, 2023.
> > >
> > > [2] Transformer neural processes: Uncertainty-aware meta learning via sequence modeling. ICML, 2022.
> > >
> > > [3] Predicting from strings: Language model embeddings for Bayesian optimization. arXiv, 2024.
> > >
> > > [4] The power of scale for parameter-efficient prompt tuning. EMNLP, 2021.

---

### Decision · Program_Chairs · 2025-05-01

**Decision:**

Accept (poster)

**Comment:**

The paper considers the problem of offline black-box optimization with the goal of handling diverse tasks with heterogeneous search spaces. The key idea is to convert numerical design parameters into string sequences and utilize LLM embeddings as the representation space for surrogate modeling. Two model architectures are explored in the paper: UniSO-T, which predicts objective score tokens in a sequence-to-sequence manner, and UniSO-N, which regresses a numerical score from the LM embeddings using an MLP head. The paper received generally positive reviews from all the reviewers. Therefore, I recommend accepting the paper.

However, I would like to point out a troubling trend  in the offline optimization literature of  not referencing any work from the online black-box optimization (primarily bayesian optimization (BO) ) literature. All the problems discussed in the paper (mixed spaces, multi-task, varying input dimensions) are widely studied in BO literature but not referenced at all in the paper. I hope the authors will add these references and properly situate the proposed paper within a large body of work in the BO literature studied over last 5-10 years.